# A FINE-GRAINED SPECTRAL PERSPECTIVE ON NEURAL NETWORKS

## ABSTRACT

Are neural networks biased toward simple functions? Does depth always help learn more complex features? Is training the last layer of a network as good as training all layers? How to set the range for learning rate tuning? These questions seem unrelated at face value, but in this work we give all of them a common treatment from the spectral perspective. We will study the spectra of the *Conjugate Kernel, CK,* (also called the *Neural Network-Gaussian Process Kernel*), and the *Neural Tangent Kernel, NTK*. Roughly, the CK and the NTK tell us respectively "what a network looks like at initialization" and "what a network looks like during and after training." Their spectra then encode valuable information about the initial distribution and the training and generalization properties of neural networks. By analyzing the eigenvalues, we lend novel insights into the questions put forth at the beginning, and we verify these insights by extensive experiments of neural networks. We believe the computational tools we develop here for analyzing the spectra of CK and NTK serve as a solid foundation for future studies of deep neural networks. We have open-sourced the code for it and for generating the plots in this paper at `github.com/jxVmnLgedVwv6mNcGCBy/NNspectra`.

## 1 INTRODUCTION

Understanding the behavior of neural networks and why they generalize has been a central pursuit of the theoretical deep learning community. Recently, Valle-Pérez et al. (2018) observed that neural networks have a certain "simplicity bias" and proposed this as a solution to the generalization question. One of the ways with which they argued that this bias exists is the following experiment: they drew a large sample of boolean functions by randomly initializing neural networks and thresholding the output. They observed that there is a bias toward some "simple" functions which get sampled disproportionately more often. However, their experiments were only done for relu networks. Can one expect this "simplicity bias" to hold universally, for any architecture?

*A priori*, this seems difficult, as the nonlinear nature seems to present an obstacle in reasoning about the distribution of random networks. However, this question turns out to be more easily treated if we allow the *width to go to infinity*. A long line of works starting with Neal (1995) and extended recently by Lee et al. (2018); Novak et al. (2018); Yang (2019) have shown that randomly initialized, infinite-width networks are distributed as Gaussian processes. These Gaussian processes also describe finite width random networks well (Valle-Pérez et al., 2018). We will refer to the corresponding kernels as the *Conjugate Kernels* (CK), following the terminology of Daniely et al. (2016). Given the CK $K$, the simplicity bias of a wide neural network can be read off quickly from the *spectrum of $K$*: If the largest eigenvalue of $K$ accounts for most of $\operatorname{tr} K$, then a typical random network looks like a function from the top eigenspace of $K$.

In this paper, we will use this spectral perspective to probe not only the simplicity bias, but more generally, questions regarding how hyperparameters affect the generalization of neural networks.

Via the usual connection between Gaussian processes and linear models with features, the CK can be thought of as the kernel matrix associated to training only the last layer of a wide randomly initialized network. It is a remarkable recent advance (Jacot et al., 2018; Allen-Zhu et al., 2018a;c; Du et al., 2018) that, under a certain regime, a wide neural network of any depth evolves like a linear model even when training all parameters. The associated kernel is call the *Neural Tangent Kernel*, which is typically different from CK. While its theory was initially derived in the infinite width setting, Lee et al. (2019) confirmed with extensive experiment that this limit is predictive of finite width neural networks as well. Thus, just as the CK reveals information about *what a network looks like at*

*initialization*, NTK reveals information about *what a network looks like after training.* As such, if we can understand how hyperparameters change the NTK, we can also hope to understand how they affect the performance of the corresponding finite-width network.

**Our Contributions**    In this paper, in addition to showing that the simplicity bias is not universal, we will attempt a first step at understanding the effects of the hyperparameters on generalization from a spectral perspective.

At the foundation is a spectral theory of the CK and the NTK on the boolean cube. In Section 3, we show that these kernels, as *integral operators* on functions over the boolean cube, are diagonalized by the natural Fourier basis, echoing similar results for over the sphere (Smola et al., 2001). We also partially diagonalize the kernels over standard Gaussian, and show that, as expected, the kernels over the different distributions (boolean cube, sphere, standard Gaussian) behave very similarly in high dimensions. However, the spectrum is much easier to compute over the boolean cube: while the sphere and Gaussian eigenvalues would require integration against a kind of polynomials known as the Gegenbauer polynomials, the boolean ones only require calculating a linear combination of a small number of terms. For this reason, in the rest of the paper we focus on analyzing the eigenvalues over the boolean cube.

Just as the usual Fourier basis over $\mathbb{R}$ has a notion of frequency that can be interpreted as a measure of complexity, so does the boolean Fourier basis (this is just the *degree*; see Section 3.1). While not perfect, we adopt this natural notion of complexity in this work; a "simple" function is then one that is well approximated by "low frequencies."

This spectral perspective immediately yields that the simplicity bias is not universal (Section 4). In particular, while it seems to hold more or less for relu networks, for sigmoidal networks, the simplicity bias can be made arbitrarily weak by changing the weight variance and the depth. In the extreme case, the random function obtained from sampling a deep erf network with large weights is distributed like a "white noise." However, there is a very weak sense in which the simplicity bias *does* hold: the eigenvalues of more "complex" eigenspaces cannot be bigger than those of less "complex" eigenspaces (Thm 4.1).

Next, we examine how hyperparameters affect the performance of neural networks through the lens of NTK and its spectrum. To do so, we first need to understand the simpler question of how a kernel affects the accuracy of the function learned by kernel regression. A coarse-grained theory, concerned with big-O asymptotics, exists from classical kernel literature (Yao et al., 2007; Raskutti et al., 2013; Wei et al.; Lin and Rosasco; Schölkopf and Smola, 2002). However, the fine-grained details, required for discerning the effect of hyperparameters, have been much less studied. We make a first attempt at a heuristic, *fractional variance* (i.e. what fraction of the trace of the kernel does an eigenspace contribute), for understanding how a minute change in kernel effects a change in performance. Intuitively, if an eigenspace has very large fractional variance, so that it accounts for most of the trace, then a ground truth function from this eigenspace should be very easy to learn.

Using this heuristic, we make two predictions about neural networks, motivated by observations in the spectra of NTK and CK, and verify them with extensive experiments.

- Deeper networks learn more complex features, but excess depth can be detrimental as well. Spectrally, depth can increase fractional variance of an eigenspace, but past an *optimal depth*, it will also decrease it. (Section 5) Thus, deeper is *not* always better.

- Training all layers is better than training just the last layer when it comes to more complex features, but the opposite is true for simpler features. Spectrally, fractional variances of more "complex" eigenspaces for the NTK are larger than the correponding quantities of the CK. (Section 6)

Finally, we use our spectral theory to predict the maximal nondiverging learning rate ("max learning rate") of SGD (Section 7).

In general, we will not only verify our theory with experiments on the theoretically interesting distributions, i.e. uniform measures over the boolean cube and the sphere, or the standard Gaussian, but also confirm these findings on real data like MNIST and CIFAR10 [1].

---

[1] The code for computing the eigenvalues and for reproducing the plots of this paper is available at `github.com/jxVmnLgedVwv6mNcGCBy/NNspectra`, which will be open sourced upon publication.

For space concerns, we review relevant literature along the flow of the main text, and relegate a more complete discussion of the related research landscape in Appendix A.

## 2 KERNELS ASSOCIATED TO NEURAL NETWORKS

As mentioned in the introduction, we now know several kernels associated to infinite width, randomly initialized neural networks. The most prominent of these are the *neural tangent kernel* (NTK) (Jacot et al., 2018) and the *conjugate kernel* (CK) (Daniely et al., 2016), which is also called the *NNGP kernel* (Lee et al., 2018). We briefly review them below. First we introduce the following notation that we will repeatedly use.

**Definition 2.1.** For $\phi : \mathbb{R} \to \mathbb{R}$, write $\mathrm{V}_\phi$ for the function that takes a PSD (positive semidefinite) kernel function to a PSD kernel of the same domain by the formula

$$\mathrm{V}_\phi(K)(x, x') = \mathbb{E}_{f \sim \mathcal{N}(0,K)} \phi(f(x))\phi(f(x')).$$

**Conjugate Kernel**    Neural networks are commonly thought of as learning a high-quality embedding of inputs to the latent space represented by the network's last hidden layer, and then using its final linear layer to read out a classification given the embedding. The conjugate kernel is just the kernel associated to the embedding induced by a random initialization of the neural network. Consider an MLP with widths $\{n^l\}_l$, weight matrices $\{W^l \in \mathbb{R}^{n^l \times n^{l-1}}\}_l$, and biases $\{b^l \in \mathbb{R}^{n^l}\}_l$, $l = 1, \ldots, L$. For simplicity of exposition, in this paper, we will only consider scalar output $n^L = 1$. Suppose it is parametrized by the *NTK parametrization*, i.e. its computation is given recursively as

$$h^1(x) = \frac{\sigma_w}{\sqrt{n^0}} W^1 x + \sigma_b b^1 \quad \text{and} \quad h^l(x) = \frac{\sigma_w}{\sqrt{n^{l-1}}} W^l \phi(h^{l-1}(x)) + \sigma_b b^l \qquad \text{(MLP)}$$

with some hyperparameters $\sigma_w, \sigma_b$ that are fixed throughout training[2]. At initialization time, suppose $W^l_{\alpha\beta}, b^l_\alpha \sim \mathcal{N}(0,1)$ for each $\alpha \in [n^l], \beta \in [n^{l-1}]$. It can be shown that, for each $\alpha \in [n^l]$, $h^l_\alpha$ is a Gaussian process with zero mean and kernel function $\Sigma^l$ in the limit as all hidden layers become infinitely wide ($n^l \to \infty, l = 1, \ldots, L - 1$), where $\Sigma^l$ is defined inductively on $l$ as

$$\Sigma^1(x, x') \overset{\text{def}}{=} \sigma_w^2 (n^0)^{-1} \langle x, x' \rangle + \sigma_b^2, \quad \Sigma^l \overset{\text{def}}{=} \sigma_w^2 \mathrm{V}_\phi(\Sigma^{l-1}) + \sigma_b^2 \qquad \text{(CK)}$$

The kernel $\Sigma^L$ corresponding the the last layer $L$ is the network's *conjugate kernel*, and the associated Gaussian process limit is the reason for its alternative name *Neural Network-Gaussian process kernel*. In short, if we were to train a linear model with features given by the embedding $x \mapsto h^{L-1}(x)$ when the network parameters are randomly sampled as above, then the CK is the kernel of this linear model. See Daniely et al. (2016); Lee et al. (2018) and Appendix F for more details.

**Neural Tangent Kernel**    On the other hand, the NTK corresponds to training the entire model instead of just the last layer. Intuitively, if we let $\theta$ be the entire set of parameters $\{W^l\}_l \cup \{b^l\}_l$ of Eq. (MLP), then for $\theta$ close to its initialized value $\theta_0$, we expect

$$h^L(x; \theta) - h^L(x; \theta_0) \approx \langle \nabla_\theta h^L(x; \theta_0), \theta - \theta_0 \rangle$$

via a naive first-order Taylor expansion. In other words, $h^L(x; \theta) - h^L(x; \theta_0)$ behaves like a linear model with feature of $x$ given by the gradient taken w.r.t. the initial network, $\nabla_\theta h^L(x; \theta_0)$, and the weights of this *linear model* are the deviation $\theta - \theta_0$ of $\theta$ from its initial value. It turns out that, in the limit as all hidden layer widths tend to infinity, this intuition is correct (Jacot et al., 2018; Lee et al., 2018; Yang, 2019), and the following inductive formula computes the corresponding infinite-width kernel of this linear model:

$$\Theta^1 \overset{\text{def}}{=} \Sigma^1, \quad \Theta^l(x, x') \overset{\text{def}}{=} \Sigma^l(x, x') + \sigma_w^2 \Theta^{l-1}(x, x') \mathrm{V}_{\phi'}(\Sigma^{l-1})(x, x'). \qquad \text{(NTK)}$$

**Computing CK and NTK**    While in general, computing $\mathrm{V}_\phi$ and $\mathrm{V}_{\phi'}$ requires evaluating a multivariate Gaussian expectation, in specific cases, such as when $\phi = \mathrm{relu}$ or $\mathrm{erf}$, there exists explicit, efficient formulas that only require pointwise evaluation of some simple functions (see Facts F.1 and F.2). This allows us to evaluate CK and NTK on a set $\mathcal{X}$ of inputs in only time $O(|\mathcal{X}|^2 L)$.

---

[2]SGD with learning rate $\alpha$ in this parametrization is roughly equivalent to SGD with learning rate $\alpha/width$ in the standard parametrization with Glorot initialization; see Lee et al. (2018)

**What Do the Spectra of CK and NTK Tell Us?** In summary, the CK governs the distribution of a randomly initialized neural network and also the properties of training only the last layer of a network, while the NTK governs the dynamics of training (all parameters of) a neural network. A study of their spectra thus informs us of the "implicit prior" of a randomly initialized neural network as well as the "implicit bias" of GD in the context of training neural networks.

In regards to the implicit prior at initialization, we know from Lee et al. (2018) that a randomly initialized network as in Eq. (MLP) is distributed as a Gaussian process $\mathcal{N}(0, K)$, where $K$ is the corresponding CK, in the infinite-width limit. If we have the eigendecomposition

$$K = \sum_{i \geq 1} \lambda_i u_i \otimes u_i \tag{1}$$

with eigenvalues $\lambda_i$ in decreasing order and corresponding eigenfunctions $u_i$, then each sample from this GP can be obtained as

$$\sum_{i \geq 1} \sqrt{\lambda_i} \omega_i u_i, \quad \omega_i \sim \mathcal{N}(0, 1).$$

If, for example, $\lambda_1 \gg \sum_{i \geq 2} \lambda_i$, then a typical sample function is just a very small perturbation of $u_1$. We will see that for relu, this is indeed the case (Section 4), and this explains the "simplicity bias" in relu networks found by Valle-Pérez et al. (2018).

Training the last layer of a randomly initialized network via full batch gradient descent for an infinite amount of time corresponds to Gaussian process inference with kernel $K$ (Lee et al., 2018; 2019). A similar intuition holds for NTK: training all parameters of the network (Eq. (MLP)) for an infinite amount of time yields the mean prediction of the GP $\mathcal{N}(0, \text{NTK})$ in expectation; see Lee et al. (2019) and Appendix F.4 for more discussion.

Thus, the more the GP prior (governed by the CK or the NTK) is consistent with the ground truth function $f^*$, the more we expect the Gaussian process inference and GD training to generalize well. We can measure this consistency in the "alignment" between the eigenvalues $\lambda_i$ and the squared coefficients $a_i^2$ of $f^*$'s expansion in the $\{u_i\}_i$ basis. The former can be interpreted as the expected magnitude (squared) of the $u_i$-component of a sample $f \sim \mathcal{N}(0, K)$, and the latter can be interpreted as the actual magnitude squared of such component of $f^*$. In this paper, we will investigate an even cleaner setting where $f^* = u_i$ is an eigenfunction. Thus we would hope to use a kernel whose $i$th eigenvalue $\lambda_i$ is as large as possible.

**Neural Kernels** From the forms of the equation Eqs. (CK) and (NTK) and the fact that $V_\phi(K)(x, x')$ only depends on $K(x, x)$, $K(x, x')$, and $K(x', x')$, we see that CK or NTK of MLPs takes the form

$$K(x, y) = \Phi\left(\frac{\langle x, y \rangle}{\|x\| \|y\|}, \frac{\|x\|^2}{d}, \frac{\|y\|^2}{d}\right) \tag{2}$$

for some function $\Phi : \mathbb{R}^3 \to \mathbb{R}$. We will refer to this kind of kernel as *Neural Kernel* in this paper.

**Kernels as Integral Operators** We will consider input spaces of various forms $\mathcal{X} \subseteq \mathbb{R}^d$ equipped with some probability measure. Then a kernel function $K$ acts as an *integral operator* on functions $f \in L^2(\mathcal{X})$ by

$$Kf(x) = (Kf)(x) = \mathop{\mathbb{E}}_{y \sim \mathcal{X}} K(x, y) f(y).$$

We will use the "juxtaposition syntax" $Kf$ to denote this application of the integral operator. [3] Under certain assumptions, it then makes sense to speak of the eigenvalues and eigenfunctions of the integral operator $K$. While we will appeal to an intuitive understanding of eigenvalues and eigenfunctions in the main text below, we include a more formal discussion of Hilbert-Schmidt operators and their spectral theory in Appendix G for completeness. In the next section, we investigate the eigendecomposition of neural kernels as integral operators over different distributions.

---

[3] In cases when $\mathcal{X}$ is finite, $K$ can be also thought of as a big matrix and $f$ as a vector — but do not confuse $Kf$ with their multiplication! If we use $\cdot$ to denote matrix multiplication, then the operator application $Kf$ is the same as the matrix multiplication $K \cdot D \cdot f$ where $D$ is the diagonal matrix encoding the probability values of each point in $\mathcal{X}$.

## 3 THE SPECTRA OF NEURAL KERNELS

### 3.1 BOOLEAN CUBE

We first consider a neural kernel $K$ on the boolean cube $\mathcal{X} = \boxdot^d \overset{\text{def}}{=} \{\pm 1\}^d$, equipped with the uniform measure. In this case, since each $x \in \mathcal{X}$ has the same norm, $K(x, y) = \Phi\left(\frac{\langle x,y \rangle}{\|x\|\|y\|}, \frac{\|x\|^2}{d}, \frac{\|y\|^2}{d}\right)$ effectively only depends on $\langle x, y \rangle$, so we will treat $\Phi$ as a single variate function in this section, $\Phi(c) = \Phi(c, 1, 1)$.

**Brief review of basic Fourier analysis on the boolean cube $\boxdot^d$ (O'Donnell (2014)).** The space of real functions on $\boxdot^d$ forms a $2^d$-dimensional space. Any such function has a *unique* expansion into a *multilinear polynomial* (polynomials whose monomials do not contain $x_i^p, p \geq 2$, of any variable $x_i$). For example, the majority function over 3 bits has the following unique multilinear expansion

$$\text{maj}_3 : \boxdot^3 \to \boxdot^1, \quad \text{maj}_3(x_1, x_2, x_3) = \frac{1}{2}(x_1 + x_2 + x_3 - x_1 x_2 x_3).$$

In the language of Fourier analysis, the $2^d$ multilinear monomial functions

$$\chi_S(x) \overset{\text{def}}{=} x^S \overset{\text{def}}{=} \prod_{i \in S} x_i, \quad \text{for each } S \subseteq [d] \tag{3}$$

form a Fourier basis of the function space $L^2(\boxdot^d) = \{f : \boxdot^d \to \mathbb{R}\}$, in the sense that their inner products satisfy

$$\mathop{\mathbb{E}}_{x \sim \boxdot^d} \chi_S(x)\chi_T(x) = \mathbb{I}(S = T).$$

Thus, any function $f : \boxdot^d \to \mathbb{R}$ can be always written as

$$f(x) = \sum_{S \subseteq [d]} \hat{f}(S)\chi_X(x)$$

for a *unique* set of coefficients $\{\hat{f}(S)\}_{S \subseteq [d]}$.

It turns out that $K$ is always diagonalized by this Fourier basis $\{\chi_S\}_{S \subseteq [d]}$.

**Theorem 3.1.** *On the $d$-dimensional boolean cube $\boxdot^d$, for every $S \subseteq [d]$, $\chi_S$ is an eigenfunction of $K$ with eigenvalue*

$$\mu_{|S|} \overset{\text{def}}{=} \mathop{\mathbb{E}}_{x \in \boxdot^d} x^S K(x, \mathbb{1}) = \mathop{\mathbb{E}}_{x \in \boxdot^d} x^S \Phi\left(\sum_i x_i/d\right), \tag{4}$$

*where $\mathbb{1} = (1, \ldots, 1) \in \boxdot^d$. This definition of $\mu_{|S|}$ does not depend on the choice $S$, only on the cardinality of $S$. These are all of the eigenfunctions of $K$ by dimensionality considerations.*[4]

Define $\mathcal{T}_\Delta$ to be the shift operator on functions over $[-1, 1]$ that sends $\Phi(\cdot)$ to $\Phi(\cdot - \Delta)$. Then we can re-express the eigenvalue as follows.

**Lemma 3.2.** *With $\mu_k$ as in Thm 3.1,*

$$\mu_k = 2^{-d}(I - \mathcal{T}_\Delta)^k (I + \mathcal{T}_\Delta)^{d-k} \Phi(1) \tag{5}$$

$$= 2^{-d} \sum_{r=0}^d C_r^{d-k,k} \Phi\left(\left(\frac{d}{2} - r\right)\Delta\right) \tag{6}$$

*where*

$$C_r^{d-k,k} \overset{\text{def}}{=} \sum_{j=0}^r (-1)^{r+j}\binom{d-k}{j}\binom{k}{r-j}. \tag{7}$$

Eq. (5) will be important for computational purposes, and we will come back to discuss this more in Section 3.5. It also turns out $\mu_k$ affords a pretty expression via the Fourier series coefficients of $\Phi$. As this is not essential to the main text, we relegate its exposition to Appendix H.1.

---

[4]Readers familiar with boolean Fourier analysis may be reminded of the *noise operator* $T_\rho$, $\rho \leq 1$ (O'Donnell, 2014, Defn 2.46). In the language of this work, $T_\rho$ is a neural kernel with eigenvalues $\mu_k = \rho^k$.

## 3.2 Sphere

Now let's consider the case when $\mathcal{X} = \sqrt{d}\mathcal{S}^{d-1}$ is the radius-$\sqrt{d}$ sphere in $\mathbb{R}^d$ equipped with the uniform measure. Again, because $x \in \mathcal{X}$ all have the same norm, we will treat $\Phi$ as a univariate function with $K(x, y) = \Phi(\langle x, y \rangle / \|x\|\|y\|) = \Phi(\langle x, y \rangle / d)$. As is long known (Schoenberg, 1942; Gneiting, 2013; Xu and Cheney, 1992; Smola et al., 2001), $K$ is diagonalized by spherical harmonics, and the eigenvalues are given by the coefficients of $\Phi$ against a system of orthogonal polynomials called Gegenbuaer polynomials. We relegate a complete review of this topic to Appendix H.2.

## 3.3 Isotropic Gaussian

Now let's consider $\mathcal{X} = \mathbb{R}^d$ equipped with standard isotropic Gaussian $\mathcal{N}(0, I)$, so that $K$ behaves like

$$Kf(x) = \mathbb{E}_{y \sim \mathcal{N}(0,I)} K(x, y)f(y) = \mathbb{E}_{y \sim \mathcal{N}(0,I)} \Phi\left(\frac{\langle x, y \rangle}{\|x\|\|y\|}, \frac{\|x\|^2}{d}, \frac{\|y\|^2}{d}\right) f(y)$$

for any $f \in L^2(\mathcal{N}(0, I))$. In contrast to the previous two sections, $K$ will essentially depend on the effect of the norms $\|x\|$ and $\|y\|$ on $\Phi$.

Nevertheless, because an isotropic Gaussian vector can be obtained by sampling its direction uniformly from the sphere and its magnitude from a chi distribution, $K$ can still be partially diagonalized into a sum of products between spherical harmonics and kernels on $\mathbb{R}$ equipped with a chi distribution (Thm H.14). In certain cases, we can obtain complete eigendecompositions, for example when $\Phi$ is positive homogeneous. See Appendix H.3 for more details.

## 3.4 Kernel is Same over Boolean Cube, Sphere, or Gaussian when $d \gg 1$

The reason we have curtailed a detailed discussion of neural kernels on the sphere and on the standard Gaussian is because, in high dimension, the kernel behaves the same under these distributions as under uniform distribution over the boolean cube. Indeed, by intuition along the lines of the central limit theorem, we expect that uniform distribution over a high dimension boolean cube should approximate high dimensional standard Gaussian. Similarly, by concentration of measure, most of the mass of a Gaussian is concentrated around a thin shell of radius $\sqrt{d}$. Thus, morally, we expect the same kernel function $K$ induces approximately the same integral operator on these three distributions in high dimension, and as such, their eigenvalues *should* also approximately coincide. We verify empirically and theoretically this is indeed the case in Appendix H.4.

## 3.5 Computing the Eigenvalues

As the eigenvalues of $K$ over the different distributions are very close, we will focus in the rest of this paper on eigenvalues over the boolean cube. This has the additional benefit of being much easier to compute.

Each eigenvalue over the sphere and the standard Gaussian requires an integration of $\Phi$ against a Gegenbauer polynomial. In high dimension $d$, these Gegenbauer polynomials varies wildly in a sinusoidal fashion, and blows up toward the boundary (see Fig. 15 in the Appendix). As such, it is difficult to obtain a numerically stable estimate of this integral in an efficient manner when $d$ is large.

In contrast, we have multiple ways of computing boolean cube eigenvalues, via Eqs. (5) and (6). In either case, we just take some linear combination of the values of $\Phi$ at a grid of points on $[-1, 1]$, spaced apart by $\Delta = 2/d$. While the coefficients $C_r^{d-k,k}$ (defined in Eq. (7)) are relatively efficient to compute, the change in the sign of $C_r^{d-k,k}$ makes this procedure numerically unstable for large $d$. Instead, we use Eq. (5) to isolate the alternating part to evaluate in a numerically stable way: Since $\mu_k = \left(\frac{I + \mathcal{T}_\Delta}{2}\right)^{d-k} \left(\frac{I - \mathcal{T}_\Delta}{2}\right)^k \Phi(1)$, we can evaluate $\tilde{\Phi} \stackrel{\text{def}}{=} \left(\frac{I - \mathcal{T}_\Delta}{2}\right)^k \Phi$ via $k$ finite differences, and then compute

$$\left(\frac{I + \mathcal{T}_\Delta}{2}\right)^{d-k} \tilde{\Phi}(1) = \frac{1}{2^{d-k}} \sum_{r=0}^{d-k} \binom{d-k}{r} \tilde{\Phi}(1 - r\Delta). \tag{8}$$

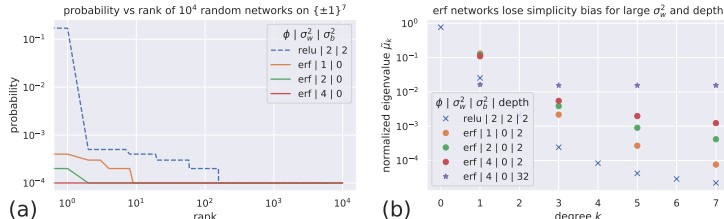

Figure 1: **The "simplicity bias" is not so simple. (a)** Following Valle-Pérez et al. (2018), we sample $10^4$ boolean functions $\{\pm 1\}^7 \rightarrow \{\pm 1\}$ as follows: for each combination of nonlinearity, weight variance $\sigma_w^2$, and bias variance $\sigma_b^2$ (as used in Eq. (MLP)), we randomly initialize a network of 2 hidden layers, 40 neurons each. Then we threshold the function output to a boolean output, and obtain a boolean function sample. We repeat this for $10^4$ random seeds to obtain all samples. Then we sort the samples according to their empirical probability (this is the x-axis, *rank*), and plot their empirical probability (this is the y-axis, *probability*). The high values at the left of the relu curve indicates that a few functions get sampled repeatedly, while this is less true for erf. For erf and $\sigma_w^2 = 4$, no function got sampled more than once. **(b)** For different combination of nonlinearity, $\sigma_w^2$, $\sigma_b^2$, and depth, we study the eigenvalues of the corresponding CK. Each CK has 8 different eigenvalues $\mu_0, \ldots, \mu_7$ corresponding to homogeneous polynomials of degree $0, \ldots, 7$. We plot them in log scale against the degree. Note that for erf and $\sigma_b = 0$, the even degree $\mu_k$ vanish. See main text for explanations.

When $\Phi$ arises from the CK or the NTK of an MLP, all derivatives of $\Phi$ at 0 are nonnegative (Thm I.3). Thus intuitively, the finite difference $\tilde{\Phi}$ should be also all nonnegative, and this sum can be evaluated without worry about floating point errors from cancellation of large terms.

A slightly more clever way to improve the numerical stability when $2k \leq d$ is to note that

$$(I + \mathcal{T}_\Delta)^{d-k} (I - \mathcal{T}_\Delta)^k \Phi(1) = (I + \mathcal{T}_\Delta)^{d-2k} (I - \mathcal{T}_\Delta^2)^k \Phi(1) = (I + \mathcal{T}_\Delta)^{d-2k} (I - \mathcal{T}_{2\Delta})^k \Phi(1).$$

So an improved algorithm is to first compute the $k$th finite difference $(I - \mathcal{T}_{2\Delta})^k$ with the larger step size $2\Delta$, then compute the sum $(I + \mathcal{T}_\Delta)^{d-2k}$ as in Eq. (8).

## 4 Clarifying the "Simplicity Bias" of Random Neural Networks

As mentioned in the introduction, Valle-Pérez et al. (2018) claims that neural networks are *biased toward simple functions*. We show that this phenomenon depends crucially on the nonlinearity, the sampling variances, and the depth of the network. In Fig. 1(a), we have repeated their experiment for $10^4$ random functions obtained by sampling relu neural networks with 2 hidden layers, 40 neurons each, following Valle-Pérez et al. (2018)'s architectural choices[5]. We also do the same for erf networks of the same depth and width, varying as well the sampling variances of the weights and biases, as shown in the legend. As discussed in Valle-Pérez et al. (2018), for relu, there is indeed this bias, where a single function gets sampled more than 10% of the time. However, for erf, as we increase $\sigma_w^2$, we see this bias disappear, and every function in the sample gets sampled only once.

This phenomenon can be explained by looking at the eigendecomposition of the CK, which is the Gaussian process kernel of the distribution of the random networks as their hidden widths tend to infinity. In Fig. 1(b), we plot the normalized eigenvalues $\{\mu_k / \sum_{i=0}^{7} \binom{7}{i} \mu_i\}_{k=0}^{7}$ for the CKs corresponding to the networks sampled in Fig. 1(a). Immediately, we see that for relu and $\sigma_w^2 = \sigma_b^2 = 2$, the degree 0 eigenspace, corresponding to constant functions, accounts for more than 80% of the variance. This means that a typical infinite-width relu network of 2 layers is expected to be almost constant, and this should be even more true after we threshold the network to be a boolean function. On the other hand, for erf and $\sigma_b = 0$, the even degree $\mu_k$s all vanish, and most of the variance comes from degree 1 components (i.e. linear functions). This concentration in degree 1 also lessens as $\sigma_w^2$ increases. But because this variance is spread across a dimension 7 eigenspace, we don't see duplicate function samples nearly as much as in the relu case. As $\sigma_w$ increases, we also see the eigenvalues become more equally distributed, which corresponds to the flattening of

---

[5]Valle-Pérez et al. (2018) actually performed their experiments over the $\{0, 1\}^7$ cube, not the $\{\pm 1\}^7$ cube we are using here. This does not affect our conclusion. See Appendix J for more discussion

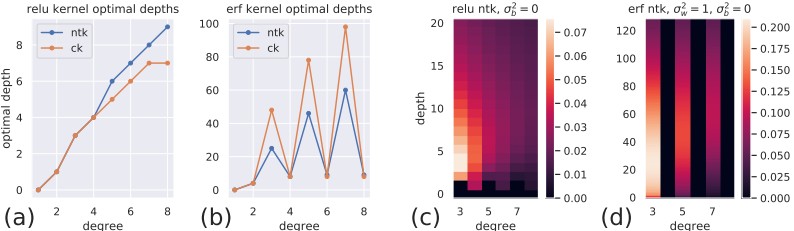

Figure 2: **The depth maximizing degree $k$ fractional variance increases with $k$ for both relu and erf.** For relu **(a)** and erf **(b)**, we plot for each degree $k$ the depth such that there exists some combination of other hyperparameters (such as $\sigma_b^2$ or $\sigma_w^2$) that maximizes the degree $k$ fractional variance. For both relu, $\sigma_b^2 = 0$ maximizes fractional variance in general, and same holds for erf in the odd degrees (see Appendix D), so we take a closer look at this slice by plotting heatmaps of fractional variance of various degrees versus depth for relu **(c)** and erf **(d)** NTK, with bright colors representing high variance. Clearly, we see the brightest region of each column, corresponding to a fixed degree, moves up as we increase the degree, barring for the even/odd degree alternating pattern for erf NTK. The pattern for CKs are similar and their plots are omitted.

the probability-vs-rank curve in Fig. 1(a). Finally, we observe that a 32-layer erf network with $\sigma_w^2 = 4$ has all its nonzero eigenvalues (associated to odd degrees) all equal (see points marked by $*$ in Fig. 1(b)). This means that its distribution is a "white noise" on the space of *odd* functions, and the distribution of boolean functions obtained by thresholding the Gaussian process samples is the *uniform distribution* on *odd* functions. This is the complete lack of simplicity bias modulo the oddness constraint.

However, from the spectral perspective, there is a weak sense in which a simplicity bias holds for all neural network-induced CKs and NTKs.

**Theorem 4.1** (Weak Spectral Simplicity Bias)**.** *Let $K$ be the CK or NTK of an MLP on a boolean cube $\boxed{\pm}^d$. Then the eigenvalues $\mu_k, k = 0, \ldots, d$, satisfy*

$$\mu_0 \geq \mu_2 \geq \cdots \geq \mu_{2k} \geq \cdots, \quad \mu_1 \geq \mu_3 \geq \cdots \geq \mu_{2k+1} \geq \cdots. \tag{9}$$

Even though it's not true that the fraction of variance contributed by the degree $k$ eigenspace is decreasing with $k$, the eigenvalue themselves will be in a nonincreasing pattern across even and odd degrees. In fact, if we fix $k$ and let $d \to \infty$, then we can show that (Thm I.6)

$$\mu_k = \Theta(d^{-k}).$$

Of course, as we have seen, this is a *very weak* sense of simplicity bias, as it doesn't prevent "white noise" behavior as in the case of erf CK with large $\sigma_w^2$ and large depth.

## 5 DEEPER NETWORKS LEARN MORE COMPLEX FEATURES

In the rest of this work, we compute the eigenvalues $\mu_k$ over the 128-dimensional boolean cube ($\boxed{\pm}^d$, with $d = 128$) for a large number of different hyperparameters, and analyze how the latter affect the former. We vary the degree $k \in [0, 8]$, the nonlinearity between relu and erf, the depth (number of hidden layers) from 1 to 128, and $\sigma_b^2 \in [0, 4]$. We fix $\sigma_w^2 = 2$ for relu kernels, but additionally vary $\sigma_w^2 \in [1, 5]$ for erf kernels. Comprehensive contour plots of how these hyperparameters affect the kernels are included in Appendix D, but in the main text we summarize several trends we see.

We will primarily measure the change in the spectrum by the *degree $k$ fractional variance*, which is just

$$\text{degree } k \text{ fractional variance} \stackrel{\text{def}}{=} \frac{\binom{d}{k}\mu_k}{\sum_{i=0}^{d}\binom{d}{i}\mu_i}.$$

This terminology comes from the fact that, if we were to sample a function $f$ from a Gaussian process with kernel $K$, then we expect that $r\%$ of the total variance of $f$ comes from degree $k$ components of $f$, where $r\%$ is the degree $k$ fractional variance. If we were to try to learn a homogeneous degree-$k$

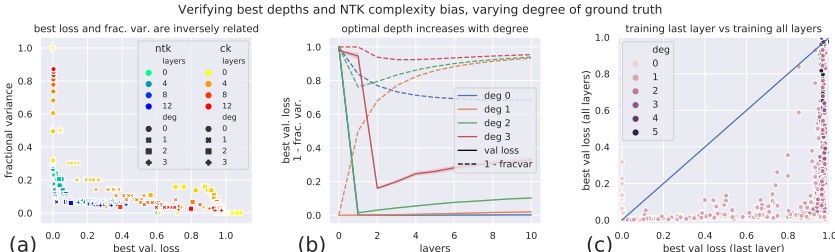

Figure 3: **(a)** We train relu networks of different depths against a ground truth polynomial on $\mathbb{O}^{128}$ of different degrees $k$. We either train only the last layer (marked "ck") or all layers (marked "ntk"), and plot the degree $k$ fractional variance of the corresponding kernel against the best validation loss over the course of training. We see that the best validation loss is in general inversely correlated with fraction variance, as expected. However, their precise relationship seems to change depending on the degree, or whether training all layers or just the last. See Appendix E for experimental details. **(b)** Same experimental setting as (a), with slightly different hyperparameters, and plotting depth against best validation loss (solid curves), as well as the corresponding kernel's ($1-$ fractional variance) (dashed curves). We see that the loss-minimizing depth increases with the degree, as predicted by Fig. 2. Note that we do not expect the dashed and solid curves to match up, just that they are positively correlated as shown by (a). In higher degrees, the losses are high across all depths, and the variance is large, so we omit them. See Appendix E for experimental details. **(c)** Similar experimental setting as (a), but with more hyperparameters, and now comparing training last layer vs training all layers. The color of each dot indicates the degree of the ground truth polynomial. Below the identity line, training all layers is better than training last layer. We see that the only nontrivial case where this is not true is when learning degree 0 polynomials, i.e. constant functions. See Appendix E for experimental details. We also replicate (b) for MNIST and CIFAR10, and moreover both (b) and (c) over the input distributions of standard Gaussian and the uniform measure over the sphere. See Figs. 6 to 8.

polynomial using a kernel $K$, intuitively we should try to choose $K$ such that its $\mu_k$ is maximized, relative to other eigenvalues. Fig. 3(a) shows that this is indeed the case even with neural networks: over a large number of different hyperparameter settings, degree $k$ fractional variance is inversely related to the validation loss incurred when learning a degree $k$ polynomial. However, this plot also shows that there does not seem like a precise, clean relationship between fractional variance and validation loss. Obtaining a better measure for predicting generalization is left for future work.

Before we continue, we remark that the fractional variance of a fixed degree $k$ converges to a fixed value as the input dimension $d \to \infty$:

**Theorem 5.1** (Asymptotic Fractional Variance). *Let $K$ be the CK or NTK of an MLP on a boolean cube $\mathbb{O}^d$. Then $K$ can be expressed as $K(x, y) = \Phi(\langle x, y \rangle / d)$ for some analytic function $\Phi : \mathbb{R} \to \mathbb{R}$. If we fix $k$ and let the input dimension $d \to \infty$, then the fractional variance of degree $k$ converges to*

$$(k!)^{-1}\Phi^{(k)}(0)/\Phi(1) = \frac{(k!)^{-1}\Phi^{(k)}(0)}{\sum_{j \geq 0}(j!)^{-1}\Phi^{(j)}(0)}$$

*where $\Phi^{(k)}$ denotes the kth derivative of $\Phi$.*

For the fractional variances we compute in this paper, their values at $d = 128$ are already very close to their $d \to \infty$ limit, so we focus on the $d = 128$ case experimentally.

If $K$ were to be the CK or NTK of a relu or erf MLP, then we find that for higher $k$, the depth of the network helps increase the degree $k$ fractional variance. In Fig. 2(a) and (b), we plot, for each degree $k$, the depth that (with some combination of other hyperparameters like $\sigma_b^2$) achieves this maximum, for respectively relu and erf kernels. Clearly, the maximizing depths are increasing with $k$ for relu, and also for erf when considering either odd $k$ or even $k$ only. The slightly differing behavior between even and odd $k$ is expected, as seen in the form of Thm 4.1. Note the different scales of y-axes for relu and erf — the depth effect is much stronger for erf than relu.

For relu NTK and CK, $\sigma_b^2 = 0$ maximizes fractional variance in general, and the same holds for erf NTK and CK in the odd degrees (see Appendix D). In Fig. 2(c) and Fig. 2(d) we give a more fine-grained look at the $\sigma_b^2 = 0$ slice, via heatmaps of fractional variance against degree and depth.

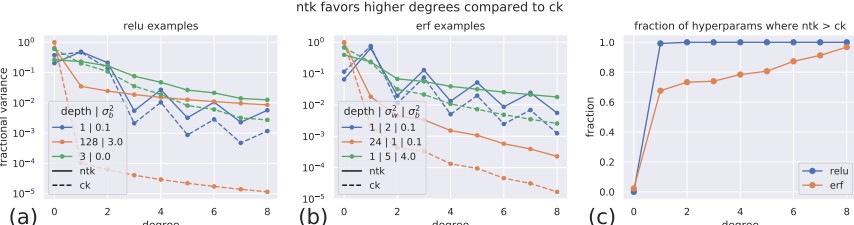

Figure 4: **Across nonlinearities and hyperparameters, NTK tends to have higher fraction of variance attributed to higher degrees than CK.** In **(a)**, we give several examples of the fractional variance curves for relu CK and NTK across several representative hyperparameters. In **(b)**, we do the same for erf CK and NTK. In both cases, we clearly see that, while for degree 0 or 1, the fractional variance is typically higher for CK, the reverse is true for larger degrees. In **(c)**, for each degree $k$, we plot the *fraction of hyperparameters* where the degree $k$ fractional variance of NTK is greater than that of CK. Consistent with previous observations, this fraction increases with the degree.

Brighter color indicates higher variance, and we see the optimal depth for each degree $k$ clearly increases with $k$ for relu NTK, and likewise for odd degrees of erf NTK. However, note that as $k$ increases, the difference between the maximal fractional variance and those slightly suboptimal becomes smaller and smaller, reflected by suppressed range of color moving to the right. The heatmaps for relu and erf CKs look similar and are omitted.

We verify this increase of optimal depth with degree in Fig. 3(b). There we have trained relu networks of varying depth against a ground truth multilinear polynomial of varying degree. We see clearly that the optimal depth is increasing with degree. We also verify this phenomenon when the input distribution changes to the standard Gaussian or the uniform distribution over the sphere $\sqrt{d}\mathcal{S}^{d-1}$; see Fig. 6.

Note that implicit in our results here is a highly nontrivial observation: Past some point (the *optimal depth*), high depth can be detrimental to the performance of the network, beyond just the difficulty to train, and this detriment can already be seen in the corresponding NTK or CK. In particular, it's *not* true that the optimal depth is infinite. We confirm the existence of such an optimal depth even in real distributions like MNIST and CIFAR10; see Fig. 7. This adds significant nuance to the folk wisdom that "depth increases expressivity and allows neural networks to learn more complex features."

## 6 NTK FAVORS MORE COMPLEX FEATURES THAN CK

We generally find the degree $k$ fractional variance of NTK to be higher than that of CK when $k$ is large, and vice versa when $k$ is small, as shown in Fig. 4. This means that, if we train only the last layer of a neural network (i.e. CK dynamics), we intuitively should expect to learn simpler features better, while, if we train all parameters of the network (i.e. NTK dynamics), we should expect to learn more complex features better. Similarly, if we were to sample a function from a Gaussian process with the CK as kernel (recall this is just the distribution of randomly initialized infinite width MLPs (Lee et al., 2018)), this function is more likely to be accurately approximated by low degree polynomials than the same with the NTK.

We verify this intuition by training a large number of neural networks against ground truth functions of various homogeneous polynomials of different degrees, and show a scatterplot of how training the last layer only measures against training all layers (Fig. 3(c)). This phenomenon remains true over the standard Gaussian or the uniform distribution on the sphere (Fig. 8). Consistent with our theory, the only place training the last layer works meaningfully better than training all layers is when the ground truth is a constant function. However, we reiterate that fractional variance is an imperfect indicator of performance. Even though for erf neural networks and $k \geq 1$, degree $k$ fractional variance of NTK is not always greater than that of the CK, we do not see any instance where training the last layer of an erf network is better than training all layers. We leave an investigation of this discrepancy to future work.

## 7 PREDICTING THE MAXIMUM LEARNING RATE

In any setup that tries to push deep learning benchmarks, learning rate tuning is a painful but indispensable part. In this section, we show that our spectral theory can accurately predict the maximal nondiverging learning rate over *real datasets* as well as toy input distributions, which would help set the correct upper limit for a learning rate search.

By Jacot et al. (2018), in the limit of large width and infinite data, the function $g : \mathcal{X} \to \mathbb{R}$ represented by our neural network evolves like

$$g^{t+1} = g^t - 2\alpha K(g^t - g^*), \quad t = 0, 1, 2, \dots, \tag{10}$$

when trained under full batch GD (with the entire population) with L2 loss $L(f, g) = \mathbb{E}_{x \sim \mathcal{X}}(f(x) - g(x))^2$, ground truth $g^*$, and learning rate $\alpha$, starting from randomly initialization. If we train only the last layer, then $K$ is the CK; if we train all layers, then $K$ is the NTK. Given an eigendecomposition of $K$ as in Eq. (1), if $g^0 - g^* = \sum_i a_i u_i$ is the decomposition of $g^0$ in the eigenbasis $\{u_i\}_i$, then one can easily deduce that

$$g^t - g^* = \sum_i a_i (1 - 2\alpha\lambda_i)^t u_i.$$

Consequently, we must have $\alpha < (\max_i \lambda_i)^{-1}$ in order for Eq. (10) to converge [6]

When the input distribution is the uniform distribution over $\mathbb{Z}^d$, the maximum learning rate is $\max(\mu_0, \mu_1)$ by Thm 4.1. By Thm 5.1, as long as the $\Phi$ function corresonding to $K$ has $\Phi(0) \neq 0$, when $d$ is large, we expect $\mu_0 \approx \Phi(0)$ but $\mu_1 \sim d^{-1}\Phi'(0) \ll \mu_0$. Therefore, we should predict $\frac{1}{\Phi(0)}$ for the maximal learning rate when training on the boolean cube. However, as Fig. 5 shows, this prediction is accurate not only for the boolean cube, but also over the sphere, the standard Gaussian, and even MNIST and CIFAR10!

## 8 CONCLUSION

In this work, we have taken a first step at studying how hyperparameters change the initial distribution and the generalization properties of neural networks through the lens of neural kernels and their spectra. We obtained interesting insights by computing kernel eigenvalues over the boolean cube and relating them to generalization through the *fractional variance* heuristic. While it inspired valid predictions that are backed up by experiments, fractional variance is clearly just a rough indicator. We hope future work can refine on this idea to produce a much more precise prediction of test loss. Nevertheless, we believe the spectral perspective is the right line of research that will not only shed light on mysteries in deep learning but also inform design choices in practice.

## REFERENCES

Joshua Achiam, Ethan Knight, and Pieter Abbeel. Towards Characterizing Divergence in Deep Q-Learning. *arXiv:1903.08894 [cs]*, March 2019.

Zeyuan Allen-Zhu, Yuanzhi Li, and Yingyu Liang. Learning and Generalization in Overparameterized Neural Networks, Going Beyond Two Layers. *arXiv:1811.04918 [cs, math, stat]*, November 2018a.

Zeyuan Allen-Zhu, Yuanzhi Li, and Zhao Song. On the Convergence Rate of Training Recurrent Neural Networks. *arXiv:1810.12065 [cs, math, stat]*, October 2018b.

Zeyuan Allen-Zhu, Yuanzhi Li, and Zhao Song. A Convergence Theory for Deep Learning via Over-Parameterization. *arXiv:1811.03962 [cs, math, stat]*, November 2018c.

Sanjeev Arora, Simon S. Du, Wei Hu, Zhiyuan Li, Ruslan Salakhutdinov, and Ruosong Wang. On Exact Computation with an Infinitely Wide Neural Net. *arXiv:1904.11955 [cs, stat]*, April 2019a.

Sanjeev Arora, Simon S. Du, Wei Hu, Zhiyuan Li, and Ruosong Wang. Fine-Grained Analysis of Optimization and Generalization for Overparameterized Two-Layer Neural Networks. January 2019b.

---

[6]Note that this is the max learning rate of the infinite-width neural network evolving in the NTK regime, but not necessarily the max learning rate of the finite-wdith neural network, as a larger learning rate just means that the network no longer evolves in the NTK regime.

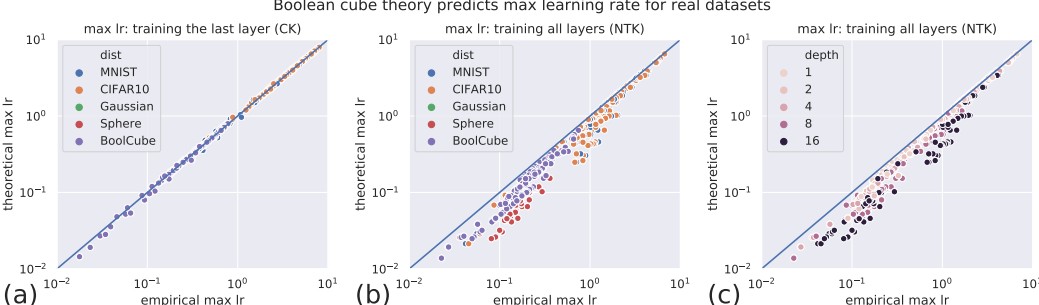

Figure 5: **Spectral theory of CK and NTK over boolean cube predicts max learning rate for SGD over real datasets MNIST and CIFAR10 as well as over boolean cube** $\boxdot^{128}$**, the sphere** $\sqrt{128}\mathcal{S}^{128-1}$**, and the standard Gaussian** $\mathcal{N}(0, I_{128})$**.** In all three plots, for different depth, nonlinearity, $\sigma_w^2, \sigma_b^2$ of the MLP, we obtain its maximal nondiverging learning rate ("max learning rate") via binary search. We center and normalize each image of MNIST and CIFAR10 to the $\sqrt{d}\mathcal{S}^{d-1}$ sphere, where $d = 28^2 = 784$ for MNIST and $d = 3 \times 32^2 = 3072$ for CIFAR10. See Appendix E.2 for more details. **(a)** We empirically find max learning rate for training only the last layer of an MLP. Theoretically, we predict $1/\Phi(0)$ where $\Phi$ corresponds to the *CK* of the MLP. We see that our theoretical prediction is highly accurate. Note that the Gaussian and Sphere points in the scatter plot coincide with and hide behind the BoolCube points. **(b) and (c)** We empirically find max learning rate for training all layers. Theoretically, we predict $1/\Phi(0)$ where $\Phi$ corresponds to the *NTK* of the MLP. The points are identical between (b) and (c), but the color coding is different. Note that the Gaussian points in the scatter plots coincide with and hide behind the Sphere points. In **(b)** we see that our theoretical prediction when training all layers is not as accurate as when we train only the last layer, but it is still highly correlated with the empirical max learning rate. It in general underpredicts, so that half of the theoretical learning rate should always have SGD converge. This is expected, since the NTK limit of training dynamics is only exact in the large width limit, and larger learning rate just means the training dynamics diverges from the NTK regime, but not necessarily that the training diverges. In **(c)**, we see that deeper networks tend to accept higher learning rate than our theoretical prediction. If we were to preprocess MNIST and CIFAR10 differently, then our theory is less accurate at predicting the max learning rate; see Fig. 9 for more details.

A. R. Barron. Universal approximation bounds for superpositions of a sigmoidal function. *IEEE Transactions on Information Theory*, 39(3):930–945, May 1993. ISSN 0018-9448. doi: 10.1109/18.256500.

Ronen Basri, David Jacobs, Yoni Kasten, and Shira Kritchman. The Convergence Rate of Neural Networks for Learned Functions of Different Frequencies. *arXiv:1906.00425 [cs, eess, stat]*, June 2019.

Agata Bezubik, Agata Dąbrowska, and Aleksander Strasburger. On spherical expansions of zonal functions on Euclidean spheres. *Archiv der Mathematik*, 90(1):70–81, January 2008. ISSN 0003-889X, 1420-8938. doi: 10.1007/s00013-007-2308-y.

Alberto Bietti and Julien Mairal. On the Inductive Bias of Neural Tangent Kernels. *arXiv:1905.12173 [cs, stat]*, May 2019.

Kenneth Blomqvist, Samuel Kaski, and Markus Heinonen. Deep convolutional Gaussian processes. *arXiv preprint arXiv:1810.03052*, 2018.

Anastasia Borovykh. A gaussian process perspective on convolutional neural networks. *arXiv preprint arXiv:1810.10798*, 2018.

John Bradshaw, Alexander G de G Matthews, and Zoubin Ghahramani. Adversarial examples, uncertainty, and transfer testing robustness in gaussian process hybrid deep networks. *arXiv preprint arXiv:1707.02476*, 2017.

Emmanuel J. Candès. Harmonic Analysis of Neural Networks. *Applied and Computational Harmonic Analysis*, 6(2):197–218, March 1999. ISSN 1063-5203. doi: 10.1006/acha.1998.0248.

Minmin Chen, Jeffrey Pennington, and Samuel Schoenholz. Dynamical Isometry and a Mean Field Theory of RNNs: Gating Enables Signal Propagation in Recurrent Neural Networks. In *Proceedings of the 35th International Conference on Machine Learning*, volume 80 of *Proceedings of Machine Learning Research*, pages 873–882, Stockholmsmässan, Stockholm Sweden, July 2018. PMLR.

Youngmin Cho and Lawrence K. Saul. Kernel methods for deep learning. In *Advances in Neural Information Processing Systems*, pages 342–350, 2009.

Andreas Damianou and Neil Lawrence. Deep gaussian processes. In *Artificial Intelligence and Statistics*, pages 207–215, 2013.

Amit Daniely, Roy Frostig, and Yoram Singer. Toward Deeper Understanding of Neural Networks: The Power of Initialization and a Dual View on Expressivity. *arXiv:1602.05897 [cs, stat]*, February 2016.

Simon S. Du, Xiyu Zhai, Barnabas Poczos, and Aarti Singh. Gradient Descent Provably Optimizes Over-parameterized Neural Networks. *arXiv:1810.02054 [cs, math, stat]*, October 2018.

Ronen Eldan and Ohad Shamir. The Power of Depth for Feedforward Neural Networks. In *Conference on Learning Theory*, pages 907–940, June 2016.

Adrià Garriga-Alonso, Laurence Aitchison, and Carl Edward Rasmussen. Deep Convolutional Networks as shallow Gaussian Processes. *arXiv:1808.05587 [cs, stat]*, August 2018.

Behrooz Ghorbani, Song Mei, Theodor Misiakiewicz, and Andrea Montanari. Linearized two-layers neural networks in high dimension. *arXiv:1904.12191 [cs, math, stat]*, April 2019.

Xavier Glorot and Yoshua Bengio. Understanding the difficulty of training deep feedforward neural networks. In Yee Whye Teh and Mike Titterington, editors, *Proceedings of the Thirteenth International Conference on Artificial Intelligence and Statistics*, volume 9 of *Proceedings of Machine Learning Research*, pages 249–256, Chia Laguna Resort, Sardinia, Italy, May 2010. PMLR. 02641.

Tilmann Gneiting. Strictly and non-strictly positive definite functions on spheres. *Bernoulli*, 19(4):1327–1349, September 2013. ISSN 1350-7265. doi: 10.3150/12-BEJSP06.

Boris Hanin. Which Neural Net Architectures Give Rise To Exploding and Vanishing Gradients? January 2018.

Boris Hanin and David Rolnick. How to Start Training: The Effect of Initialization and Architecture. *arXiv:1803.01719 [cs, stat]*, March 2018.

Soufiane Hayou, Arnaud Doucet, and Judith Rousseau. On the Selection of Initialization and Activation Function for Deep Neural Networks. *arXiv:1805.08266 [cs, stat]*, May 2018.

Tamir Hazan and Tommi Jaakkola. Steps Toward Deep Kernel Methods from Infinite Neural Networks. *arXiv:1508.05133 [cs]*, August 2015.

Kaiming He, Xiangyu Zhang, Shaoqing Ren, and Jian Sun. Delving deep into rectifiers: Surpassing human-level performance on imagenet classification. In *Proceedings of the IEEE International Conference on Computer Vision*, pages 1026–1034, 2015.

Arthur Jacot, Franck Gabriel, and Clément Hongler. Neural Tangent Kernel: Convergence and Generalization in Neural Networks. *arXiv:1806.07572 [cs, math, stat]*, June 2018. 00000.

Ryo Karakida, Shotaro Akaho, and Shun-ichi Amari. Universal Statistics of Fisher Information in Deep Neural Networks: Mean Field Approach. *arXiv:1806.01316 [cond-mat, stat]*, June 2018.

Yitzhak Katznelson. *An Introduction to Harmonic Analysis*. Cambridge Mathematical Library. Cambridge University Press, Cambridge, UK ; New York, 3rd ed edition, 2004. ISBN 978-0-521-83829-0 978-0-521-54359-0.

Vinayak Kumar, Vaibhav Singh, PK Srijith, and Andreas Damianou. Deep Gaussian Processes with Convolutional Kernels. *arXiv preprint arXiv:1806.01655*, 2018.

Neil D Lawrence and Andrew J Moore. Hierarchical Gaussian process latent variable models. In *Proceedings of the 24th International Conference on Machine Learning*, pages 481–488. ACM, 2007.

Nicolas Le Roux and Yoshua Bengio. Continuous neural networks. In *Artificial Intelligence and Statistics*, pages 404–411, 2007.

Jaehoon Lee, Yasaman Bahri, Roman Novak, Sam Schoenholz, Jeffrey Pennington, and Jascha Sohl-dickstein. Deep Neural Networks as Gaussian Processes. In *International Conference on Learning Representations*, 2018.

Jaehoon Lee, Lechao Xiao, Samuel S. Schoenholz, Yasaman Bahri, Jascha Sohl-Dickstein, and Jeffrey Pennington. Wide Neural Networks of Any Depth Evolve as Linear Models Under Gradient Descent. *arXiv:1902.06720 [cs, stat]*, February 2019.

Junhong Lin and Lorenzo Rosasco. Optimal Rates for Multi-pass Stochastic Gradient Methods. page 47.

Alexander\ G. \ de\ G. Matthews, Jiri Hron, Mark Rowland, Richard E. Turner, and Zoubin Ghahramani. Gaussian Process Behaviour in Wide Deep Neural Networks. In *International Conference on Learning Representations*, April 2018.

Radford M Neal. *BAYESIAN LEARNING FOR NEURAL NETWORKS*. PhD Thesis, University of Toronto, 1995.

Roman Novak, Lechao Xiao, Jaehoon Lee, Yasaman Bahri, Daniel A Abolafia, Jeffrey Pennington, and Jascha Sohl-Dickstein. Bayesian Deep Convolutional Networks with Many Channels are Gaussian Processes. *arXiv preprint arXiv:1810.05148*, 2018.

Ryan O'Donnell. *Analysis of Boolean Functions*. Cambridge University Press, New York, NY, 2014. ISBN 978-1-107-03832-5.

Jeffrey Pennington and Yasaman Bahri. Geometry of Neural Network Loss Surfaces via Random Matrix Theory. In Doina Precup and Yee Whye Teh, editors, *Proceedings of the 34th International Conference on Machine Learning*, volume 70 of *Proceedings of Machine Learning Research*, pages 2798–2806, International Convention Centre, Sydney, Australia, August 2017. PMLR. 00006.

Jeffrey Pennington and Pratik Worah. Nonlinear random matrix theory for deep learning. In *Advances in Neural Information Processing Systems*, pages 2634–2643, 2017. 00000.

Jeffrey Pennington and Pratik Worah. The Spectrum of the Fisher Information Matrix of a Single-Hidden-Layer Neural Network. In *Advances in Neural Information Processing Systems 31*, page 10, 2018.

Jeffrey Pennington, Samuel Schoenholz, and Surya Ganguli. Resurrecting the sigmoid in deep learning through dynamical isometry: Theory and practice. In I. Guyon, U. V. Luxburg, S. Bengio, H. Wallach, R. Fergus, S. Vishwanathan, and R. Garnett, editors, *Advances in Neural Information Processing Systems 30*, pages 4788–4798. Curran Associates, Inc., 2017a. 00004.

Jeffrey Pennington, Samuel S. Schoenholz, and Surya Ganguli. Resurrecting the sigmoid in deep learning through dynamical isometry: Theory and practice. *arXiv:1711.04735 [cs, stat]*, November 2017b. 00005.

Jeffrey Pennington, Samuel S. Schoenholz, and Surya Ganguli. The Emergence of Spectral Universality in Deep Networks. *arXiv:1802.09979 [cs, stat]*, February 2018.

George Philipp and Jaime G. Carbonell. The Nonlinearity Coefficient - Predicting Overfitting in Deep Neural Networks. *arXiv:1806.00179 [cs, stat]*, May 2018. 00000.

Ben Poole, Subhaneil Lahiri, Maithreyi Raghu, Jascha Sohl-Dickstein, and Surya Ganguli. Exponential expressivity in deep neural networks through transient chaos. In *Advances In Neural Information Processing Systems*, pages 3360–3368, 2016. 00047.

Nasim Rahaman, Aristide Baratin, Devansh Arpit, Felix Draxler, Min Lin, Fred A. Hamprecht, Yoshua Bengio, and Aaron Courville. On the Spectral Bias of Neural Networks. *arXiv:1806.08734 [cs, stat]*, June 2018.

Garvesh Raskutti, Martin J. Wainwright, and Bin Yu. Early stopping and non-parametric regression: An optimal data-dependent stopping rule. *arXiv:1306.3574 [stat]*, June 2013.

I. J. Schoenberg. Positive definite functions on spheres. *Duke Mathematical Journal*, 9(1):96–108, March 1942. ISSN 0012-7094, 1547-7398. doi: 10.1215/S0012-7094-42-00908-6.

Samuel S. Schoenholz, Justin Gilmer, Surya Ganguli, and Jascha Sohl-Dickstein. Deep Information Propagation. 2017.

Bernhard Schölkopf and Alexander J. Smola. *Learning with Kernels: Support Vector Machines, Regularization, Optimization, and Beyond*. Adaptive Computation and Machine Learning. MIT Press, Cambridge, Mass, 2002. ISBN 978-0-262-19475-4.

Alex J. Smola, Zoltán L. Óvári, and Robert C Williamson. Regularization with Dot-Product Kernels. In T. K. Leen, T. G. Dietterich, and V. Tresp, editors, *Advances in Neural Information Processing Systems 13*, pages 308–314. MIT Press, 2001.

Sho Sonoda and Noboru Murata. Neural Network with Unbounded Activation Functions is Universal Approximator. *Applied and Computational Harmonic Analysis*, 43(2):233–268, September 2017. ISSN 10635203. doi: 10.1016/j.acha.2015.12.005.

P.K. Suetin. Ultraspherical polynomials - Encyclopedia of Mathematics. https://www.encyclopediaofmath.org/index.php/Ultraspherical_polynomials.

Joel A. Tropp. An Introduction to Matrix Concentration Inequalities. *arXiv:1501.01571 [cs, math, stat]*, January 2015.

Guillermo Valle-Pérez, Chico Q. Camargo, and Ard A. Louis. Deep learning generalizes because the parameter-function map is biased towards simple functions. *arXiv:1805.08522 [cs, stat]*, May 2018.

Mark van der Wilk, Carl Edward Rasmussen, and James Hensman. Convolutional Gaussian Processes. In *Advances in Neural Information Processing Systems 30*, pages 2849–2858, 2017.

Yuting Wei, Fanny Yang, and Martin J Wainwright. Early stopping for kernel boosting algorithms: A general analysis with localized complexities. page 11.

Christopher K I Williams. Computing with Infinite Networks. In *Advances in Neural Information Processing Systems*, page 7, 1997.

Andrew G Wilson, Zhiting Hu, Ruslan R Salakhutdinov, and Eric P Xing. Stochastic Variational Deep Kernel Learning. In *Advances in Neural Information Processing Systems*, pages 2586–2594, 2016a.

Andrew Gordon Wilson, Zhiting Hu, Ruslan Salakhutdinov, and Eric P Xing. Deep kernel learning. In *Artificial Intelligence and Statistics*, pages 370–378, 2016b.

Lechao Xiao, Yasaman Bahri, Sam Schoenholz, and Jeffrey Pennington. Training ultra-deep CNNs with critical initialization. In *NIPS Workshop*, 2017.

Bo Xie, Yingyu Liang, and Le Song. Diverse Neural Network Learns True Target Functions. *arXiv:1611.03131 [cs, stat]*, November 2016.

Yuan Xu and E. W. Cheney. Strictly Positive Definite Functions on Spheres. *Proceedings of the American Mathematical Society*, 116(4):977–981, 1992. ISSN 0002-9939. doi: 10.2307/2159477.

Zhi-Qin John Xu, Yaoyu Zhang, and Yanyang Xiao. Training behavior of deep neural network in frequency domain. *arXiv:1807.01251 [cs, math, stat]*, July 2018.

Zhi-Qin John Xu, Yaoyu Zhang, Tao Luo, Yanyang Xiao, and Zheng Ma. Frequency Principle: Fourier Analysis Sheds Light on Deep Neural Networks. *arXiv:1901.06523 [cs, stat]*, January 2019.

Zhiqin John Xu. Understanding training and generalization in deep learning by Fourier analysis. *arXiv:1808.04295 [cs, math, stat]*, August 2018.

Greg Yang. Scaling Limits of Wide Neural Networks with Weight Sharing: Gaussian Process Behavior, Gradient Independence, and Neural Tangent Kernel Derivation. *arXiv:1902.04760 [cond-mat, physics:math-ph, stat]*, February 2019.

Greg Yang and Sam S. Schoenholz. Deep Mean Field Theory: Layerwise Variance and Width Variation as Methods to Control Gradient Explosion. February 2018.

Greg Yang and Samuel S. Schoenholz. Mean Field Residual Network: On the Edge of Chaos. In *Advances in Neural Information Processing Systems*, 2017.

Greg Yang, Jeffrey Pennington, Vinay Rao, Jascha Sohl-Dickstein, and Samuel S. Schoenholz. A Mean Field Theory of Batch Normalization. *arXiv:1902.08129 [cond-mat]*, February 2019.

Yuan Yao, Lorenzo Rosasco, and Andrea Caponnetto. On Early Stopping in Gradient Descent Learning. *Constructive Approximation*, 26(2):289–315, August 2007. ISSN 0176-4276, 1432-0940. doi: 10.1007/s00365-006-0663-2.

Yaoyu Zhang, Zhi-Qin John Xu, Tao Luo, and Zheng Ma. Explicitizing an Implicit Bias of the Frequency Principle in Two-layer Neural Networks. *arXiv:1905.10264 [cs, stat]*, May 2019.

Difan Zou, Yuan Cao, Dongruo Zhou, and Quanquan Gu. Stochastic Gradient Descent Optimizes Over-parameterized Deep ReLU Networks. *arXiv:1811.08888 [cs, math, stat]*, November 2018.

# A   RELATED WORKS

The Gaussian process behavior of neural networks was found by Neal (1995) for shallow networks and then extended over the years to different settings and architectures (Williams, 1997; Le Roux and Bengio, 2007; Hazan and Jaakkola, 2015; Daniely et al., 2016; Lee et al., 2018; Matthews et al., 2018; Novak et al., 2018). This connection was exploited implicitly or explicitly to build new models (Cho and Saul, 2009; Lawrence and Moore, 2007; Damianou and Lawrence, 2013; Wilson et al., 2016a;b; Bradshaw et al., 2017; van der Wilk et al., 2017; Kumar et al., 2018; Blomqvist et al., 2018; Borovykh, 2018; Garriga-Alonso et al., 2018; Novak et al., 2018; Lee et al., 2018). The Neural Tangent Kernel is a much more recent discovery by Jacot et al. (2018) and later Allen-Zhu et al. (2018a;c;b); Du et al. (2018); Arora et al. (2019b); Zou et al. (2018) came upon the same reasoning independently. Like CK, NTK has also been applied toward building new models or algorithms (Arora et al., 2019a; Achiam et al., 2019).

Closely related to the discussion of CK and NTK is the signal propagation literature, which tries to understand how to prevent pathological behaviors in randomly initialized neural networks when they are deep (Poole et al., 2016; Schoenholz et al., 2017; Yang and Schoenholz, 2017; 2018; Hanin, 2018; Hanin and Rolnick, 2018; Chen et al., 2018; Yang et al., 2019; Pennington et al., 2017a; Hayou et al., 2018; Philipp and Carbonell, 2018). This line of work can trace its original at least to the advent of the *Glorot* and *He* initialization schemes for deep networks (Glorot and Bengio, 2010; He et al., 2015). The investigation of *forward signal propagation*, or how random neural networks change with depth, corresponds to studying the infinite-depth limit of CK, and the investigation of *backward signal propagation*, or how gradients of random networks change with depth, corresponds to studying the infinite-depth limit of NTK. Some of the quite remarkable results from this literature includes how to train a 10,000 layer CNN (Xiao et al., 2017) and that, counterintuitively, batch normalization causes gradient explosion (Yang et al., 2019).

This signal propagation perspective can be refined via random matrix theory (Pennington et al., 2017a; 2018). In these works, free probability is leveraged to compute the singular value distribution of the input-output map given by the random neural network, as the input dimension and width tend to infinity together. Other works also investigate various questions of neural network training and generalization from the random matrix perspective (Pennington and Worah, 2017; Pennington and Bahri, 2017; Pennington and Worah, 2018).

Yang (2019) presents a common framework, known as *Tensor Programs*, unifying the GP, NTK, signal propagation, and random matrix perspectives, as well as extending them to new scenarios, like recurrent neural networks. It proves the existence of and allows the computation of a large number of infinite-width limits (including ones relevant to the above perspectives) by expressing the quantity of interest as the output of a computation graph and then manipulating the graph mechanically.

Several other works also adopt a spectral perspective on neural networks (Candès, 1999; Sonoda and Murata, 2017; Eldan and Shamir, 2016; Barron, 1993; Xu et al., 2018; Zhang et al., 2019; Xu et al., 2019; Xu, 2018); here we highlight a few most relevant to us. Rahaman et al. (2018) studies the *real Fourier frequencies* of relu networks and perform experiments on real data as well as synthetic ones. They convincingly show that relu networks learn low frequencies components first. They also investigate the subtleties when the data manifold is low dimensional and embedded in various ways in the input space. In contrast, our work focuses on the spectra of the CK and NTK (which indirectly informs the Fourier frequencies of a typical network). Nevertheless, our results are complementary to theirs, as they readily explain the low frequency bias in relu that they found. Karakida et al. (2018) studies the spectrum of the Fisher information matrix, which share the nonzero eigenvalues with the NTK. They compute the mean, variance, and maximum of the eigenvalues Fisher eigenvalues (taking the width to infinity first, and then considering finite amount of data sampled iid from a Gaussian). In comparison, our spectral results yield all eigenvalues of the NTK (and thus also all nonzero eigenvalues of the Fisher) as well as eigenfunctions.

Finally, we note that several recent works (Xie et al., 2016; Bietti and Mairal, 2019; Basri et al., 2019; Ghorbani et al., 2019) studied one-hidden layer neural networks over the sphere, building on Smola et al. (2001)'s observation that spherical harmonics diagonalize dot product kernels, with the latter two concurrent to us. This is in contrast to the focus on boolean cube here, which allows us to study the fine-grained effect of hyperparameters on the spectra, leading to a variety of insights into neural networks' generalization properties.

## B    UNIVERSALITY OF OUR BOOLEAN CUBE OBSERVATIONS IN OTHER INPUT DISTRIBUTIONS

Using the spectral theory we developed in this paper, we made three observations, that can be roughly summarized as follows: 1) the simplicity bias noted by Valle-Pérez et al. (2018) is not universal; 2) for each function of fixed "complexity" there is an optimal depth such that networks shallower or deeper will not learn it as well; 3) training last layer only is better than training all layers when learning "simpler" features, and the opposite is true for learning "complex" features.

In this section, we discuss the applicability of these observations to distributions that are not uniform over the boolean cube: in particular, the uniform distribution over the sphere $\sqrt{d}\mathcal{S}^{d-1}$, the standard Gaussian $\mathcal{N}(0, I_d)$, as well as realistic data distributions such as MNIST and CIFAR10.

**Simplicity bias**    The simplicity bias noted by Valle-Pérez et al. (2018), in particular Fig. 1, depends on the finiteness of the boolean cube as a domain, so we cannot effectively test this on the distributions above, which all have uncountable support.

**Optimal depth**    With regard to the second observation, we can test whether an *optimal depth* exists for learning functions over the distributions above. Since polynomial degrees remain the natural indicator of complexity for the sphere and the Gaussian (see Appendices H.2 and H.3 for the relevant spectral theory), we replicated the experiment in Fig. 3(b) for these distributions, using the same ground truth functions of polynomials of different degrees. The results are shown in Fig. 6. We see the same phenomenon as in the boolean cube case, with an optimal depth for each degree, and with the optimal depth increasing with degree.

For MNIST and CIFAR10, the notion of "feature complexity" is less clear, so we will not test the hypothesis that "optimal depth increases with degree" for these distributions but only test for the existence of the optimal depth for the ground truth marked by the labels of the datasets. We do so by training a large number of MLPs of varying depth on these datasets until convergence, and plot the results in Fig. 7. This figure clearly shows that such an optimal depth exists, such that shallower or deeper networks do monotonically worse as the depth diverge away from this optimal depth.

Again, the existence of the optimal depth is not obvious at all, as conventional deep learning wisdom would have one believe that adding depth should always help.

**Training last layer only vs training all layers**    Finally, we repeat the experiment in Fig. 3(c) for the sphere and the standard Gaussian, with polynomials of different degrees as ground truth functions. The results are shown in Fig. 8. We see the same phenomenon as in the boolean cube case: for degree 0 polynomials, training last layer works better in general, but for higher degree polynomials, training all layers fares better.

Note that, unlike the sphere and the Gaussian, whose spectral theory tells us that (harmonic) polynomial degree is a natural notion of complexity, for MNIST and CIFAR10 we have much less clear idea of what a "complex" or a "simple" feature is. Therefore, we did not attempt a similar experiment on these datasets.

## C    THEORETICAL VS EMPIRICAL MAX LEARNING RATES UNDER DIFFERENT PREPROCESSING FOR MNIST AND CIFAR10

In the main text Fig. 5, on the MNIST and CIFAR10 datasets, we preprocessed the data by centering and normalizing to the sphere (see Appendix E.2 for a precise description). With this preprocessing, our theory accurately predicts the max learning rate in practice.

In general, if we go by another preprocessing, such as PCA or ZCA, or no preprocessing, our theoretical max learning rate $1/\Phi(0)$ is less accurate but still correlated in general. The only exception seems to be relu networks on PCA- or ZCA- preprocessed CIFAR10. See Fig. 9.

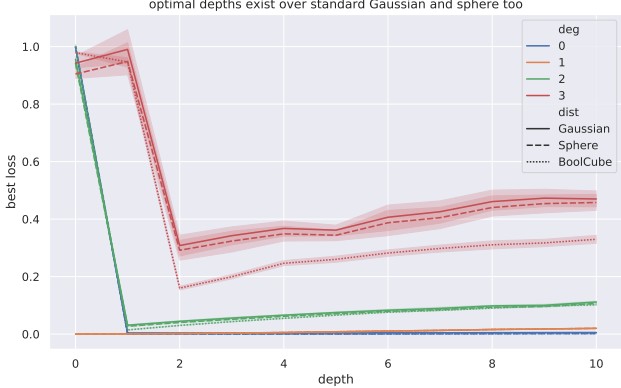

Figure 6: **Optimal depths exist over the standard Gaussian $\mathcal{N}(0, I_d)$ and the uniform distribution over the sphere $\sqrt{d}\mathcal{S}^{d-1}$ as well.** Here we use the exact same experimental setup as Fig. 3(b) (see Appendix E for details) except that the input distribution is changed from uniform over the boolean cube $\boxdot^d$ to standard Gaussian $\mathcal{N}(0, I_d)$ (solid lines) and uniform over the sphere $\sqrt{d}\mathcal{S}^{d-1}$ (dashed lines), where $d = 128$. We also compare against the results over the boolean cube from Fig. 3(b), which are drawn with dotted lines. Colors indicate the degrees of the ground truth polynomial functions. The best validation loss for degree 0 to 2 are all very close no matter which distribution the input is sampled from, such that the curves all sit on top of each other. For degree 3, there is less precise agreement between the validation loss over the different distributions, but the overall trend is unmistakably the same. We see that for networks deeper or shallower than the optimal depth, the loss monotonically increases as the depth moves away from the optimum.

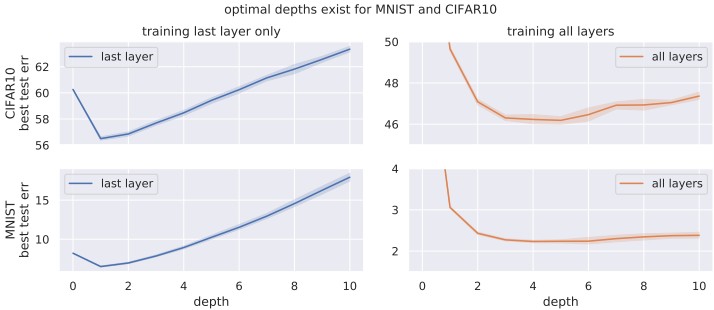

Figure 7: **Optimal depths exist over realistic distributions of MNIST and CIFAR10.** Here, we trained relu networks with $\sigma_w^2 = 2, \sigma_b^2 = 0$ for all depths from 0 to 10. We used SGD with learning rate 10 and batch size 256, and trained until convergence. We record the best test error throughout the training procedure for each depth. For each configuration, we repeat the randomly initialization and training for 10 random seeds to estimate the variance of the best test error. The **rows** demonstrate the best test error over the course of training on CIFAR10 and MNIST, and the **columns** demonstrate the same for training only the last layer or training all layers. As one can see, the best depth when training only the last layer is 1, for both CIFAR10 and MNIST. The best depth when training all layers is around 5 for both CIFAR10 and MNIST. Performance monotically decreases for networks shallower or deeper than the optimal depth. Note that we have reached the SOTA accuracy for MLPs reported in Lee et al. (2018) on CIFAR10, and within 1 point of their accuracy on MNIST.

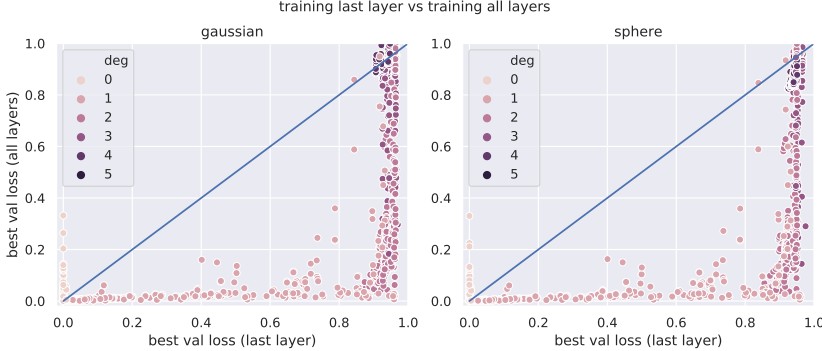

Figure 8: **Just like over the boolean cube: Over the sphere $\sqrt{d}\mathcal{S}^{d-1}$ and the standard Gaussian $\mathcal{N}(0, I_d)$, training only the last layer is better for learning low degree polynomials, but training all layers is better for learning high degree polynomials.** Here we use the exact same experimental setup as Fig. 3(c) (see Appendix E for details) except that the input distribution is changed from uniform over the boolean cube $\boxdot^d$ to standard Gaussian $\mathcal{N}(0, I_d)$ **(left)** and uniform over the sphere $\sqrt{d}\mathcal{S}^{d-1}$ **(right)**, where $d = 128$.

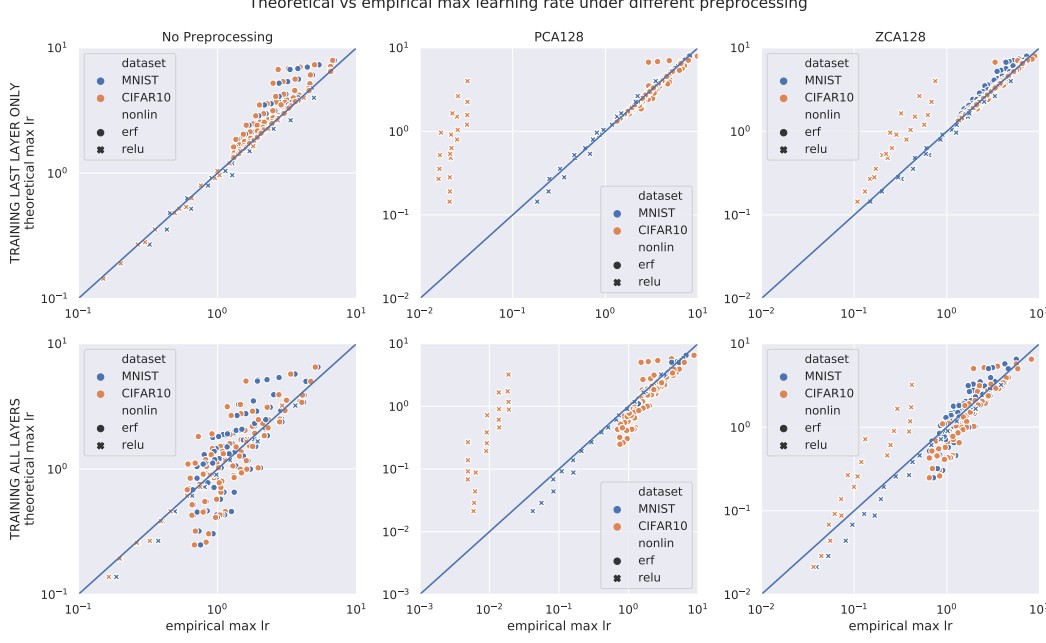

Figure 9: We perform binary search for the empirical max learning rate of MLPs of different depth, activations, $\sigma_w^2$, and $\sigma_b^2$ on MNIST and CIFAR10 preprocessed in different ways. See Appendix E.2 for experimental details. The **first row** compares the theoretical and empirical max learning rates when training only the last layer. The **second row** compares the same when training all layers (under NTK parametrization (Eq. (MLP))). The **three columns** correspond to the different preprocessing procedures: no preprocessing, PCA projection to the first 128 components (PCA128), and ZCA projection to the first 128 components (ZCA128). In general, the theoretical prediction is less accurate (compared to preprocessing by centering and projecting to the sphere, as in Fig. 5), though still well correlated with the empirical max learning rate. The most blatant caveat is the relu networks trained on PCA128- and ZCA128-processed CIFAR10.

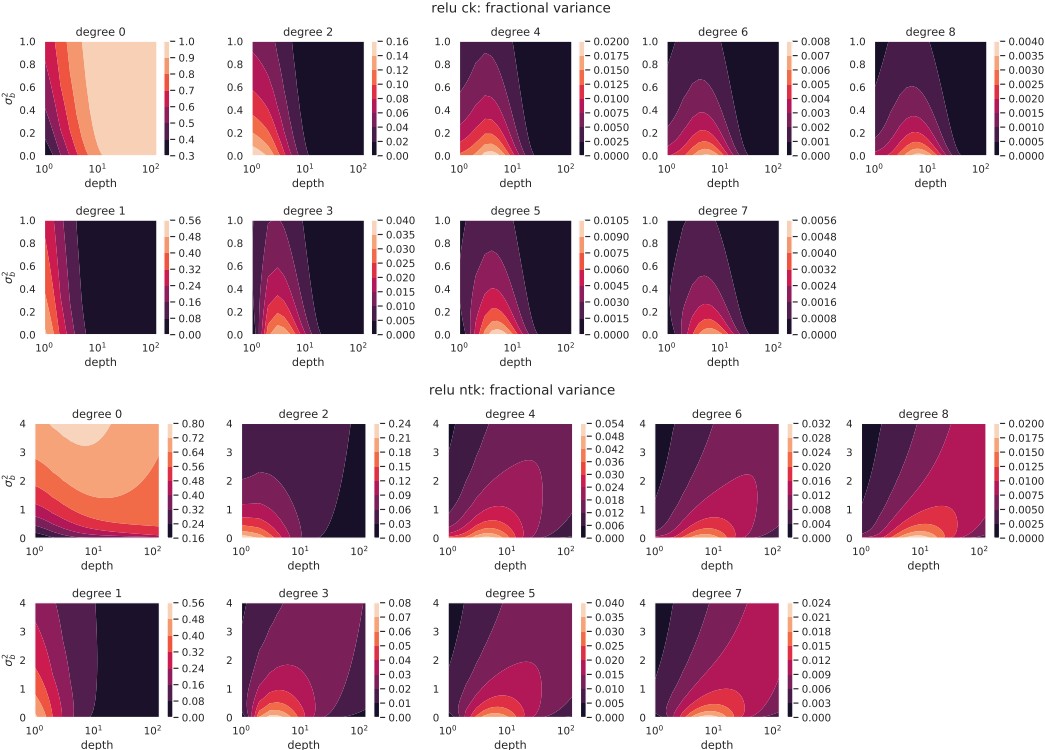

Figure 10: **2D contour plots of how fractional variance of each degree varies with $\sigma_b^2$ and depth, fixing $\sigma_w^2 = 2$, for relu CK and NTK**. For each degree $k$, and for each selected fractional variance value, we plot the level curve of $(\mathrm{depth}, \sigma_b^2)$ achieving this value. The color indicates the fractional variance, as given in the color bars.

# D   VISUALIZING THE SPECTRAL EFFECTS OF $\sigma_w^2, \sigma_b^2$, AND DEPTH

While in the main text, we summarized several trends of interest kn several plots, they do not give the entire picture of how eigenvalues and fractional variances vary with $\sigma_w^2, \sigma_b^2$, and depth. Here we try to present this relationship more completely in a series of contour plots. Fig. 10 shows how varying depth and $\sigma_b^2$ changes the fractional variances of each degree, for relu CK and NTK. We are fixing $\sigma_w^2 = 2$ in the CK plots, as the fractional variances only depend on the ratio $\sigma_b^2/\sigma_w^2$; even though this is not true for relu NTK, we fix $\sigma_w^2 = 2$ as well for consistency. For erf, however, the fractional variance will crucially depend on both $\sigma_w^2$ and $\sigma_b^2$, so we present 3D contour plots of how $\sigma_w^2, \sigma_b^2$, and depth changes fractional variance in Fig. 13. Complementarily, we also show in Figs. 11 and 12 a few slices of these 3D contour plots for different fixed values of $\sigma_b^2$, for erf CK and NTK.

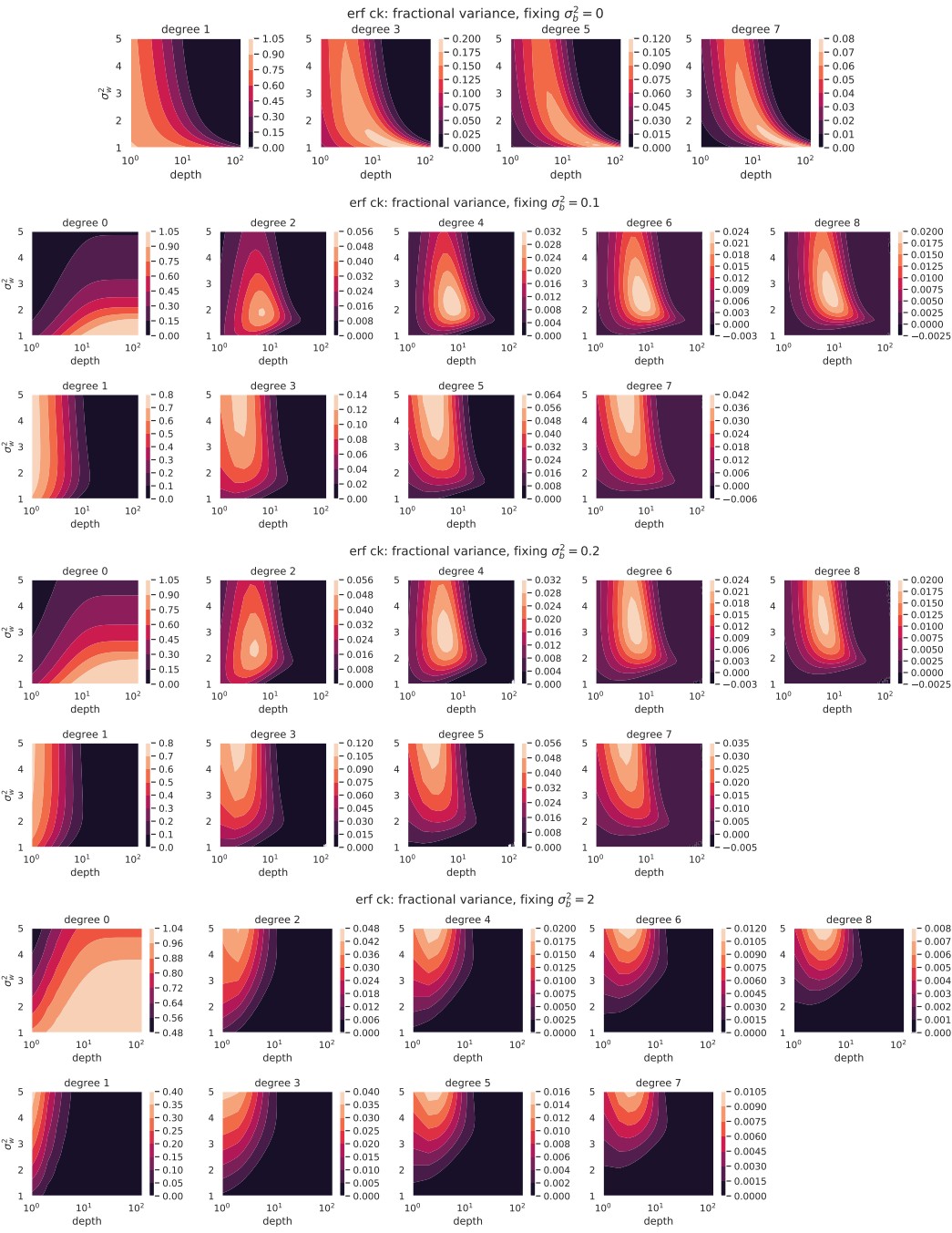

Figure 11: **2D contour plots of how fractional variance of each degree varies with $\sigma_w^2$ and depth, for different slices of $\sigma_b^2$, for erf NTK**. These plots essentially show slices of the NTK 3D contour plots in Fig. 13. For $\sigma_b = 0$, $\mu_k$ for all even degrees $k$ are 0, so we omit the plots. Note the rapid change in the shape of the contours for odd degrees, going from $\sigma_b^2 = 0$ to $\sigma_b^2 = 0.1$. This is reflected in Fig. 13 as well.

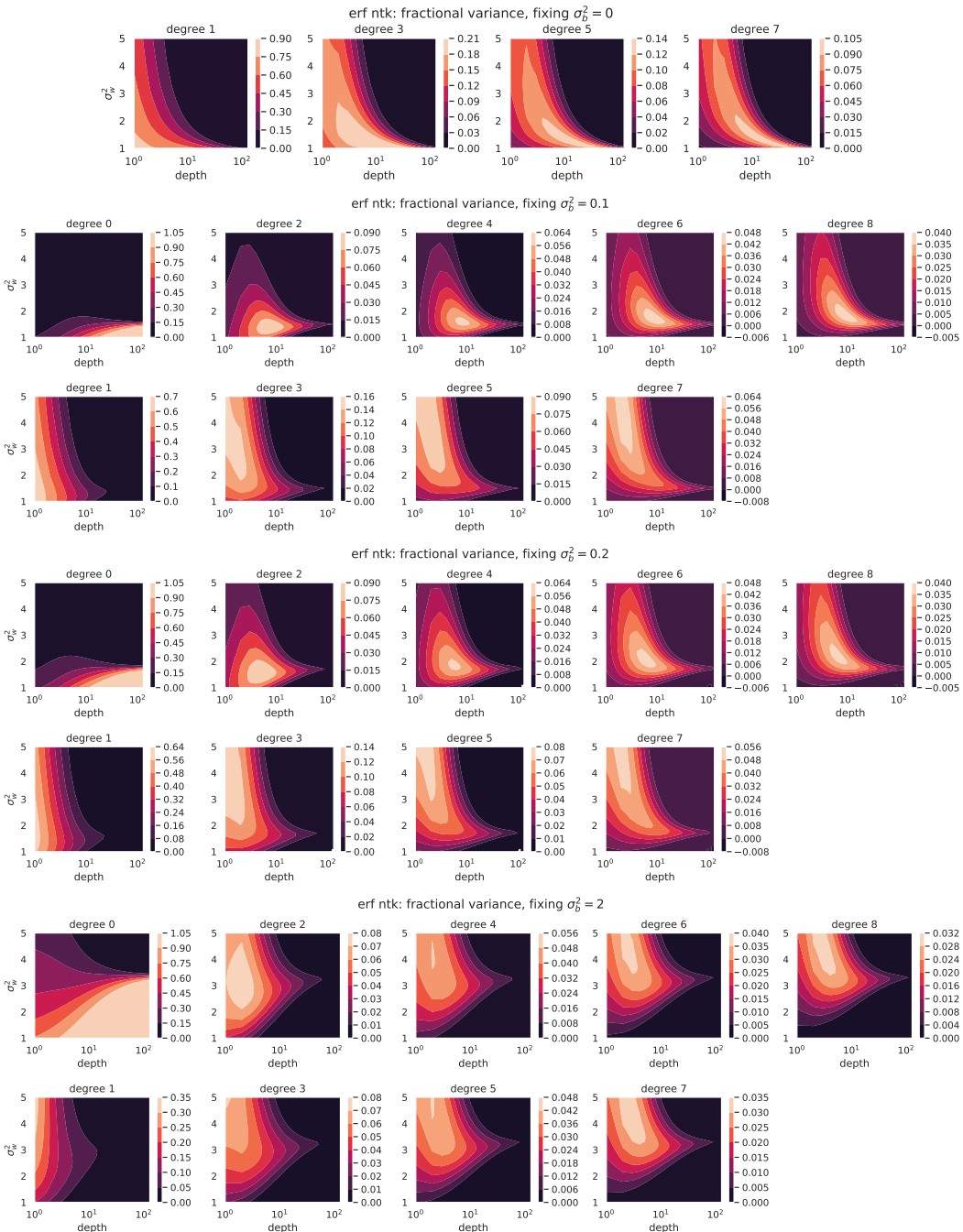

Figure 12: **2D contour plots of how fractional variance of each degree varies with $\sigma_w^2$ and depth, for different slices of $\sigma_b^2$, for erf CK**. These plots essentially show slices of the CK 3D contour plots in Fig. 13. For $\sigma_b = 0$, $\mu_k$ for all even degrees $k$ are 0, so we omit the plots. Note the rapid change in the shape of the contours for odd degrees, going from $\sigma_b^2 = 0$ to $\sigma_b^2 = 0.1$. This is reflected in Fig. 13 as well.

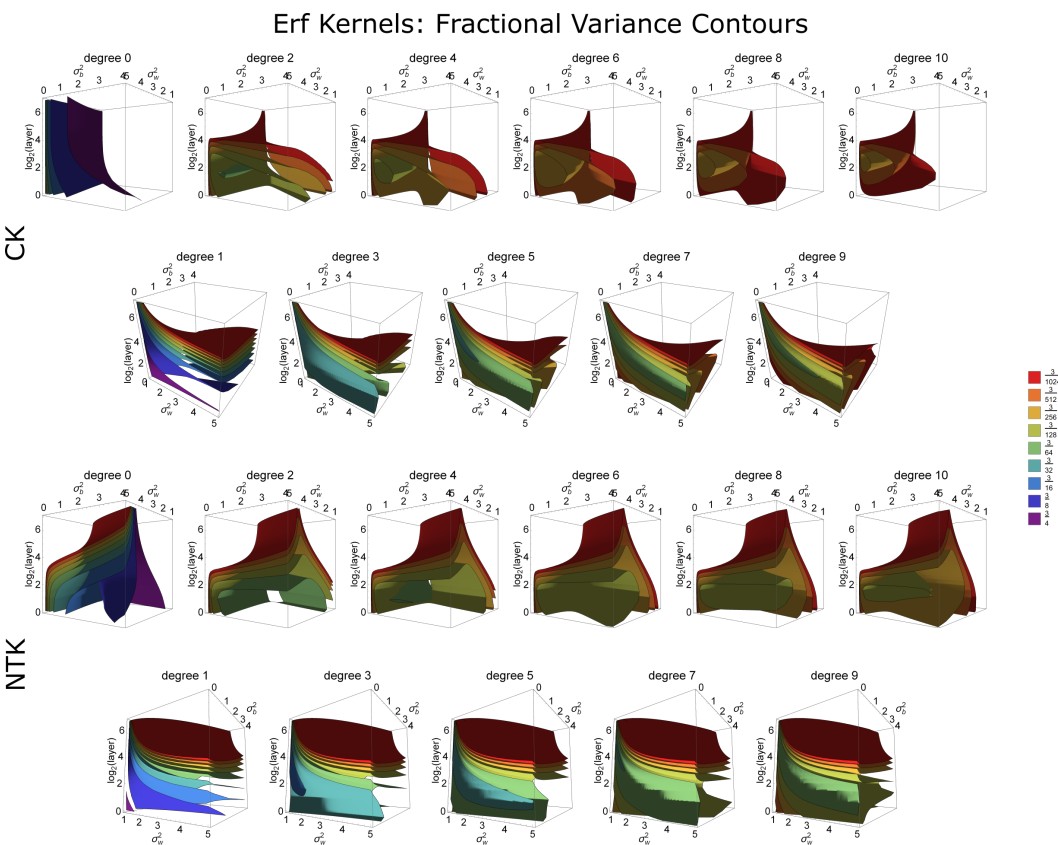

Figure 13: **3D contour plots of how fractional variance of each degree varies with $\sigma_w^2, \sigma_b^2$ and** $\log_2(\text{depth})$**, for erf CK and NTK**. For each value of fractional variance, as given in the legend on the right, we plot the level surface in the $(\sigma_w^2, \sigma_b^2, \log_2(\text{depth}))$-space achieving this value in the corresponding color. The closer to blue the color, the higher the value. Note that the contour for the highest values in higher degree plots "floats in mid-air", implying that there is an optimal depth for learning features of that degree that is not particularly small nor particularly big.

## E  EXPERIMENTAL DETAILS

### E.1  FIG. 3

Fig. 3(a), (b) and (c) differ in the set of hyperparameters they involve (to be specified below), but in all of them, we train relu networks against a randomly generated ground truth multilinear polynomial, with input space $\boxplus^{128}$ and L2 loss $L(f) = \mathbb{E}_{x \in \boxplus^d}(f(x) - f^*(x))^2$.

**Training**  We perform SGD with batch size 1000. In each iteration, we freshly sample a new batch, and we train for a total of 100,000 iterations, so the network potentially sees $10^8$ different examples. At every 1000 iterations, we validate the current network on a freshly drawn batch of 10,000 examples. We thus record a total of 100 validation losses, and we take the lowest to be the "best validation loss."

**Generating the Ground Truth Function**  The ground truth function $f^*(x)$ is generated by first sampling 10 monomials $m_1, \ldots, m_{10}$ of degree $k$, then randomly sampling 10 coefficients $a_1, \ldots, a_{10}$ for them. The final function is obtained by normalizing $\{a_i\}$ such that the sum of their squares is 1:

$$f^*(x) \stackrel{\text{def}}{=} \sum_{i=1}^{10} a_i m_i / \sum_{j=1}^{10} a_j^2. \tag{11}$$

**Hyperparameters for Fig. 3(a)**

- The learning rate is half the theoretical maximum learning rate[7] $\frac{1}{2} \max(\mu_0, \mu_1)^{-1}$
- Ground truth degree $k \in \{0, 1, 2, 3\}$
- Depth $\in \{0, \ldots, 10\}$
- activation = relu
- $\sigma_w^2 = 2$
- $\sigma_b^2 = 0$
- width = 1000
- 10 random seeds per hyperparameter combination
- training last layer (marked "ck"), or all layers (marked "ntk"). In the latter case, we use the NTK parametrization of the MLP (Eq. (MLP)).

**Hyperparameters for Fig. 3(b)**

- The learning rate is half the theoretical maximum learning rate $\frac{1}{2} \max(\mu_0, \mu_1)^{-1}$
- Ground truth degree $k \in \{0, 1, 2, 3\}$
- Depth $\in \{0, \ldots, 10\}$
- activation = relu
- $\sigma_w^2 = 2$
- $\sigma_b^2 = 0$
- width = 1000
- 100 random seeds per hyperparameter combination
- training last layer weight and bias only

---

[7]Note that, because the L2 loss here is $L(f) = \mathbb{E}_{x \in \boxplus^d}(f(x) - f^*(x))^2$, the maximum learning rate is $\lambda_{max}^{-1} = \max(\mu_0, \mu_1)^{-1}$ (see Thm 4.1). If we instead adopt the convention $L(f) = \mathbb{E}_{x \in \boxplus^d} \frac{1}{2}(f(x) - f^*(x))^2$, then the maximum learning rate would be $2\lambda_{max}^{-1} = 2 \max(\mu_0, \mu_1)^{-1}$

---

**Algorithm 1** Binary Search for Empirical Max Learning Rate

---

$upper \leftarrow 16 \times$ theoretical max lr
$lower \leftarrow 0$
$tol \leftarrow 0.01 \times$ theoretical max lr
**while** $|upper - lower| > tol$ **do**
    $\alpha \leftarrow (upper + lower)/2$
    Run SGD with learning rate $\alpha$ for 1000 iterations
    **if** loss diverges **then**
        $upper \leftarrow \alpha$
    **else**
        $lower \leftarrow \alpha$
    **end if**
**end while**
**Output:** $upper$

---

**Hyperparameters for Fig. 3(c)**

- The learning rate $\in \{0.05, 0.1, 0.5\}$
- Ground truth degree $k \in \{0, 1, \ldots, 6\}$
- Depth $\in \{1, \ldots, 5\}$
- activation $\in \{\mathrm{relu}, \mathrm{erf}\}$
- $\sigma_w^2 = 2$ for relu, but $\sigma_w^2 \in \{1, 2, \ldots, 5\}$ for erf
- $\sigma_b^2 \in \{0, 1, \ldots, 4\}$
- width = 1000
- 1 random seed per hyperparameter combination
- Training all layers, using the NTK parametrization of the MLP (Eq. (MLP))

### E.2 MAX LEARNING RATE EXPERIMENTS

Here we describe the experimental details for the experiments underlying Figs. 5 and 9.

**Theoretical max learning rate** For a fixed setup, we compute $\Phi$ according to Eq. (CK) (if only last layer is trained) or Eq. (NTK) (if all layers are trained). For ground truth problems where the output is $n$-dimensional, the theoretical max learning rate is $n\Phi(0)^{-1}$; in particular, the max learning rates for MNIST and CIFAR10 are 10 times those for boolean cube, sphere, and Gaussian. This is because the kernel for an multi-output problem effectively becomes

$$\frac{1}{n} K^{\oplus n} = \frac{1}{n} \begin{pmatrix} K & 0 & 0 \\ 0 & \ddots & 0 \\ 0 & 0 & K \end{pmatrix}$$

where the $\frac{1}{n}$ factor is due to the $\frac{1}{n}$ factor in the scaled square loss $\mathcal{L}(f, f^*) = \mathbb{E}_{x \sim \mathcal{X}} \frac{1}{n} \sum_{i=1}^{n} (f(x)_i - f^*(x)_i)^2$. The top eigenvalue for $\frac{1}{n} K^{\oplus n}$ is just $\frac{1}{n}$ times the top eigenvalue for $K$.

**Empirical max learning rate** For a fixed setup, we perform binary search for the empirical max learning rate as in Algorithm 1.

**Preprocessing** In Fig. 5, for MNIST and CIFAR10, we center and project each image onto the sphere $\sqrt{d}\mathcal{S}^{d-1}$, where $d = 28 \times 28 = 784$ for MNIST and $d = 3 \times 32 \times 32 = 3072$ for CIFAR10. More precisely, we compute the average image $\bar{x}$ over the entire dataset, and we preprocess each image $x$ as $\sqrt{d}\frac{x-\bar{x}}{\|x-\bar{x}\|}$. In Fig. 9, there are three different preprocessing schemes. For "no preprocessing," we load the MNIST and CIFAR10 data as is. In "PCA128," we take the top 128 eigencomponents of the data, so that the data has only 128 dimensions. In "ZCA128," we take the top 128 eigencomponents but rotate it back to the original space, so that the data still has dimension $d$, where $d = 28 \times 28 = 784$ for MNIST and $d = 3 \times 32 \times 32 = 3072$ for CIFAR10.

**Hyperparameters**

- Target function: For boolean cube, sphere, and standard Gaussian, we randomly sample a degree 1 polynomial as in Eq. (11). For MNIST and CIFAR10, we just use the label in the dataset, encoded as a one-hot vector for square-loss regression.
- Depth $\in \{1, 2, 4, 8, 16\}$
- activation $\in \{\mathrm{relu}, \mathrm{erf}\}$
- $\sigma_w^2 = 2$ for relu, but $\sigma_w^2 \in \{1, 2, \ldots, 5\}$ for erf
- $\sigma_b^2 \in \{1, \ldots, 4\}$
- width = 1000
- 1 random seed per hyperparameter combination
- Training last layer (CK) or all layers (NTK). In the latter case, we use the NTK parametrization of the MLP (Eq. (MLP)).

## F    REVIEW OF THE THEORY OF NEURAL TANGENT KERNELS

### F.1    CONVERGENCE OF INFINITE-WIDTH KERNELS AT INITIALIZATION

**Conjugate Kernel**    Via a central-limit-like intuition, each unit $h^l(x)_\alpha$ of Eq. (MLP) should behave like a Gaussian as width $n^{l-1} \to \infty$, as it is a sum of a large number of roughly independent random variables (Poole et al., 2016; Schoenholz et al., 2017; Yang and Schoenholz, 2017). The devil, of course, is in what "roughly independent" means and how to apply the central limit theorem (CLT) to this setting. It can be done, however, (Lee et al., 2018; Matthews et al., 2018; Novak et al., 2018), and in the most general case, using a "Gaussian conditioning" technique, this result can be rigorously generalized to almost any architecture Yang (2019). In any case, the consequence is that, for any finite set $S \subseteq \mathcal{X}$,

$$\{h_\alpha^l(x)\}_{x \in S} \quad \text{converges in distribution to} \quad \mathcal{N}(0, \Sigma^l(S, S)),$$

as $\min\{n^1, \ldots, n^{l-1}\} \to \infty$, where $\Sigma^l$ is the CK as given in Eq. (CK).

**Neural Tangent Kernel**    By a slightly more involved version of the "Gaussian conditioning" technique, Yang (2019) also showed that, for any $x, y \in \mathcal{X}$,

$$\langle \nabla_\theta h^L(x), \nabla_\theta h^L(y) \rangle \quad \text{converges almost surely to} \quad \Theta^L(x, y)$$

as the widths tend to infinity, where $\Theta^l$ is the NTK as given in Eq. (NTK).

### F.2    FAST EVALUATIONS OF CK AND NTK

For certain $\phi$ like relu or erf, $V_\phi$ and $V_\phi'$ can be evaluated very quickly, so that both the CK and NTK can be computed in $O(|\mathcal{X}|^2 L)$ time, where $\mathcal{X}$ is the set of points we want to compute the kernel function over, and $L$ is the number of layers.

**Fact F.1** (Cho and Saul (2009)). *For any kernel $K$*

$$V_{\mathrm{relu}}(K)(x, x') = \frac{1}{2\pi}(\sqrt{1 - c^2} + (\pi - \arccos c)c)\sqrt{K(x, x)K(x', x')}$$

$$V_{\mathrm{relu}}'(K)(x, x') = \frac{1}{2\pi}(\pi - \arccos c)$$

*where $c = K(x, x')/\sqrt{K(x, x)K(x', x')}$.*

**Fact F.2** (Neal (1995)). *For any kernel $K$,*

$$V_{\mathrm{erf}}(K)(x, x') = \frac{2}{\pi} \arcsin \frac{K(x, x')}{\sqrt{(K(x, x) + 0.5)(K(x', x') + 0.5)}}$$

$$V_{\mathrm{erf}}'(K)(x, x') = \frac{4}{\pi\sqrt{(1 + 2K(x, x))(1 + 2K(x', x')) - 4K(x, x')^2}}.$$

**Fact F.3.** *Let $\phi(x) = \exp(x/\sigma)$ for some $\sigma > 0$. For any kernel $K$,*

$$V_\phi(K)(x, x') = \exp\left(\frac{K(x,x) + 2K(x,x') + K(x',x')}{2\sigma^2}\right).$$

## F.3 LINEAR EVOLUTION OF NEURAL NETWORK UNDER GD

Remarkably, the NTK governs the evolution of the neural network function under gradient descent in the infinite-width limit. First, let's consider how the parameters $\theta$ and the neural network function $f$ evolve under continuous time gradient flow. Suppose $f$ is only defined on a finite input space $\mathcal{X} = \{x^1, \ldots, x^k\}$. We will visualize

$$f(\mathcal{X}) = \begin{bmatrix} f(x^1) \\ \vdots \\ f(x^k) \end{bmatrix}, \quad \nabla_f\mathcal{L} = \begin{bmatrix} \frac{\partial \mathcal{L}}{\partial f(x^1)} \\ \vdots \\ \frac{\partial \mathcal{L}}{\partial f(x^k)} \end{bmatrix}, \quad \theta = \begin{bmatrix} \theta_1 \\ \vdots \\ \theta_n \end{bmatrix}, \quad \nabla_\theta f = \begin{bmatrix} \frac{\partial f(x^1)}{\partial \theta_1} & \cdots & \frac{\partial f(x^k)}{\partial \theta_1} \\ \vdots & \ddots & \vdots \\ \frac{\partial f(x^1)}{\partial \theta_n} & \cdots & \frac{\partial f(x^k)}{\partial \theta_n} \end{bmatrix}$$

(best viewed in color). Then under continuous time gradient descent with learning rate $\eta$,

$$\partial_t \, \theta_t = -\eta \nabla_\theta \mathcal{L}(f_t) = -\eta \, \nabla_\theta f_t \, \cdot \, \nabla_f \mathcal{L}(f_t),$$

$$\partial_t \, f_t = \nabla_\theta f_t^\top \cdot \partial_t \, \theta_t = -\eta \, \nabla_\theta f_t^\top \cdot \nabla_\theta f_t \cdot \nabla_f \mathcal{L}(f_t) = -\eta \, \Theta_t \cdot \nabla_f \mathcal{L}(f_t) \qquad (12)$$

where $\Theta_t = \nabla_\theta f_t^\top \cdot \nabla_\theta f_t \in \mathbb{R}^{k \times k}$ is of course the (finite width) NTK. These equations can be visualized as

$$\partial_t \, \boxed{\phantom{x}} = -\eta \, \boxed{\phantom{xx}} \cdot \boxed{\phantom{x}}, \quad \partial_t \, \boxed{\phantom{x}} = \boxed{\phantom{xx}} \cdot \partial_t \, \boxed{\phantom{x}} = -\eta \, \boxed{\phantom{xx}} \cdot \boxed{\phantom{xx}} \cdot \boxed{\phantom{x}} = -\eta \, \boxed{\phantom{xx}} \cdot \boxed{\phantom{x}}$$

Thus $f$ undergoes kernel gradient descent with (functional) loss $\mathcal{L}(f)$ and kernel $\Theta_t$. This kernel $\Theta_t$ of course changes as $f$ evolves, but remarkably, it in fact stays constant for $f$ being an infinitely wide MLP (Jacot et al., 2018):

$$\partial_t f_t = -\eta \Theta \cdot \nabla_f \mathcal{L}(f_t), \qquad \text{(Training All Layers)}$$

where $\Theta$ is the infinite-width NTK corresponding to $f$. A similar equation holds for the CK $\Sigma$ if we train only the last layer,

$$\partial_t f_t = -\eta \Sigma \cdot \nabla_f \mathcal{L}(f_t). \qquad \text{(Training Last Layer)}$$

If $\mathcal{L}$ is the square loss against a ground truth function $f^*$, then $\nabla_f \mathcal{L}(f_t) = \frac{1}{2k} \nabla_f \|f_t - f^*\|^2 = \frac{1}{k}(f_t - f^*)$, and the equations above become linear differential equations. However, typically we only have a training set $\mathcal{X}^{\text{train}} \subseteq \mathcal{X}$ of size far less than $|\mathcal{X}|$. In this case, the loss function is effectively

$$\mathcal{L}(f) = \frac{1}{2|\mathcal{X}^{\text{train}}|} \sum_{x \in \mathcal{X}^{\text{train}}} (f(x) - f^*(x))^2,$$

with functional gradient

$$\nabla_f \mathcal{L}(f) = \frac{1}{|\mathcal{X}^{\text{train}}|} D^{\text{train}} \cdot (f - f^*),$$

where $D^{\text{train}}$ is a diagonal matrix of size $k \times k$ whose diagonal is 1 on $x \in \mathcal{X}^{\text{train}}$ and 0 else. Then our function still evolves linearly

$$\partial_t f_t = -\eta (K \cdot D^{\text{train}}) \cdot (f_t - f^*) \qquad (13)$$

where $K$ is the CK or the NTK depending on which parameters are trained.

### F.4 RELATIONSHIP TO GAUSSIAN PROCESS INFERENCE.

Recall that the initial $f_0$ in Eq. (13) is distributed as a Gaussian process $\mathcal{N}(0, \Sigma)$ in the infinite width limit. As Eq. (13) is a linear differential equation, the distribution of $f_t$ will remain a Gaussian process for all $t$, whether $K$ is CK or NTK. Under suitable conditions, it can be shown that (Lee et al., 2019), in the limit as $t \to \infty$, if we train only the last layer, then the resulting function $f_\infty$ is distributed as a Gaussian process with mean $\bar{f}_\infty$ given by

$$\bar{f}_\infty(x) = \Sigma(x, \mathcal{X}^{\text{train}})\Sigma(\mathcal{X}^{\text{train}}, \mathcal{X}^{\text{train}})^{-1}f^*(\mathcal{X}^{\text{train}})$$

and kernel $\mathrm{Var}\, f_\infty$ given by

$$\mathrm{Var}\, f_\infty(x, x') = \Sigma(x, x') - \Sigma(x, \mathcal{X}^{\text{train}})\Sigma(\mathcal{X}^{\text{train}}, \mathcal{X}^{\text{train}})^{-1}\Sigma(\mathcal{X}^{\text{train}}, x').$$

These formulas precisely described the posterior distribution of $f$ given prior $\mathcal{N}(0, \Sigma)$ and data $\{(x, f^*(x))\}_{x \in \mathcal{X}^{\text{train}}}$.

If we train all layers, then similarly as $t \to \infty$, the function $f_\infty$ is distributed as a Gaussian process with mean $\bar{f}_\infty$ given by (Lee et al., 2019)

$$\bar{f}_\infty(x) = \Theta(x, \mathcal{X}^{\text{train}})\Theta(\mathcal{X}^{\text{train}}, \mathcal{X}^{\text{train}})^{-1}f^*(\mathcal{X}^{\text{train}}).$$

This is, again, the mean of the Gaussian process posterior given prior $\mathcal{N}(0, \Theta)$ and the training data $\{(x, f^*(x))\}_{x \in \mathcal{X}^{\text{train}}}$. However, the kernel of $f_\infty$ is no longer the kernel of this posterior, but rather is an expression involving both the NTK $\Theta$ and the CK $\Sigma$; see Lee et al. (2019).

In any case, we can make the following informal statement in the limit of large width

*Training the last layer (resp. all layers) of an MLP infinitely long, in expectation, yields the mean prediction of the GP inference given prior $\mathcal{N}(0, \Sigma)$ (resp. $\mathcal{N}(0, \Theta)$).*

## G A BRIEF REVIEW OF HILBERT-SCHMIDT OPERATORS AND THEIR SPECTRAL THEORY

In this section, we briefly review the theory of Hilbert-Schmidt kernels, and more importantly, to properly define the notion of eigenvalues and eigenfunctions. A function $K : \mathcal{X}^2 \to \mathbb{R}$ is called a *Hilbert-Schmidt* operator if $K \in L^2(\mathcal{X} \times \mathcal{X})$, i.e.

$$\|K\|_{HS}^2 \overset{\text{def}}{=} \underset{x,y \sim \mathcal{X}}{\mathbb{E}} K(x, y)^2 < \infty.$$

$\|K\|_{HS}^2$ is known as the *Hilbert-Schmidt* norm of $K$. $K$ is called *symmetric* if $K(x, y) = K(y, x)$ and *positive definite (resp. semidefinite)* if

$$\underset{x,y \sim \mathcal{X}}{\mathbb{E}} f(x)K(x, y)f(y) > 0 \text{ (resp. } \geq 0) \quad \text{for all } f \in L^2(\mathcal{X}) \text{ not a.e. zero.}$$

A spectral theorem (Mercer's theorem) holds for Hilbert-Schmidt operators.

**Fact G.1.** *If $K$ is a symmetric positive semidefinite Hilbert-Schmidt kernel, then there is a sequence of scalars $\lambda_i \geq 0$ (eigenvalues) and functions $f_i \in L^2(\mathcal{X})$ (eigenfunctions), for $i \in \mathbb{N}$, such that*

$$\forall i, j, \ \langle f_i, f_j \rangle = \mathbb{I}(i = j), \quad and \quad K(x, y) = \sum_{i \in \mathbb{N}} \lambda_i f_i(x) f_i(y)$$

*where the convergence is in $L^2(\mathcal{X} \times \mathcal{X})$ norm.*

This theorem allows us to speak of the eigenfunctions and eigenvalues, which are important for training and generalization considerations when $K$ is a kernel used in machine learning, as discussed in the main text.

A sufficient condition for $K$ to be a Hilbert-Schmidt kernel in our case (concerning only probability measure on $\mathcal{X}$) is just that $K$ is bounded. All $K$s in this paper satisfy this property.

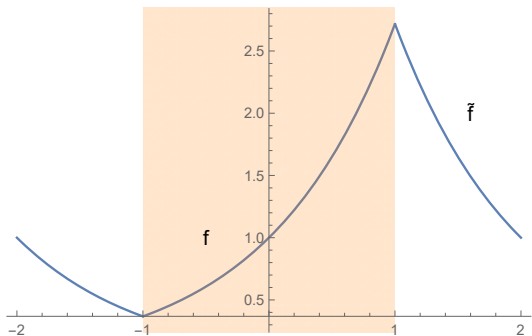

Figure 14: Example of $\tilde{f}$ given $f$

# H EIGENDECOMPOSITION OF NEURAL KERNEL ON DIFFERENT DOMAINS

## H.1 BOOLEAN CUBE

**From the Fourier Series Perspective.** We continue from the discussion of the boolean cube in the main text. Recall that $\mathcal{T}_\Delta$ is the shift operator on functions that sends $\Phi(\cdot)$ to $\Phi(\cdot - \Delta)$. Notice that, if we let $\Phi(t) = e^{\kappa t}$ for some $\kappa \in \mathbb{C}$, then $\mathcal{T}_\Delta \Phi(s) = e^{-\kappa\Delta} \cdot e^{\kappa t}$. Thus $\Phi$ is an "eigenfunction" of the operator $\mathcal{T}_\Delta$ with eigenvalue $e^{-\kappa\Delta}$. In particular, this implies that

**Proposition H.1.** *Suppose $\Phi(t) = e^{t/\sigma^2}$, as in the case when $K$ is the CK or NTK of a 1-layer neural network with nonlinearity $\exp(\cdot/\sigma)$, up to multiplicative constant (Fact F.3). Then the eigenvalue $\mu_k$ over the boolean cube $\mathbb{D}^d$ equals*

$$\mu_k = 2^{-d}(1 - \exp(-\Delta/\sigma^2))^k(1 + \exp(-\Delta/\sigma^2))^{d-k} \cdot \exp(1/\sigma^2)$$

*where $\Delta = 2/d$.*

It would be nice if we can express any $\Phi$ as a linear combination of exponentials, so that Eq. (5) simplifies in the fashion of Prop H.1 — this is precisely the idea of Fourier series.

We will use the theory of Fourier analysis on the circle, and for this we need to discuss periodic functions. Let $\tilde{\Phi} : [-2, 2] \to \mathbb{R}$ be defined as

$$\tilde{\Phi}(x) = \begin{cases} \Phi(x) & \text{if } x \in [-1, 1] \\ \Phi(2 - x) & \text{if } x \in [1, 2] \\ \Phi(-2 - x) & \text{if } x \in [-2, -1]. \end{cases}$$

See Fig. 14 for an example illustration. Note that if $\Phi$ is continuous on $[-1, 1]$, then $\tilde{\Phi}$ is continuous as a periodic function on $[-2, 2]$.

The Fourier basis on functions over $[-2, 2]$ is the collection $\{t \mapsto e^{\frac{1}{2}\pi i s t}\}_{s \in \mathbb{Z}}$. Under generic conditions (for example if $\Psi \in L^2[-2, 2]$), a function $\Psi$ has an associated Fourier series $\sum_{s \in \mathbb{Z}} \hat{\Psi}(s) e^{\frac{1}{2}\pi i s t}$. We briefly review basic facts of Fourier analysis on the circle. Recall the following notion of *functions of bounded variation*.

**Definition H.2.** A function $f : [a, b] \to \mathbb{R}$ is said to have *bounded variation* if

$$\sup_P \sum_{i=0}^{n_P - 1} |f(x_{i+1}) - f(x_i)| < \infty,$$

where the supremum is taken over all partitions $P$ of the interval $[a, b]$,

$$P = \{x_0, \ldots, x_{n_P}\}, \quad x_0 \le x_1 \le \cdots \le x_{n_P}.$$

Intuitively, a function of bounded variation has a graph (in $[a, b] \times \mathbb{R}$) of finite length.

**Fact H.3** (Katznelson (2004)). *A bounded variation function $f : [-2, 2] \to \mathbb{R}$ that is periodic (i.e. $f(-2) = f(2)$) has a pointwise-convergent Fourier series:*

$$\lim_{T \to \infty} \sum_{s \in [-T, T]} \hat{\Psi}(s) e^{\frac{1}{2}\pi i s t} \to \Psi(t), \forall t \in [-2, 2].$$

From this fact easily follows the following lemma.

**Lemma H.4.** *Suppose $\Phi$ is continuous and has bounded variation on $[-1, 1]$. Then $\tilde{\Phi}$ is also continuous and has bounded variation, and its Fourier Series (on $[-2, 2]$) converges pointwise to $\tilde{\Phi}$.*

*Proof.* $\tilde{\Phi}$ is obviously continuous and has bounded variation as well, and from Fact H.3, we know a periodic continuous function with bounded variation has a pointwise-convergent Fourier Series. $\square$

Certainly, $\mathcal{T}_\Delta$ sends continuous bounded variation functions to continuous bounded variation functions. Because $\mathcal{T}_\Delta e^{\frac{1}{2}\pi i s t} = e^{-\frac{1}{2}\pi i s \Delta} e^{\frac{1}{2}\pi i s t}$,

$$\mathcal{T}_\Delta \sum_{s \in \mathbb{Z}} \hat{\Psi}(s) e^{\frac{1}{2}\pi i s t} = \sum_{s \in \mathbb{Z}} \hat{\Psi}(s) e^{-\frac{1}{2}\pi i s \Delta} e^{\frac{1}{2}\pi i s t}$$

whenever both sides are well defined. If $\Psi$ is continuous and has bounded variation then $\mathcal{T}_\Delta \Psi$ is also continuous and has bounded variation, and thus its Fourier series, the RHS above, converges pointwise to $\mathcal{T}_\Delta \Psi$.

Now, observe

$$(I - \mathcal{T}_\Delta)^k (I + \mathcal{T}_\Delta)^{d-k} \tilde{\Phi}(x) = \sum_{r=0}^{d} C_r^{d-k,k} \tilde{\Phi}(x - r\Delta)$$

$$(I - \mathcal{T}_\Delta)^k (I + \mathcal{T}_\Delta)^{d-k} \tilde{\Phi}(1) = \sum_{r=0}^{d} C_r^{d-k,k} \Phi\left(\left(\frac{d}{2} - r\right)\Delta\right)$$

$$= \mu_k$$

Expressing the LHS in Fourier basis, we obtain

**Theorem H.5.**

$$\mu_k = \sum_{s \in \mathbb{Z}} i^s (1 - e^{-\frac{1}{2}\pi i s \Delta})^k (1 + e^{-\frac{1}{2}\pi i s \Delta})^{d-k} \hat{\tilde{\Phi}}(s)$$

*where*

$$\hat{\tilde{\Phi}}(s) = \frac{1}{4} \int_{-2}^{2} \tilde{\Phi}(t) e^{-\frac{1}{2}\pi i s t} \, dt$$

$$= \frac{1}{4} \int_{-1}^{1} \Phi(t) (e^{-\frac{1}{2}\pi i s t} + (-1)^s e^{\frac{1}{2}\pi i s t}) \, dt$$

$$= \begin{cases} \frac{1}{2} \int_{-1}^{1} \Phi(t) \cos(\frac{1}{2}\pi s t) \, dt & \text{if } s \text{ is even} \\ -\frac{i}{2} \int_{-1}^{1} \Phi(t) \sin(\frac{1}{2}\pi s t) \, dt & \text{if } s \text{ is odd} \end{cases}$$

*denote the Fourier coefficients of $\tilde{\Phi}$ on $[-2, 2]$. (Here $i$ is the imaginary unit here, not an index).*

**Recovering the values of $\Phi$ given the eigenvalues $\mu_0, \ldots, \mu_d$.** Conversely, given eigenvalues $\mu_0, \ldots, \mu_d$ corresponding to each monomial degree, we can recover the entries of the matrix $K$.

**Theorem H.6.** *For any $x, y \in \boxed{\pm 1}^d$ with Hamming distance $r$,*

$$K(x, y) = \Phi\left(\left(\frac{d}{2} - r\right)\Delta\right) = \sum_{k=0}^{d} C_k^{d-r,r} \mu_k,$$

*where $C_k^{d-r,r} = \sum_{j=0}^{}(-1)^{k+j} \binom{d-r}{j} \binom{r}{k-j}$ as in Eq. (7).*

*Proof.* Recall that for any $S \subseteq [d]$, $\chi_S(x) = x^S$ is the Fourier basis corresponding to $S$ (see Eq. (3)). Then by converting from the Fourier basis to the regular basis, we get

$$\Phi\left(\left(\frac{d}{2} - r\right)\Delta\right) = K(x, y) \quad \text{for any } x, y \in \boxed{\pm 1}^d \text{ with Hamming distance } r$$

$$= \sum_{k=0}^{d} \mu_k \sum_{|S|=k} \chi_S(x) \chi_S(y).$$

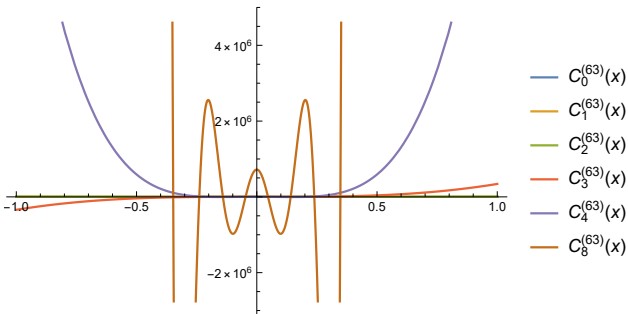

Figure 15: Examples of Gegenbauer Polynomials for $d = 128$ (or $\alpha = 63$).

If $x$ and $y$ differ on a set $T \subseteq [d]$, then we can simplify the inner sum

$$\Phi\left(\left(\frac{d}{2} - r\right)\Delta\right) = \sum_{k=0}^{d} \mu_k \sum_{|S|=k} (-1)^{|S \cap T|} = \sum_{k=0}^{d} \mu_k C_k^{d-r,r}.$$

$\square$

*Remark* H.7. If we let $\mathcal{T}$ be the operator that sends $\mu_\bullet \mapsto \mu_{\bullet+1}$, then we have the following operator expression

$$\Phi\left(\left(\frac{d}{2} - r\right)\Delta\right) = [(1 + \mathcal{T})^{d-r}(1 - \mathcal{T})^r \mu]_0$$

*Remark* H.8. The above shows that the matrix $C = \{C_k^{d-r,r}\}_{k,r=0}^{d}$ satisfies

$$C^2 = 2^d I.$$

### H.2 SPHERE

Now let's consider the case when $\mathcal{X} = \sqrt{d}\mathcal{S}^{d-1}$ is the radius-$\sqrt{d}$ sphere in $\mathbb{R}^d$ equipped with the uniform measure. Again, because $x \in \mathcal{X}$ all have the same norm, we will consider $\Phi$ as a univariate function with $K(x, y) = \Phi(\langle x, y \rangle / \|x\|\|y\|) = \Phi(\langle x, y \rangle / d)$.

As is long known (Schoenberg, 1942; Gneiting, 2013; Xu and Cheney, 1992; Smola et al., 2001), $K$ is diagonalized by spherical harmonics. We review these results briefly below, as we will build on them to deduce spectral information of $K$ on isotropic Gaussian distributions.

**Review: spherical harmonics and Gegenbauer polynomials.** *Spherical harmonics* are $L^2$ functions on $\mathcal{S}^{d-1}$ that are eigenfunctions of the Laplace-Beltrami operator $\Delta_{\mathcal{S}^{d-1}}$ of $\mathcal{S}^{d-1}$. They can be described as the restriction of certain homogeneous polynomials in $\mathbb{R}^d$ to $\mathcal{S}^{d-1}$. Denote by $\mathcal{H}^{d-1,(l)}$ the space of spherical harmonics of degree $l$ on sphere $\mathcal{S}^{d-1}$. Then we have the orthogonal decomposition $L^2(\mathcal{S}^{d-1}) \cong \bigoplus_{l=0}^{\infty} \mathcal{H}^{d-1,(l)}$. It is a standard fact that $\dim \mathcal{H}^{d-1,(l)} = \binom{d-1+l}{d-1} - \binom{d-3+l}{d-1}$.

There is a special class of spherical harmonics called *zonal harmonics* that can be represented as $x \mapsto p(\langle x, y \rangle)$ for specific polynomials $p : \mathbb{R} \to \mathbb{R}$, and that possess a special *reproducing property* which we will describe shortly. Intuitively, the value of any zonal harmonics only depends on the "height" of $x$ along some fixed axis $y$, so a typical zonal harmonics looks like Fig. 16. The polynomials $p$ must be one of the Gegenbauer polynomials. *Gegenbauer polynomials* $\{C_l^{(\alpha)}(t)\}_{l=0}^{\infty}$ are orthogonal polynomials with respect to the measure $(1 - t^2)^{\alpha - \frac{1}{2}}$ on $[-1, 1]$ (see Fig. 15 for examples), and here we adopt the convention that

$$\int_{-1}^{1} C_n^{(\alpha)}(t) C_l^{(\alpha)}(t)(1 - t^2)^{\alpha - \frac{1}{2}} \, \mathrm{d}t = \frac{\pi 2^{1-2\alpha}\Gamma(n + 2\alpha)}{n!(n + \alpha)[\Gamma(\alpha)]^2} \mathbb{I}(n = l). \tag{14}$$

Then for each (oriented) axis $y \in \mathcal{S}^{d-1}$ and degree $l$, there is a unique zonal harmonic

$$Z_y^{d-1,(l)} \in \mathcal{H}^{d-1,(l)}, \quad Z_y^{d-1,(l)}(x) \stackrel{\text{def}}{=} c_{d,l}^{-1} C_l^{(\frac{d-2}{2})}(\langle x, y \rangle)$$

for any $x, y \in \mathcal{S}^{d-1}$, where $c_{d,l} = \frac{d-2}{d+2l-2}$. Very importantly, they satisfy the following

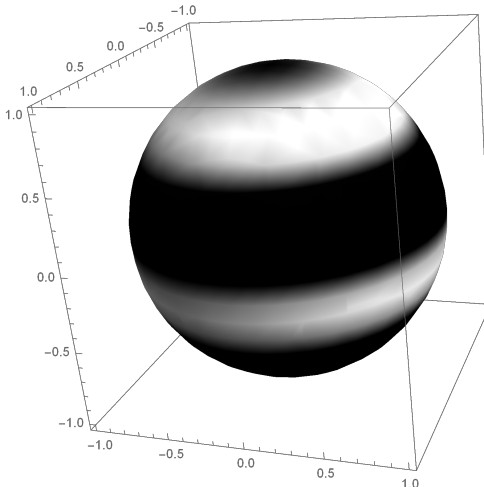

Figure 16: Visualization of a zonal harmonic, which depends only on the "height" of the input along a fixed axis. Color indicates function value.

**Fact H.9** (Reproducing property (Suetin)). *For any $f \in \mathcal{H}^{d-1,(m)}$,*

$$\mathbb{E}_{z \sim \mathcal{S}^{d-1}} Z_y^{d-1,(l)}(z) f(z) = f(y) \mathbb{I}(l = m)$$

$$\mathbb{E}_{z \sim \mathcal{S}^{d-1}} Z_y^{d-1,(l)}(z) Z_x^{d-1,(m)}(z) = Z_y^{d-1,(l)}(x) \mathbb{I}(l = m) = c_{d,l}^{-1} C_l^{(\frac{d-2}{2})}(\langle x, y \rangle) \mathbb{I}(l = m)$$

We also record a useful fact about Gegenbauer polynomials.

**Fact H.10** (Suetin).

$$C_l^{(\alpha)}(\pm 1) = (\pm 1)^l \binom{l + 2\alpha - 1}{l}$$

By a result of Schoenberg (1942), we have the following eigendecomposition of $K$ on the sphere.

**Theorem H.11** (Schoenberg). *Suppose $\Phi : [-1, 1] \to \mathbb{R}$ is in $L^2((1 - t^2)^{\frac{d-1}{2} - 1})$, so that it has the Gegenbauer expansion*

$$\Phi(t) \overset{a.e.}{=} \sum_{l=0}^{\infty} a_l c_{d,l}^{-1} C_l^{(\frac{d-2}{2})}(t).$$

*Then $K$ has eigenspaces $\mathcal{H}_{\sqrt{d}}^{d-1,(l)} \overset{def}{=} \{f(x/\sqrt{d}) : f \in \mathcal{H}^{d-1,(l)}\}$ with corresponding eigenvalues $a_l$. Since $\bigoplus_{l=0}^{\infty} \mathcal{H}_{\sqrt{d}}^{d-1,(l)}$ is an orthogonal decomposition of $L^2(\sqrt{d}\mathcal{S}^{d-1})$, this describes all eigenfunctions of $K$ considered as an operator on $L^2(\sqrt{d}\mathcal{S}^{d-1})$.*

For completeness, we include the proof of this theorem in Appendix I.

By Bezubik et al. (2008), we can express the Gegenbauer coefficients, and thus equivalently the eigenvalues, via derivatives of $\Phi$:

**Theorem H.12** (Bezubik et al. (2008)). *If the Taylor expansion of $\Phi$ at 0,*

$$\Phi(t) = \sum_{n=0}^{\infty} \frac{\Phi^{(n)}(0)}{n!} t^n,$$

*is absolutely convergent on the closed interval $[-1, 1]$, then the Gegenbauer coefficients $a_l$ in Thm H.11 in dimension $d$ is equal to the absolute convergent series*

$$a_l = \Gamma\left(\frac{d}{2}\right) \sum_{k=0}^{\infty} \frac{\Phi^{(l+2k)}(0)}{2^{l+2k} k! \Gamma\left(\frac{d}{2} + l + k\right)}. \tag{15}$$

As the dimension $d$ of the sphere tends to $\infty$, the eigenvalues in fact simplify to the derivatives of $\Phi$:

**Theorem H.13.** *Let $K$ be the CK or NTK of an MLP on the sphere $\sqrt{d}\mathcal{S}^{d-1}$. Then $K$ can be expressed as $K(x,y) = \Phi(\langle x,y \rangle/d)$ for some smooth $\Phi : [-1,1] \to \mathbb{R}$. Let $a_\ell$ denote $K$'s eigenvalues on the sphere (as in Thm H.11). If we fix $\ell$ and let $d \to \infty$, then*

$$\lim_{d\to\infty} d^\ell a_\ell = \Phi^{(\ell)}(0),$$

*where $\Phi^{(\ell)}$ denotes the $\ell$th derivative of $\Phi$.*

This theorem is the same as Thm I.6 except that it concerns the sphere rather than the boolean cube.

*Proof.* By Thm I.3, $\Phi$'s Taylor expansion around 0 is absolutely convergent on $[-1,1]$, so that the condition of Thm H.12 is satisfied. Therefore, Eq. (15) holds and is absolutely convergent. By dominated convergence theorem, we can exchange the limit and the summation, and get

$$\begin{aligned}
\lim_{d\to\infty} d^l a_l &= \lim_{d\to\infty} d^l \Gamma\left(\frac{d}{2}\right) \sum_{k=0}^{\infty} \frac{\Phi^{(l+2k)}(0)}{2^{l+2k}k!\Gamma\left(\frac{d}{2}+l+k\right)} \\
&= \sum_{k=0}^{\infty} \Phi^{(l+2k)}(0) \lim_{d\to\infty} \frac{d^l \Gamma\left(\frac{d}{2}\right)}{2^{l+2k}k!\Gamma\left(\frac{d}{2}+l+k\right)} \\
&= \sum_{k=0}^{\infty} \Phi^{(l+2k)}(0) \lim_{d\to\infty} \left(\frac{d}{2}\right)^{-k} (k!)^{-1} 2^{-2k} \\
&= \Phi^{(l)}(0)
\end{aligned}$$

as desired. $\qquad\square$

### H.3 ISOTROPIC GAUSSIAN

Now let's consider $\mathcal{X} = \mathbb{R}^d$ equipped with standard isotropic Gaussian $\mathcal{N}(0,I)$, so that $K$ behaves like

$$Kf(x) = \mathop{\mathbb{E}}_{y\sim\mathcal{N}(0,I)} K(x,y)f(y) = \mathop{\mathbb{E}}_{y\sim\mathcal{N}(0,I)} \Phi\left(\frac{\langle x,y \rangle}{\|x\|\|y\|}, \frac{\|x\|^2}{d}, \frac{\|y\|^2}{d}\right) f(y)$$

for any $f \in L^2(\mathcal{N}(0,I))$. In contrast to the previous two sections, $K$ will essentially depend on the effect of the norms $\|x\|$ and $\|y\|$ on $\Phi$.

Note that an isotropic Gaussian vector $z \sim \mathcal{N}(0,I)$ can be sampled by independently sampling its direction $v$ uniformly from the sphere $\mathcal{S}^{d-1}$ and sampling its magnitude $r$ from a chi distribution $\chi_d$ with $d$ degrees of freedom. Proceeding along this line of logic yields the following spectral theorem:

**Theorem H.14.** *A function $K : (\mathbb{R}^d)^2 \to \mathbb{R}$ of the form*

$$K(x,y) = \Phi\left(\frac{\langle x,y \rangle}{\|x\|\|y\|}, \frac{\|x\|^2}{d}, \frac{\|y\|^2}{d}\right)$$

*forms a positive semidefinite Hilbert-Schmidt operator on $L^2(\mathcal{N}(0,I))$ iff $\Phi$ can be decomposed as*

$$\Phi(t,q,q') = \sum_{l=0}^{\infty} A_l(q,q') c_{d,l}^{-1} C_l^{\left(\frac{d-2}{2}\right)}(t) \tag{16}$$

*satisfying*

$$\sum_{l=0}^{\infty} \|A_l\|^2 c_{d,l}^{-2} \|C_l^{\left(\frac{d-2}{2}\right)}\|^2 = \sum_{l=0}^{\infty} \|A_l\|^2 c_{d,l}^{-2} \frac{\pi 2^{3-d}\Gamma(l+d-2)}{l!(l+\frac{d-2}{2})\Gamma\left(\frac{d-2}{2}\right)^2} < \infty, \tag{17}$$

*where*

- $c_{d,l} = \frac{d-2}{d+2l-2}$

- $C_l^{(\frac{d-2}{2})}(t)$ *are Gegenbauer polynomials as in Appendix H.2, with* $\|C_l^{(\frac{d-2}{2})}\| = \sqrt{\int_{-1}^{1} C_l^{(\frac{d-2}{2})}(t)^2(1-t^2)^{\frac{d-1}{2}-1}\,\mathrm{d}t}$ *denoting the norm of* $C_l^{(\frac{d-2}{2})}$ *in* $L^2((1-t^2)^{\frac{d-1}{2}-1})$.

- *and* $A_l$ *are positive semidefinite Hilbert-Schmidt kernels on* $L^2(\frac{1}{d}\chi_d^2)$, *the* $L^2$ *space over the probability measure of a* $\chi_d^2$-*variable divided by* $d$, *and with* $\|A_l\|$ *denoting the Hilbert-Schmidt norm of* $A_l$.

*In addition,* $K$ *is positive definite iff all* $A_l$ *are.*

See Appendix I for a proof. As a consequence, $K$ has an eigendecomposition as follows under the standard Gaussian measure in $d$ dimensions.

**Corollary H.15.** *Suppose* $K$ *and* $\Phi$ *are as in Thm H.14 and* $K$ *is a positive semidefinite Hilbert-Schmidt operator, so that Eq.* (16) *holds, with Hilbert-Schmidt kernels* $A_l$. *Let* $A_l$ *have eigendecomposition*

$$A_l(q,q') = \sum_{i=0}^{\infty} a_{li} u_{li}(q) u_{li}(q') \tag{18}$$

*for eigenvalues* $a_{li} \geq 0$ *and eigenfunctions* $u_{li} \in L^2(\frac{1}{d}\chi_d^2)$ *with* $\mathbb{E}_{q\sim\frac{1}{d}\chi_d^2} u_{li}(q)^2 = 1$. *Then* $K$ *has eigenvalues* $\{a_{li} : l, i \in [0,\infty)\}$, *and each eigenvalue* $a_{li}$ *corresponds to the eigenspace*

$$u_{li} \otimes \mathcal{H}^{d-1,(l)} \overset{\mathrm{def}}{=} \left\{ u_{li}\left(\frac{\|x\|^2}{d}\right) f\left(\frac{x}{\|x\|}\right) : f \in \mathcal{H}^{d-1,(l)} \right\},$$

*where* $\mathcal{H}^{d-1,(l)}$ *is the space of degree* $l$ *spherical harmonics on the unit sphere* $\mathcal{S}^{d-1}$ *of ambient dimension* $d$.

For certain simple $F$, we can obtain $\{A_l\}_{l\geq0}$ explicitly. For example, suppose $K$ is *degree-s positive-homogeneous*, in the sense that, for $a, b > 0$,

$$K(ax, by) = (ab)^s K(x,y).$$

This happens when $K$ is the CK or NTK of an MLP with degree-$s$ positive-homogeneous. Then it's easy to see that $\Phi(t,q,q') = (qq')^s \bar{\Phi}(t)$ for some $\bar{\Phi} : [-1,1] \to \mathbb{R}$, and

$$\Phi(t,q,q') = \sum_{l=0}^{\infty} (qq')^s a_l c_{d,l}^{-1} C_l^{(\frac{d-2}{2})}(t)$$

where $\{a_l\}_l$ are the Gegenbauer coefficients of $\bar{\Phi}$,

$$\bar{\Phi}(t) = \sum_{l=0}^{\infty} a_l c_{d,l}^{-1} C_l^{(\frac{d-2}{2})}(t).$$

We can then conclude with the following theorem.

**Theorem H.16.** *Suppose* $K : (\mathbb{R}^d)^2 \to \mathbb{R}$ *is a kernel given by*

$$K(x,y) = R(\|x\|/d)R(\|x\|/d)\bar{\Phi}(\langle x,y \rangle / \|x\|\|y\|)$$

*for some functions* $R : [0,\infty) \to \mathbb{R}$, $\bar{\Phi} : [-1,1] \to \mathbb{R}$. *Let*

$$\bar{\Phi}(t) = \sum_{l=0}^{\infty} a_l c_{d,l}^{-1} C_l^{(\frac{d-2}{2})}(t)$$

*be the Gegenbauer expansion of* $\bar{\Phi}$. *Also define*

$$\lambda \overset{\mathrm{def}}{=} \sqrt{\mathbb{E}_{q\sim\frac{1}{d}\chi_d^2} R(q)^2}.$$

*Then over the standard Gaussian in* $\mathbb{R}^d$, $K$ *has the following eigendecomposition*

- *For each $l \geq 0$,*

$$\lambda^{-1} R \otimes \mathcal{H}^{d-1,(l)} = \{\lambda^{-1} R(\|x\|^2/d) f(x/\|x\|) : f \in \mathcal{H}^{d-1,(l)}\}$$

  *is an eigenspace with eigenvalue $\lambda^2 a_l$.*

- *For any $S \in L^2(\frac{1}{d}\chi_d^2)$ that is orthogonal to $R$, i.e.*

$$\mathop{\mathbb{E}}_{q \sim \frac{1}{d}\chi_d^2} S(q) R(q) = 0,$$

  *the function*

$$S(\|x\|^2/d) f(x/\|x\|)$$

  *for any $f \in L^2(\mathcal{S}^{d-1})$ is in the null space of $K$.*

*Proof.* The $A_l$ in Eq. (16) for $K$ are all equal to

$$A_l(q, q') = \lambda^2 \frac{R(q)}{\lambda} \frac{R(q')}{\lambda}.$$

This is a rank 1 kernel (on $L^2(\frac{1}{d}\chi_d^2)$), with eigenfunction $R/\lambda$ and eigenvalue $\lambda^2$. The rest then follows straightforwardly from Thm H.11. $\qquad\square$

A common example where Thm H.16 applies is when $K$ is the CK of an MLP with relu, or more generally degree-$s$ positive homogeneous activation functions, so that the $R$ in Thm H.16 is a polynomial.

In general, we cannot expect $K$ can be exactly diagonalized in a natural basis, as $\{A_l\}_{l \geq 0}$ cannot even be simultaneously diagonalizable. We can, however, investigate the "variance due to each degree of spherical harmonics" by computing

$$a_l \overset{\text{def}}{=} \mathop{\mathbb{E}}_{q \sim d^{-1}\chi_d^2} A_l(q, q) \tag{19}$$

which is the coefficient of Gegenbauer polynomials in

$$\hat{\Phi}_d(t) \overset{\text{def}}{=} \mathop{\mathbb{E}}_{q \sim d^{-1}\chi_d^2} \Phi(t, q, q) = \sum_{l=0}^{\infty} a_l c_{d,l}^{-1} C_l^{(\frac{d-2}{2})}(t). \tag{20}$$

**Proposition H.17.** *Assume that $\Phi(t, q, q')$ is continuous in $q$ and $q'$. Suppose for any $d$ and any $t \in [-1, 1]$, the random variable $\Phi(t, q, q')$ with $q, q' \sim d^{-1}\chi_d^2$ has a bound $|\Phi(t, q, q')| \leq Y$ for some random variable $Y$ with $\mathbb{E}|Y| < \infty$. Then for every $t \in [-1, 1]$,*

$$\lim_{d \to \infty} |\hat{\Phi}_d(t) - \Phi(t, 1, 1)| = 0.$$

*Proof.* By the strong law of large number, $d^{-1}\chi_d^2$ converges to 1 almost surely. Because $\Phi(t, q, q')$ is continuous in $q$ and $q'$, almost surely we have $\Phi(t, q, q') \to \Phi(t, 1, 1)$ almost surely. Since $\Phi$ is bounded by $Y$, by dominated convergence, we have

$$\hat{\Phi}_d(t) - \Phi(t, 1, 1) = \mathop{\mathbb{E}}_{q,q' \sim d^{-1}\chi_d^2} \Phi(t, q, q') - \Phi(t, 1, 1) \to 0$$

as desired. $\qquad\square$

In the final part of this section, we show that in the limit of large input dimension $d$, the top eigenvalues of $K$ over the standard Gaussian can be easily described (Thm H.19). First, we need to specify some conditions on $\Phi$.

**Definition H.18.** $\Phi : [-1, 1] \times \mathbb{R}^+ \times \mathbb{R}^+ \to \mathbb{R}$ is called *reasonable* if

- There is a Taylor expansion in $t$,

$$\Phi(t, q, q') = \sum_{\ell=0}^{\infty} \frac{B_\ell(q, q')}{\ell!} t^\ell \tag{21}$$

  that is absolutely convergent on $(t, q, q') \in [-1, 1] \times \mathbb{R}^+ \times \mathbb{R}^+$, and such that each $B_\ell$ is smooth ($C^\infty$) on $(q, q') \in \mathbb{R}^+ \times \mathbb{R}^+$.

- for all $(t, q, q') \in [-1, 1] \times \mathbb{R}^+ \times \mathbb{R}^+$, and for any $l \in [0, \infty)$, we have $|B_l(t, q, q')| \leq C(1 + |q|^r + |q'|^r)$ for some constants $C, r > 0$ that may depend on $l$ but not on $t, q, q'$.

**Theorem H.19.** *Suppose $K$ is a the CK or NTK of an MLP with polynomially bounded activation function. For every degree $l$, $K$ over $\mathcal{N}(0, I_d)$ has an eigenvalue $a_{l0}$ at spherical harmonics degree $l$ (in the sense of Cor H.15) with*

$$a_{l0} \in \left[ \mathop{\mathbb{E}}_{q, q' \sim \frac{1}{d} \chi_d^2} A_l(q, q'), \; \mathop{\mathbb{E}}_{q \sim \frac{1}{d} \chi_d^2} A_l(q, q) \right],$$

*where $A_l$ is as in Eq. (18). Furthermore,*

$$\lim_{d \to \infty} d^l \mathop{\mathbb{E}}_{q, q' \sim \frac{1}{d} \chi_d^2} A_l(q, q') = \lim_{d \to \infty} d^l \mathop{\mathbb{E}}_{q \sim \frac{1}{d} \chi_d^2} A_l(q, q) = \Phi^{(l)}(0, 1, 1).$$

*Here $\Phi^{(l)}$ is the lth derivative of $\Phi(t, q, q')$ against $t$.*

*Proof.* By Lem H.20 below, $\Phi$ is reasonable. Let $a_l = \mathbb{E}_{q \sim \frac{1}{d} \chi_d^2} A_l(q, q)$ as in Eq. (19), and let $b_l \stackrel{\text{def}}{=} \mathbb{E}_{q' \sim \frac{1}{d} \chi_d^2} A_l(q, q')$. Let $a_{li}$ be the $i$th largest eigenvalue of $A_l$ as in Eq. (18), with $a_{l0}$ being the largest. Note that all of these quantities $a_l, b_l, a_{li}$ depend on $d$, but we suppress this notationally. We seek to prove the following claims:

1. $a_{l0} \leq a_l$
2. $a_{l0} \geq b_l$
3. $d^l b_l \to \Phi^{(l)}(0, 1, 1)$
4. $d^l a_l \to \Phi^{(l)}(0, 1, 1)$

*Claim 1: $a_{l0} \leq a_l$.* First note that $a_l$ is the trace of the operator $A_l$, so that $a_l = \sum_{i=0}^{\infty} a_{li}$. Thus, any eigenvalue $a_{li}$ is at most $a_l$.

*Claim 2: $a_{l0} \geq b_l$.* Now, by Min-Max theorem, the largest eigenvalue $a_{l0}$ of $A_l$ is equal to

$$a_{l0} = \sup_f \frac{\mathbb{E}_{q, q'} f(q) f(q') A_l(q, q')}{\mathbb{E}_q f(q)^2}$$

where $0 \neq f \in L^2(\frac{1}{d} \chi_d^2)$ and $q, q' \sim \frac{1}{d} \chi_d^2$. If we set $f(q) = 1$ identically, then we get

$$a_{l0} \geq \mathop{\mathbb{E}}_{q, q'} A_l(q, q') = b_l,$$

as desired.

*Claims 3 and 4.* Now note that we have the following equalities in the Hilbert space $L^2((1 - t^2)^{\frac{d-1}{2} - 1})$, because of the absolute convergence in Eq. (17):

$$\hat{\Phi}(t) \stackrel{\text{def}}{=} \mathop{\mathbb{E}}_{q \sim \frac{1}{d} \chi_d^2} \Phi(t, q, q) = \sum_{\ell=0}^{\infty} \mathop{\mathbb{E}}_{q \sim \frac{1}{d} \chi_d^2} A_\ell(q, q) c_{d,\ell}^{-1} C_\ell^{(\frac{d-2}{2})}(t) = \sum_{\ell=0}^{\infty} a_\ell c_{d,\ell}^{-1} C_\ell^{(\frac{d-2}{2})}(t)$$

$$\check{\Phi}(t) \stackrel{\text{def}}{=} \mathop{\mathbb{E}}_{q, q' \sim \frac{1}{d} \chi_d^2} \Phi(t, q, q') = \sum_{\ell=0}^{\infty} \mathop{\mathbb{E}}_{q, q' \sim \frac{1}{d} \chi_d^2} A_\ell(q, q') c_{d,\ell}^{-1} C_\ell^{(\frac{d-2}{2})}(t) = \sum_{\ell=0}^{\infty} b_\ell c_{d,\ell}^{-1} C_\ell^{(\frac{d-2}{2})}(t).$$

Thus, by Eq. (15),

$$a_l = \Gamma\left(\frac{d}{2}\right) \sum_{k=0}^{\infty} \frac{\hat{\Phi}^{(l+2k)}(0)}{2^{l+2k} k! \Gamma\left(\frac{d}{2} + l + k\right)}.$$

By the absolute convergence of Eq. (21), differentiation commutes with expectation:

$$\hat{\Phi}^{(l+2k)}(0) = \partial_t^{l+2k} \mathop{\mathbb{E}}_{q \sim \frac{1}{d} \chi_d^2} \Phi(0, q, q) = \mathop{\mathbb{E}}_{q \sim \frac{1}{d} \chi_d^2} B_{l+2k}(q, q) = \mathop{\mathbb{E}}_{q \sim \frac{1}{d} \chi_d^2} \Phi^{(l+2k)}(0, q, q),$$

where $B_{l+2k}$ is as in Eq. (21). Furthermore, as $d \to \infty$, because $\Phi^{(l+2k)}(t, q, q')$ is smooth and polynomially bounded in $q$ and $q'$, while $\frac{1}{d}\chi_d^2$ has exponential tails and converges a.s. to 1, we have

$$\underset{q \sim \frac{1}{d}\chi_d^2}{\mathbb{E}} \Phi^{(l+2k)}(0, q, q) \to \Phi^{(l+2k)}(0, 1, 1).$$

Finally, by dominated convergence, we get

$$d^l a_l = \Gamma\left(\frac{d}{2}\right) \sum_{k=0}^{\infty} \frac{d^l \hat{\Phi}^{(l+2k)}(0)}{2^{l+2k} k! \Gamma\left(\frac{d}{2} + l + k\right)} \to \Phi^{(l)}(0, 1, 1)$$

as desired.

The proof of $\lim d^l b_l = \Phi^{(l)}(0, 1, 1)$ is similar.

$\square$

**Lemma H.20.** *Let $\Phi$ be the function corresponding to the CK or NTK of an MLP with polynomially bounded activation functions. Then $\Phi$ is reasonable (Defn H.18).*

*Proof.* The Taylor expansion Eq. (21), its absolute convergence, and the smoothness of $B_l$ follows from the proof of Thm I.3. The polynomial-boundedness of $B_l$ follows trivially from the polynomial-boundedness of activation. $\square$

## H.4 IN HIGH DIMENSION, BOOLEAN CUBE $\approx$ SPHERE $\approx$ STANDARD GAUSSIAN

For the same neural kernel $K$ with $K(x, y) = \Phi(\langle x, y \rangle / \|x\|\|y\|, \|x\|^2/d, \|y\|^2/d)$, let $\hat{\Phi}_d$ be as defined in Eq. (20), and let $\Phi_{\mathcal{S}} \overset{\text{def}}{=} \Phi(t, 1, 1)$. Thus $\Phi_{\mathcal{S}}$ is the univariate $\Phi$ that we studied in Appendix H.2 in the context of the sphere.

**Empirical verification of the spectral closeness** As $d \to \infty$, $\frac{1}{d}\chi_d^2$ converges almost surely to 1, and each $A_l$ in Eq. (16) converges weakly to the simple operator that multiplies the input function by $A_l(1, 1)$. We verify in Fig. 17 that, for different erf kernels, $\hat{\Phi}_d$ approximates $\Phi_{\mathcal{S}}$ when $d$ is large, which would then imply that their spectra become identical for large $d$. Note that this is tautologically true for relu kernels with no bias $\sigma_b = 0$ because they are positive-homogeneous.

Next, we compute the eigenvalues of $K$ on the boolean cube and the sphere, as well as the eigenvalues of the kernel $\hat{K}(x, y) \overset{\text{def}}{=} \hat{\Phi}_d(\langle x, y \rangle / \|x\|\|y\|)$ on the sphere, which "summarizes" the eigenvalues of $K$ on the standard Gaussian distribution. In Fig. 18, we compare them for different dimensions $d$ up to degree 5 (there, "Gaussian" means $\hat{K}$ on the sphere). We see that by dimension 128, all eigenvalues shown are very close to each other for each data distribution.

**Theoretical verification of the spectral closeness** Thms H.13, H.19 and I.6 show that when the dimension $d$ is sufficiently large, the top eigenvalues of $K$ as an operator on $\mathbb{Z}^d$ is very close to the top eigenvalues of $K$ as an operator on $\sqrt{d}\mathcal{S}^{d-1}$ and those of $K$ as an operator on $\mathcal{N}(0, I_d)$:

**Corollary H.21.** *Let $K$ be the CK or NTK of an MLP with polynomially bounded activation function. Then $K$ can be expressed as $K(x, y) = \Phi(\langle x, y \rangle / d)$ for some smooth $\Phi : [-1, 1] \to \mathbb{R}$. Let $\mu_k$ denote the eigenvalue of $K$ as an operator on $\mathbb{Z}^d$ corresponding to the eigenspace of degree $k$ polynomials (see Thm 3.1). Let $a_k$ denote the eigenvalue of $K$ as an operator on $\sqrt{d}\mathcal{S}^{d-1}$ corresponding to the eigenspace of degree $k$ spherical harmonics (see Thm H.11). Let $a_{k0}$ denote the largest eigenvalue of $K$ as an operator on $\mathcal{N}(0, I_d)$ restricted to degree $k$ spherical harmonics (see Thm H.19).*

*Then for any fixed $k \geq 0$, for sufficiently large $d$, we have*

$$\lim_{d \to \infty} d^k \mu_k = \lim_{d \to \infty} d^k a_k = \lim_{d \to \infty} d^k a_{k0} = \Phi^{(k)}(0)$$

*where $\Phi^{(k)}(0)$ is the kth derivative of $\Phi$ at 0. If $\Phi^{(k)}(0) \neq 0$, then*

$$\lim_{d \to \infty} \mu_k / a_k = \lim_{d \to \infty} \mu_k / a_{k0} = 1.$$

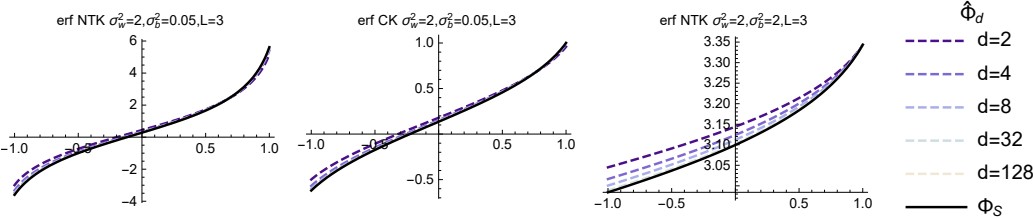

Figure 17: **In high dimension $d$, $\hat{\Phi}_d$ as defined in Eq. (20) approximates $\Phi$ very well.** We show for 3 different erf kernels that the function $\Phi$ defining $K$ as an integral operator on the sphere is well-approximated by $\hat{\Phi}_d$ when $d$ is moderately large ($d \geq 32$ seems to work well). This suggests that the eigenvalues of $K$ as an integral operator on the standard Gaussian should approximate those of $K$ as an integral operator on the sphere.

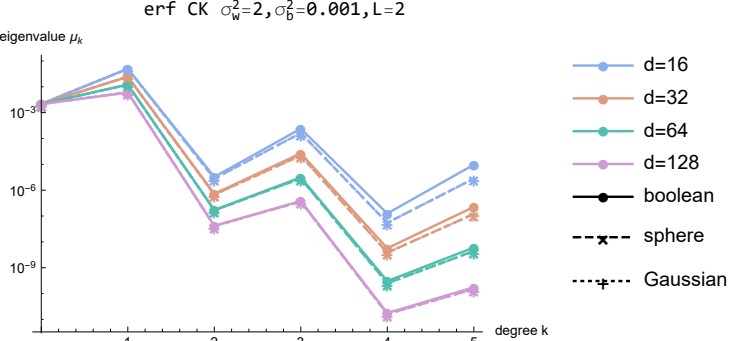

Figure 18: **In high dimension $d$, the eigenvalues are very close for the kernel over the boolean cube, the sphere, and standard Gaussian.** We plot the eigenvalues $\mu_k$ of the erf CK, with $\sigma_w^2 = 2, \sigma_b^2 = 0.001$, depth 2, over the boolean cube, the sphere, as well as kernel on the sphere induced by $\hat{\Phi}_d$ (Eq. (20)). We do so for each degree $k \leq 5$ and for dimensions $d = 16, 32, 64, 128$. We see that by dimension $d = 128$, the eigenvalues shown are already very close to each other.

*If we fix $k$ and let the input dimension $d \to \infty$, then the fractional variance of degree $k$ converges to*

$$(k!)^{-1}\Phi^{(k)}(0)/\Phi(1) = \frac{(k!)^{-1}\Phi^{(k)}(0)}{\sum_{j\geq 0}(j!)^{-1}\Phi^{(j)}(0)}$$

*for all three input distributions $\boxplus^d$, $\sqrt{d}\mathcal{S}^{d-1}$, and $\mathcal{N}(0, I_d)$.*

**Practically speaking, only the top eigenvalues matter.** Observe that, in the empirical and theoretical results above, we only verify that the top eigenvalues ($\mu_k$, $a_k$, or $a_{k0}$ for $k$ small compared to $d$) are close when $d$ is large. While this result may seem very weak at face value, in practice, the closeness of these top eigenvalues is the only thing that matters. Indeed, in machine learning, we will only ever have a finite number, say $N$, of training samples to work with. Thus, we can only use a finite $N \times N$ submatrix of the kernel $K$. This submatrix, of course, has only $N$ eigenvalues. Furthermore, if these samples are collected in an iid fashion (as is typically assumed), then these eigenvalues approximate the largest $N$ eigenvalues (top $N$ counting multiplicity) of the kernel $K$ itself (Tropp, 2015). As such, the smaller eigenvalues of $K$ can hardly be detected in the training sample, and cannot affect the machine learning process very much.

Let's discuss a more concrete example: Fig. 18 shows that the boolean cube eigenvalues $\mu_k$ are very close to the sphere eigenvalues $a_k$ for all $k \leq 5$. Over the boolean cube, $\mu_0, \dots, \mu_5$ cover eigenspaces of total dimension $\binom{d}{0} + \cdots + \binom{d}{5}$, which is 275,584,033 when $d = 128$. We need at least that many samples to be able to even detect the eigenvalue $\mu_6$ and the possible difference between it and the sphere eigenvalue $a_6$. But note in comparison, Imagenet, one of the most common large datasets in use today, has only about 15 million samples, 10 times less than the number above.

Additionally, in this same comparison, $d = 128$ dramatically pales compared to Imagenet's input dimension $3 \times 256^2 = 196608$, and even to the those of the smaller common datasets like CIFAR10 ($d = 3 \times 32^2 = 3072$) and MNIST ($d = 24^2 = 576$) — if we were to even use the input dimension of MNIST above, then $\mu_0, \dots, \mu_5$ would cover eigenspaces of 523 *billion* total dimensions! Hence, it is quite practically relevant to consider the effect of large $d$ on the eigenvalues, while keeping $k$ small. Again, we remark that even when one fixes $k$ and increases $d$, the dimension of eigenspaces affected by our limit theorems Thms H.13, H.19 and I.6 increases like $\Theta(d^k)$, which implies one needs an increasing number $\Theta(d^k)$ of training samples to see the difference of eigenvalues in higher degrees $k$.

Finally, from the perspective of fractional variance, we also can see that only the top $k$ spectral closeness matters: By Cor H.21, for any $\epsilon > 0$, there is a $k$ such that the total fractional variance of degree 0 to degree $k$ (corresponding to eigenspaces of total dimension $\Theta(d^k)$) sums up to more than $1 - \epsilon$, for the cube, the sphere, and the standard Gaussian simultaneously, when $d$ is sufficiently large. This is because the asymptotic fractional variance is completely determined by the derivatives of $\Phi$ at $t = 0$.

## I OMITTED PROOFS

**Theorem 3.1.** *On the $d$-dimensional boolean cube $\boxplus^d$, for every $S \subseteq [d]$, $\chi_S$ is an eigenfunction of $K$ with eigenvalue*

$$\mu_{|S|} \overset{\text{def}}{=} \underset{x \in \boxplus^d}{\mathbb{E}} x^S K(x, \mathbb{1}) = \underset{x \in \boxplus^d}{\mathbb{E}} x^S \Phi\left(\sum_i x_i/d\right), \tag{4}$$

*where $\mathbb{1} = (1, \dots, 1) \in \boxplus^d$. This definition of $\mu_{|S|}$ does not depend on the choice $S$, only on the cardinality of $S$. These are all of the eigenfunctions of $K$ by dimensionality considerations.*[8]

*Proof.* We directly verify $K\chi_S = \mu_{|S|}\chi_S$. Notice first that

$$K(x, y) = \Phi(\langle x, y \rangle) = \Phi(\langle x \odot y, \mathbb{1} \rangle) = K(x \odot y, \mathbb{1})$$

where $\odot$ is Hadamard product. We then calculate

$$K\chi_S(y) = \underset{x}{\mathbb{E}} K(y, x)x^S$$

$$= \underset{x}{\mathbb{E}} K(\mathbb{1}, x \odot y)(x \odot y)^S y^S.$$

---

[8]Readers familiar with boolean Fourier analysis may be reminded of the *noise operator* $T_\rho$, $\rho \leq 1$ (O'Donnell, 2014, Defn 2.46). In the language of this work, $T_\rho$ is a neural kernel with eigenvalues $\mu_k = \rho^k$.

Here we are using the fact that $x$ and $y$ are boolean to get $x^S = (x \odot y)^S y^S$. Changing variable $z \stackrel{\text{def}}{=} x \odot y$, we get

$$K\chi_S(y) = y^S \underset{z}{\mathbb{E}}\, K(\mathbb{1}, z) z^S = \mu_{|S|}\chi_S(y)$$

as desired. Finally, note that $\mu_{|S|}$ is invariant under permutation of $[d]$, so indeed it depends only on the size of $S$. $\qquad\square$

**Theorem H.11** (Schoenberg). *Suppose* $\Phi : [-1, 1] \to \mathbb{R}$ *is in* $L^2((1 - t^2)^{\frac{d-1}{2}-1})$, *so that it has the Gegenbauer expansion*

$$\Phi(t) \stackrel{a.e.}{=} \sum_{l=0}^{\infty} a_l c_{d,l}^{-1} C_l^{(\frac{d-2}{2})}(t).$$

*Then* $K$ *has eigenspaces* $\mathcal{H}_{\sqrt{d}}^{d-1,(l)} \stackrel{\text{def}}{=} \{f(x/\sqrt{d}) : f \in \mathcal{H}^{d-1,(l)}\}$ *with corresponding eigenvalues* $a_l$. *Since* $\bigoplus_{l=0}^{\infty} \mathcal{H}_{\sqrt{d}}^{d-1,(l)}$ *is an orthogonal decomposition of* $L^2(\sqrt{d}\mathcal{S}^{d-1})$, *this describes all eigenfunctions of* $K$ *considered as an operator on* $L^2(\sqrt{d}\mathcal{S}^{d-1})$.

*Proof.* Let $f \in \mathcal{H}^{d-1,(n)}$. Then for any $x \in \sqrt{d}\mathcal{S}^{d-1}$,

$$\underset{y \in \sqrt{d}\mathcal{S}^{d-1}}{\mathbb{E}}\, K(x, y)f(y/\sqrt{d}) = \underset{y \in \sqrt{d}\mathcal{S}^{d-1}}{\mathbb{E}}\, \Phi(\langle x, y\rangle/d)f(y/\sqrt{d})$$

$$= \underset{\tilde{y} \in \mathcal{S}^{d-1}}{\mathbb{E}}\, \Phi(\langle x/\sqrt{d}, \tilde{y}\rangle)f(\tilde{y})$$

$$= \underset{\tilde{y} \in \mathcal{S}^{d-1}}{\mathbb{E}}\, \left(\sum_{l=0}^{\infty} a_l \frac{1}{c_{d,l}} C_l^{(\frac{d-2}{2})}(\langle x/\sqrt{d}, \tilde{y}\rangle)\right) f(\tilde{y})$$

$$= \underset{\tilde{y} \in \mathcal{S}^{d-1}}{\mathbb{E}}\, \left(\sum_{l=0}^{\infty} a_l Z_{x/\sqrt{d}}^{d-1,(l)}(\tilde{y})\right) f(\tilde{y})$$

$$= a_n \underset{\tilde{y} \in \mathcal{S}^{d-1}}{\mathbb{E}}\, Z_{x/\sqrt{d}}^{d-1,(n)}(\tilde{y})f(\tilde{y})$$

$$\text{by orthogonality}$$

$$= a_n f(x/\sqrt{d})$$

by reproducing property. $\qquad\square$

**Lemma 3.2.** *With* $\mu_k$ *as in Thm 3.1,*

$$\mu_k = 2^{-d}(I - \mathcal{T}_\Delta)^k(I + \mathcal{T}_\Delta)^{d-k}\Phi(1) \tag{5}$$

$$= 2^{-d}\sum_{r=0}^{d} C_r^{d-k,k}\Phi\left(\left(\frac{d}{2} - r\right)\Delta\right) \tag{6}$$

*where*

$$C_r^{d-k,k} \stackrel{\text{def}}{=} \sum_{j=0}^{} (-1)^{r+j}\binom{d-k}{j}\binom{k}{r-j}. \tag{7}$$

*Proof.* Because $\sum_i x_i/d$ only takes on values $\{-\frac{d}{2}\Delta, (-\frac{d}{2} + 1)\Delta, \ldots, (\frac{d}{2} - 1)\Delta, \frac{d}{2}\Delta\}$, where $\Delta = \frac{2}{d}$, we can collect like terms in Eq. (4) and obtain

$$\mu_k = 2^{-d}\sum_{r=0}^{d} \sum_{x \text{ has } r \text{ '}-1\text{'s}} \left(\prod_{i=1}^{k} x_i\right)\Phi\left(\left(\frac{d}{2} - r\right)\Delta\right)$$

which can easily be shown to be equal to

$$\mu_k = 2^{-d}\sum_{r=0}^{d} C_r^{d-k,k}\Phi\left(\left(\frac{d}{2} - r\right)\Delta\right),$$

proving Eq. (6) in the claim. Finally, observe that $C_r^{d-k,k}$ is also the coefficient of $x^r$ in the polynomial $(1 - x)^k(1 + x)^{d-k}$. Some operator arithmetic then yields Eq. (5). $\qquad\square$

**Theorem H.14.** *A function $K : (\mathbb{R}^d)^2 \to \mathbb{R}$ of the form*

$$K(x, y) = \Phi\left(\frac{\langle x, y \rangle}{\|x\| \|y\|}, \frac{\|x\|^2}{d}, \frac{\|y\|^2}{d}\right)$$

*forms a positive semidefinite Hilbert-Schmidt operator on $L^2(\mathcal{N}(0, I))$ iff $\Phi$ can be decomposed as*

$$\Phi(t, q, q') = \sum_{l=0}^{\infty} A_l(q, q') c_{d,l}^{-1} C_l^{\left(\frac{d-2}{2}\right)}(t) \tag{16}$$

*satisfying*

$$\sum_{l=0}^{\infty} \|A_l\|^2 c_{d,l}^{-2} \|C_l^{\left(\frac{d-2}{2}\right)}\|^2 = \sum_{l=0}^{\infty} \|A_l\|^2 c_{d,l}^{-2} \frac{\pi 2^{3-d} \Gamma(l + d - 2)}{l!(l + \frac{d-2}{2}) \Gamma\left(\frac{d-2}{2}\right)^2} < \infty, \tag{17}$$

*where*

- $c_{d,l} = \frac{d-2}{d+2l-2}$

- $C_l^{\left(\frac{d-2}{2}\right)}(t)$ *are Gegenbauer polynomials as in Appendix H.2, with* $\|C_l^{\left(\frac{d-2}{2}\right)}\| = \sqrt{\int_{-1}^{1} C_l^{\left(\frac{d-2}{2}\right)}(t)^2 (1 - t^2)^{\frac{d-1}{2}-1} \, dt}$ *denoting the norm of* $C_l^{\left(\frac{d-2}{2}\right)}$ *in* $L^2((1 - t^2)^{\frac{d-1}{2}-1})$.

- *and $A_l$ are positive semidefinite Hilbert-Schmidt kernels on $L^2(\frac{1}{d}\chi_d^2)$, the $L^2$ space over the probability measure of a $\chi_d^2$-variable divided by d, and with $\|A_l\|$ denoting the Hilbert-Schmidt norm of $A_l$.*

*In addition, $K$ is positive definite iff all $A_l$ are.*

*Proof.* Note that an isotropic Gaussian vector $z \sim \mathcal{N}(0, I)$ can be sampled by independently sampling its direction $v$ uniformly from the sphere $\mathcal{S}^{d-1}$ and sampling its magnitude $r$ from a chi distribution $\chi_d$ with $d$ degrees of freedom. In the following, we adopt the following notations

$$x, y \in \mathbb{R}^d, \ q = \|x\|^2/d, \ q' = \|y\|^2/d, \ v = x/\|x\|, \ v' = y/\|y\|$$

Then, by the reasoning above,

$$Kf(x/\sqrt{d}) = \mathop{\mathbb{E}}_{\substack{v' \sim \mathcal{S}^{d-1} \\ q' \sim d^{-1}\chi_d^2}} \Phi\left(\langle v, v' \rangle, q, q'\right) f(\sqrt{q'}v).$$

Here $f \in L^2(\mathcal{N}(0, I/d))$ (so that $f(\cdot/\sqrt{d}) \in L^2(\mathcal{N}(0, I))$) and $d^{-1}\chi_d^2$ is the distribution of a $\chi_d^2$ random variable divided by $d$. Because of this decomposition of $\mathcal{N}(0, I)$ into a product distribution of direction and magnitude, its space of $L^2$ functions naturally decomposes into a tensor product of corresponding $L^2$ spaces

$$L^2(\mathcal{N}(0, I)) = L^2(\mathcal{S}^{d-1}) \otimes L^2(d^{-1}\chi_d^2).$$

where every $f \in L^2(\mathcal{N}(0, I))$ can be written as a sum

$$f(x) = \sum_{l=0}^{\infty} R_l(q) P_l(v)$$

where $q = \|x\|^2/d, v = x/\|x\|$ as above, $P_l \in \mathcal{H}^{d-1,(l)}$ is a spherical harmonic of degree $l$, and $R_l \in L^2(d^{-1}\chi_d^2)$ (i.e. $R_l(\cdot/d) \in L^2(\chi_d^2)$). Here, the equality is understood as a convergence of the RHS partial sums in the Hilbert space $L^2(\mathcal{S}^{d-1}) \otimes L^2(d^{-1}\chi_d^2)$.

Likewise, assuming $K$ is Hilbert-Schmidt, then its Hilbert-Schmidt norm (measured against the uniform distribution over the sphere) is bounded:

$$\|K\|_{HS}^2 = \mathop{\mathbb{E}}_{\substack{v,v' \sim \mathcal{S}^{d-1} \\ q,q' \sim d^{-1}\chi_d^2}} \Phi(\langle v, v' \rangle, q, q')^2$$

$$= \mathop{\mathbb{E}}_{q,q' \sim d^{-1}\chi_d^2} \int_{-1}^{1} \Phi(t, q, q')^2 (1 - t^2)^{\frac{d-1}{2}-1} \, dt.$$

Thus $\Phi$ resides in the tensor product space

$$\Phi \in \mathcal{F} \overset{\text{def}}{=} L^2((1-t^2)^{\frac{d-1}{2}-1}) \otimes L^2(d^{-1}\chi_d^2) \otimes L^2(d^{-1}\chi_d^2)$$

and therefore can be expanded as

$$\Phi(t, q, q') = \sum_{l=0}^{\infty} A_l(q, q') c_{d,l}^{-1} C_l^{(\frac{d-2}{2})}(t)$$

where $C_l^{(\frac{d-2}{2})}(t)$ are Gegenbauer polynomials as in Appendix H.2 and $A_l \in L^2(d^{-1}\chi_d^2)^{\otimes 2}$. By Lem I.1 below, $K$ being Hilbert-Schmidt implies that each $A_l$ is, as well. Furthermore, since $\|K\|_{HS}^2 = \|\Phi\|_{\mathcal{F}}^2$ is finite, we have

$$\|\Phi\|_{\mathcal{F}}^2 = \sum_{l=0}^{\infty} \|A_l\|^2 c_{d,l}^{-2} \|C_l^{(\frac{d-2}{2})}\|^2 < \infty.$$

Here, $\|A_l\| = \sqrt{\mathbb{E}_{q,q'\sim\frac{1}{d}\chi_d^2} A_l(q,q')^2}$ is the Hilbert-Schmidt norm of $A_l$ (i.e. its norm in $L^2(d^{-1}\chi_d^2)^{\otimes 2}$), and $\|C_l^{(\frac{d-2}{2})}\| = \sqrt{\int_{-1}^{1} C_l^{(\frac{d-2}{2})}(t)^2 (1-t^2)^{\frac{d-1}{2}-1} \, dt}$ is the norm of $C_l^{(\frac{d-2}{2})}$ in $L^2((1-t^2)^{\frac{d-1}{2}-1})$. Simplifying according to Eq. (14) yields the equality in the claim.

Conversely, if each $A_l$ is Hilbert-Schmidt satisfying Eq. (17), then $K$ is obviously a Hilbert-Schmidt kernel as it has finite Hilbert-Schmidt norm.

$\square$

**Lemma I.1.** *For each $l$, $A_l$ is a positive semidefinite Hilbert-Schmidt kernel. It is positive definite if $K$ is.*

*Proof.* With $q = \|x\|^2/d, v = x/\|x\|$ as above, let $f(x/\sqrt{d}) = f(\sqrt{q}v) = R_m(q)P_m(v)$ for some degree $m$ nonzero spherical harmonics $P_m \in \mathcal{H}^{d-1,(m)}$ and some scalar function $R_m \in L^2(d^{-1}\chi_d^2)$ that is not a.e. zero. We have

$$K\phi(x/\sqrt{d}) = \underset{\substack{v'\sim\mathcal{S}^{d-1} \\ q'\sim d^{-1}\chi_d^2}}{\mathbb{E}} \Phi(\langle v, v'\rangle, q, q') f(\sqrt{q'}v')$$

$$= \underset{\substack{v'\sim\mathcal{S}^{d-1} \\ q'\sim d^{-1}\chi_d^2}}{\mathbb{E}} R_m(q')P_m(v') \sum_{l=0}^{\infty} A_l(q, q') c_{d,l}^{-1} C_l^{(\frac{d-2}{2})}(\langle v, v'\rangle)$$

$$= \underset{\substack{v'\sim\mathcal{S}^{d-1} \\ q'\sim d^{-1}\chi_d^2}}{\mathbb{E}} R_m(q')P_m(v') \sum_{l=0}^{\infty} A_l(q, q') Z_v^{d-1,(l)}(v')$$

$$= P_m(v) \underset{q'\sim d^{-1}\chi_d^2}{\mathbb{E}} A_m(q, q') R_m(q').$$

Therefore,

$$\underset{x\sim\mathcal{N}(0,I)}{\mathbb{E}} f(x)Kf(x/\sqrt{d}) = \left( \underset{v\sim\mathcal{S}^{d-1}}{\mathbb{E}} P_m(v)^2 \right) \left( \underset{q,q'\sim d^{-1}\chi_d^2}{\mathbb{E}} R_m(q) A_m(q, q') R_m(q') \right)$$

is nonnegative, and is positive if $K$ is positive definite, by the assumption above that $f$ is not a.e. zero. Since $\mathbb{E} P_m(v)^2 > 0$ as $P_m \neq 0$, we must have

$$\underset{q,q'\sim d^{-1}\chi_d^2}{\mathbb{E}} R_m(q) A_m(q, q') R_m(q') \geq 0 \text{ (or } > 0 \text{ if } K \text{ is positive definite).}$$

This argument holds for any $R_m \in L^2(d^{-1}\chi_d^2)$ that is not a.e. zero, so this implies that $A_l$ is positive semidefinite, and positive definite if $K$ is so. $\square$

**Weak Spectral Simplicity Bias**

**Theorem 4.1** (Weak Spectral Simplicity Bias). *Let $K$ be the CK or NTK of an MLP on a boolean cube $\boxdot^d$. Then the eigenvalues $\mu_k, k = 0, \ldots, d$, satisfy*

$$\mu_0 \geq \mu_2 \geq \cdots \geq \mu_{2k} \geq \cdots, \quad \mu_1 \geq \mu_3 \geq \cdots \geq \mu_{2k+1} \geq \cdots . \tag{9}$$

*Proof.* Again, there is a function $\Phi$ such that $K(x,y) = \Phi(\langle x, y\rangle/\|x\|\|y\|)$. By Thm I.3, $\Phi$ has a Taylor expansion with only nonnegative coefficients.

$$\Phi(c) = a_0 + a_1 c + a_2 c^2 + \cdots .$$

By Lem I.2, Eq. (9) is true for polynomials $\Phi(c) = c^r$,

$$\mu_0(c^r) \geq \mu_2(c^r) \geq \cdots, \quad \mu_1(c^r) \geq \mu_3(c^r) \geq \cdots . \tag{22}$$

Then since each $\mu_k = \mu_k(\Phi)$ is a linear function of $\Phi$, $\mu_k(\Phi) = \sum_{r=0}^{\infty} a_r \mu_k(c^r)$ also follows the same ordering. $\square$

**Lemma I.2.** *Let $\mathcal{T}_\Delta$ be the shift operator with step $\Delta$ that sends a function $\Phi(\cdot)$ to $\Phi(\cdot - \Delta)$. Let $\Phi(c) = c^t$ for some $t$. Let $\mu_k^d$ be the eigenvalue of $K(x,y) = \Phi(\langle x, y\rangle/\|x\|\|y\|)$ on the boolean cube $\boxdot^d$. Then for any $0 \leq k \leq d$,*

$$\mu_k^d = 0 \qquad\qquad\qquad \text{if } t + k \text{ is odd} \tag{23}$$
$$1 \geq \mu_{k-2}^d \geq \mu_k^d \geq 0 \qquad\qquad \text{if } t + k \text{ is even.} \tag{24}$$

*We furthermore have the identity*

$$\mu_{k-2}^d - \mu_k^d = \left(\frac{d-2}{d}\right)^t \mu_{k-2}^{d-2}$$

*for any $2 \leq k \leq d$.*

*Proof.* As in Eq. (5),

$$2^{-d}(I - \mathcal{T}_\Delta)^k (I + \mathcal{T}_\Delta)^{d-k}\Phi(1) = 2^{-d}\sum_{r=0}^d C_r^{d-k,k}\Phi\left(\left(\frac{d}{2} - r\right)\Delta\right)$$

where $C_r^{d-k,k} \overset{\text{def}}{=} \sum_{j=0}(-1)^{r+j}\binom{d-k}{j}\binom{k}{r-j}$. It's easy to observe that

$$C_r^{d-k,k} = C_{d-r}^{d-k,k} \quad \text{if } k \text{ is even}$$
$$C_r^{d-k,k} = -C_{d-r}^{d-k,k} \quad \text{if } k \text{ is odd.}$$

In the first case, when $\Phi(c) = c^t$ with $t$ odd, then by symmetry $\mu_k = 0$. Similarly for the second case when $t$ is even. This finishes the proof for Eq. (23).

Note that it's clear from the form of Eq. (5) that $\mu_k^d \leq \Phi(1) = 1$ always. So we show via induction on $d$ that the rest of Eq. (24) is true for any $t \in \mathbb{N}$. The induction will reduce the case of $d$ to the case of $d - 2$. So for the base case, we need both $d = 0$ and $d = 1$. Both can be shown by some simple calculations.

Now for the inductive step, assume Eq. (24) for $d - 2$, and we observe that

$$\mu_{k-2}^d - \mu_k^d = 2^{-d}(I - \mathcal{T}_{2/d})^{k-2}(I + \mathcal{T}_{2/d})^{d-k}((I + \mathcal{T}_{2/d})^2 - (I - \mathcal{T}_{2/d})^2)\Phi(1)$$
$$= 2^{-d}(I - \mathcal{T}_{2/d})^{k-2}(I + \mathcal{T}_{2/d})^{d-k}(4\mathcal{T}_{2/d})\Phi(1)$$
$$= 2^{-d+2}(I - \mathcal{T}_{2/d})^{k-2}(I + \mathcal{T}_{2/d})^{d-k}\Phi(1 - 2/d).$$

By Eq. (5), we can expand this into

$$2^{-d+2} \sum_{r=0}^{d-2} C_r^{d-k,k-2} \Phi \left( 1 - \frac{2}{d} - r\frac{2}{d} \right)$$

$$= 2^{-d+2} \sum_{r=0}^{d-2} C_r^{d-k,k-2} \left( 1 - \frac{2}{d} - r\frac{2}{d} \right)^t$$

$$= \left( \frac{d-2}{d} \right)^t 2^{-d+2} \sum_{r=0}^{d-2} C_r^{d-k,k-2} \left( 1 - r\frac{2}{d-2} \right)^t$$

$$= \left( \frac{d-2}{d} \right)^t \mu_{k-2}^{d-2}.$$

By induction, $\mu_{k-2}^{d-2} \geq 0$, so we have

$$\mu_{k-2}^d - \mu_k^d = \left( \frac{d-2}{d} \right)^t \mu_{k-2}^{d-2} \geq 0$$

as desired.

$\square$

**Theorem I.3.** *Let $K$ be the CK or NTK of an MLP with domain $\sqrt{d}\mathcal{S}^{d-1} \subseteq \mathbb{R}^d$. Then $K(x,y) = \Phi \left( \frac{\langle x,y \rangle}{\|x\|\|y\|} \right)$ where $\Phi : [-1,1] \to \mathbb{R}$ has a Taylor series expansion around 0*

$$\Phi(c) = a_0 + a_1 c + a_2 c^2 + \cdots \tag{25}$$

*with the properties that $a_i \geq 0$ for all $i$ and $\sum_i a_i = \Phi(1) \leq \infty$, so that Eq. (25) is absolutely convergent on $c \in [-1,1]$.*

*Proof.* We first prove this statement for the CK of an $L$ layer MLP with nonlinearity $\phi$. Let $\Phi^l$ be the corresponding $\Phi$-function for the CK $\Sigma^l$. It suffices to prove this by induction on depth $L$. For $L = 1$, $\Sigma^1(x,x') = \sigma_w^2 (n^0)^{-1} \langle x,x' \rangle + \sigma_b^2$ by Eq. (CK), so clearly $\Phi^1(c) = a_0 + a_1 c$ where $a_0$ and $a_1$ are nonnegative.

Now for the inductive step, suppose $\Sigma^{l-1}$ satisfies the property that $\Phi^{l-1}$ has Taylor expansion of the desired form. We seek to show the same for $\Sigma^l$ and $\Phi^l$. First notice that it suffices to show this for $V_\phi(\Sigma^{l-1})$, since multiplication by $\sigma_w^2$ and addition of $\sigma_b^2$ preserves the property. But

$$V_\phi(\Sigma^{l-1})(x,x') = \mathbb{E}\,\phi(z)\phi(z')$$

where

$$(z,z') \sim \mathcal{N} \left( 0, \begin{pmatrix} \Sigma^{l-1}(x,x) & \Sigma^{l-1}(x,x') \\ \Sigma^{l-1}(x,x') & \Sigma^{l-1}(x',x') \end{pmatrix} \right) = \mathcal{N} \left( 0, \begin{pmatrix} \Phi^{l-1}(1) & \Phi^{l-1}(c) \\ \Phi^{l-1}(c) & \Phi^{l-1}(1) \end{pmatrix} \right).$$

Using the notation of Lem I.4, we can express this as

$$V_\phi(\Sigma^{l-1})(x,x') = \Phi^l(1)\hat{\phi} \left( \frac{\Phi^{l-1}(c)}{\Phi^{l-1}(1)} \right). \tag{26}$$

Substituting the Taylor expansion of $\frac{\Phi^{l-1}(c)}{\Phi^{l-1}(1)}$ into the Taylor expansion of $\hat{\phi}$ given by Lem I.4 gives us a Taylor series whose coefficients are all nonnegative, since those of $\frac{\Phi^{l-1}(c)}{\Phi^{l-1}(1)}$ and $\hat{\phi}$ are nonnegative as well. In addition, plugging in 1 for $c$ in this Taylor series shows that the sum of the coefficients equal $\hat{\phi}(1) = 1$, so the series is absolutely convergent for $c \in [-1,1]$. This proves the inductive step.

For the NTK, the proof is similar, except we now also have a product step where we multiply $V_{\phi'}(\Sigma^{l-1})$ with $\Theta^{l-1}$, and we simply just need to use the fact that product of two Taylor series with nonnegative coefficients is another Taylor series with nonnegative coefficients, and that the resulting radius of convergence is the minimum of the original two radii of convergence.

$\square$

**Lemma I.4.** *Consider any $\sigma > 0$ and any $\phi : \mathbb{R} \to \mathbb{R}$ square-integrable against $\mathcal{N}(0, \sigma^2)$. Then the function*

$$\hat{\phi} : [-1, 1] \to [-1, 1], \quad \hat{\phi}(c) \stackrel{\text{def}}{=} \frac{\mathbb{E}\,\phi(x)\phi(y)}{\mathbb{E}\,\phi(x)^2}, \quad \text{where } (x, y) \sim \mathcal{N}\left(0, \begin{pmatrix} \sigma^2 & \sigma^2 c \\ \sigma^2 c & \sigma^2 \end{pmatrix}\right),$$

*has a Taylor series expansion around 0*

$$\hat{\phi}(c) = a_0 + a_1 c + a_2 c^2 + \cdots \tag{27}$$

*with the properties that $a_i \geq 0$ for all $i$ and $\sum_i a_i = \hat{\phi}(1) = 1$, so that Eq. (27) is absolutely convergent on $c \in [-1, 1]$.*

*Proof.* Since the denominator of $\hat{\phi}$ doesn't depend on $c$, it suffices to prove this Taylor expansion exists for its numerator. Let $\tilde{\phi}(c) \stackrel{\text{def}}{=} \phi(\sigma c)$. Then

$$\mathbb{E}\left[\phi(x)\phi(y) : (x, y) \sim \mathcal{N}\left(0, \begin{pmatrix} \sigma^2 & \sigma^2 c \\ \sigma^2 c & \sigma^2 \end{pmatrix}\right)\right] = \mathbb{E}\left[\tilde{\phi}(x)\tilde{\phi}(y) : (x, y) \sim \mathcal{N}\left(0, \begin{pmatrix} 1 & c \\ c & 1 \end{pmatrix}\right)\right]$$

By Fact I.5, this is just

$$b_0^2 + b_1^2 c + b_2^2 c^2 + \cdots$$

where $b_i$ are the Hermite coefficients of $\tilde{\phi}$. Since $\phi \in L^2(\mathcal{N}(0, \sigma^2))$,

$$\sum_i b_i^2 < \infty$$

as desired. $\qquad\square$

**Fact I.5** (O'Donnell (2014)). *For any $\phi, \psi \in L^2(\mathcal{N}(0, 1))$,*

$$\mathbb{E}\left[\phi(x)\psi(y) : (x, y) \sim \mathcal{N}\left(0, \begin{pmatrix} 1 & c \\ c & 1 \end{pmatrix}\right)\right] = a_0 b_0 + a_1 b_1 c + a_2 b_2 c^2 + \cdots$$

*where $a_i$ and $b_i$ are respectively the Hermite coefficients of $\phi$ and $\psi$.*

**Fixing $k$, taking the $d \to \infty$ limit** In this subsection, we prove that $\mu_k \sim d^{-k}$ as $d \to \infty$ with $k$ fixed. More precisely,

**Theorem I.6.** *Let $K$ be the CK or NTK of an MLP on a boolean cube $\boxdot^d$. Then $K$ can be expressed as $K(x, y) = \Phi(\langle x, y \rangle / d)$ for some $\Phi : [-1, 1] \to \mathbb{R}$. If we fix $k$ and let $d \to \infty$, then*

$$\lim_{d \to \infty} d^k \mu_k = \Phi^{(k)}(0),$$

*where $\Phi^{(k)}$ denotes the $k$th derivative of $\Phi$.*

An immediate consequence, using the fact that the degree $k$ eigenspace has dimension $\binom{d}{k} \sim d^k/k!$ for large $d$, is

**Theorem 5.1** (Asymptotic Fractional Variance). *Let $K$ be the CK or NTK of an MLP on a boolean cube $\boxdot^d$. Then $K$ can be expressed as $K(x, y) = \Phi(\langle x, y \rangle / d)$ for some analytic function $\Phi : \mathbb{R} \to \mathbb{R}$. If we fix $k$ and let the input dimension $d \to \infty$, then the fractional variance of degree $k$ converges to*

$$(k!)^{-1}\Phi^{(k)}(0)/\Phi(1) = \frac{(k!)^{-1}\Phi^{(k)}(0)}{\sum_{j \geq 0}(j!)^{-1}\Phi^{(j)}(0)}$$

*where $\Phi^{(k)}$ denotes the $k$th derivative of $\Phi$.*

Let's first give some intuition for why Thm I.6 should be true. By Eqs. (8) and (5),

$$\mu_k = \left(\frac{1 + \mathcal{T}_\Delta}{2}\right)^{d-k}\left(\frac{1 - \mathcal{T}_\Delta}{2}\right)^k \Phi(1)$$

$$= \frac{1}{2^d}\sum_{r=0}^{d-k}\binom{d-k}{r}\Phi^{[k]}(1 - r\Delta)$$

where $\Phi^{[k]} = (1 - \mathcal{T}_\Delta)^k \Phi$ is the $k$th backward finite difference with step $\Delta = 2/d$. Approximating the finite difference with derivative, we thus would expect that, as suggested by Lem I.8,

$$\Phi^{[k]}(x) \approx \Delta^k \Phi^{(k)}(x) = (2/d)^k \Phi^{(k)}(x), \quad \text{when } d \text{ is large,}$$

where $\Phi^{(k)}$ is the $k$th derivative of $\Phi$. Since $\frac{1}{2^{d-k}}\binom{d-k}{r}$ is the probability mass at $1 - r\Delta$ of the binomial variable $B \overset{\text{def}}{=} \frac{1}{d-k}\sum_{i=1}^{d-k} X_i$, where $X_i$ is the Bernoulli variable taking $\pm 1$ value with half chance each. This random variable converges almost surely to the delta distribution at 0, by law of large numbers. Therefore, we would expect

$$\mu_k \sim (1/d)^k \Phi^{(k)}(0)$$

as $d \to \infty$.

There is a single difficulty when trying to formalize this intuition. One is the possibility that the $k$th derivative $\Phi^{(k)}$ does not exist at the endpoints 1 and $-1$ (note that it must exist in the interior, because by Thm I.3, $\Phi$ is analytic on $(-1, 1)$). In this case, we must show that the portion of the interval $[-1, 1]$ near the endpoints contributes exponentially little mass to the probability distribution of the binomial variable $B$, that it suppresses any blowup that can happen at the edge.

We formalize this reasoning in the proof below.

*Proof. We first prevent any blowup that may happen at the edge.* Let $\Phi^{[k]}$ be the $k$th backward difference of $\Phi$ with step size $\Delta = 2/d$, as in Lem I.8. First note the trivial bound

$$\Phi^{[k]}(x) = \sum_{s=0}^{k}(-1)^s\binom{k}{s}\Phi(x - s\Delta) \leq \sum_{s=0}^{k}\binom{k}{s}\Phi(x - s\Delta)$$
$$\leq 2^k\Phi(x) \leq 2^k\Phi(1)$$

since $\Phi$ is nondecreasing by Lem I.7. Then for any $R \in [0, (d-k)/2 - 1]$,

$$d^k\mu_k = \frac{d^k}{2^d}\sum_{r=0}^{d-k}\binom{d-k}{r}\Phi^{[k]}(1 - r\Delta)$$
$$= \frac{d^k}{2^d}\sum_{r=R+1}^{d-k-R-1}\binom{d-k}{r}\Phi^{[k]}(1 - r\Delta) + \frac{d^k}{2^d}\left(\sum_{r=0}^{R} + \sum_{r=d-k-R}^{d-k}\right)\binom{d-k}{r}\Phi^{[k]}(1 - r\Delta).$$

Now, in the second term, $\Phi^{[k]}(1 - r\Delta) \leq 2^k\Phi(1)$ as above, so that, by the binomial coefficient entropy bound (Fact I.9),

$$0 \leq d^k\mu_k - \frac{d^k}{2^d}\sum_{r=R+1}^{d-k-R-1}\binom{d-k}{r}\Phi^{[k]}(1 - r\Delta) \leq \frac{d^k}{2^d}\left(\sum_{r=0}^{R} + \sum_{r=d-k-R}^{d-k}\right)\binom{d-k}{r}2^k\Phi(1)$$
$$\leq \frac{d^k}{2^{d-k}}2 \cdot 2^{H(p)(d-k)}\Phi(1)$$
$$= \frac{d^k\Phi(1)}{2^{(1-H(p))d-(1-H(p))k-1}}, \qquad (28)$$

where $p = R/(d-k)$. If we choose $R = \lfloor \tilde{p}(d-k) \rfloor$ for a fixed $\tilde{p} < 1/10$, then $H(p) < 1$, and (28) is decreasing exponentially fast to 0 with $d$.

*Now we formalize the law of large number intuition.* It then suffices to show that

$$\frac{d^k}{2^d}\sum_{r=R+1}^{d-k-R-1}\binom{d-k}{r}\Phi^{[k]}(1 - r\Delta) \to \Phi^{(k)}(0).$$

We will do so by presenting an upper and a lower bound which both converge to this limit.

We first discuss the upper bound. By Lem I.8,

$$\frac{d^k}{2^d}\sum_{r=R+1}^{d-k-R-1}\binom{d-k}{r}\Phi^{[k]}(1 - r\Delta) \leq \frac{1}{2^{d-k}}\sum_{r=R+1}^{d-k-R-1}\binom{d-k}{r}\Phi^{(k)}(1 - r\Delta).$$

The RHS can be upper bounded by an expectation over a binomial random variable: Let $X_i$ be $\pm 1$ Bernoulli variables, and let $\Phi_{\frac{1}{2}\tilde{p}}^{(k)}$ be the function

$$\Phi_{\frac{1}{2}\tilde{p}}^{(k)}(x) = \begin{cases} \Phi^{(k)}(x) & \text{if } x \in [-1 + \frac{1}{2}\tilde{p}, 1 - \frac{1}{2}\tilde{p}] \\ 0 & \text{otherwise} \end{cases}$$

Note that $\Phi_{\frac{1}{2}\tilde{p}}^{(k)}(x)$ is a bounded function. Then with the binomial variable $B = \frac{1}{d}\sum_{i=1}^{d-k}(X_i + k/(d-k))$, we have

$$\frac{1}{2^{d-k}} \sum_{r=R+1}^{d-k-R-1} \binom{d-k}{r} \Phi^{(k)}(1 - r\Delta) \leq \mathbb{E}\,\Phi_{\frac{1}{2}\tilde{p}}^{(k)}(B).$$

By strong law of large numbers, $B$ converges almost surely to $0$ as $d \to \infty$ with $k$ fixed. Thus the RHS converges to

$$\Phi_{\frac{1}{2}\tilde{p}}^{(k)}(0) = \Phi^{(k)}(0)$$

as desired.

A similar argument proceeds for the lower bound, after noting that, by Lem I.8,

$$\frac{1}{2^{d-k}} \sum_{r=R+1}^{d-k-R-1} \binom{d-k}{r} \Phi^{(k)}(1 - (r+k)\Delta) \leq \frac{d^k}{2^d} \sum_{r=R+1}^{d-k-R-1} \binom{d-k}{r} \Phi^{[k]}(1 - r\Delta).$$

$\square$

**Lemma I.7.** *Let $K$ be the CK or NTK of an MLP with domain $\sqrt{d}S^{d-1} \subseteq \mathbb{R}^d$. Then $K(x,y) = \Phi\left(\frac{\langle x,y \rangle}{\|x\|\|y\|}\right)$ where $\Phi : [-1,1] \to \mathbb{R}$ is analytic on $(-1,1)$, and all derivatives of $\Phi$ are nonnegative on $(-1,1)$.*

*Proof.* Note that, by Thm I.3, $\Phi$ has a convergent Taylor expansion on $(-1,1)$, i.e $\Phi$ *is analytic on* $(-1,1)$. Thus, all derivatives of $\Phi$ exist (and are finite) on the interval $(-1,1)$ and have the obvious Taylor series derived from Eq. (25). For example,

$$\Phi'(c) = a_1 c + 2a_2 c + 3a_3 c^2 + \cdots.$$

Such Taylor series also converge on the open interval $(-1,1)$ (but could diverge on the endpoints $-1$ and $1$). We get the desired result by noting that all $a_i$ are nonnegative. $\square$

By expressing finite difference as an integral, it's easy to see that

**Lemma I.8.** *With the same setting as in Lem I.7, let $\Phi^{[k]}$ be the kth backward finite difference with step size $\Delta = 2/d$,*

$$\Phi^{[k]}(x) \stackrel{\text{def}}{=} (1 - \mathcal{T}_\Delta)^k \Phi(x).$$

*Then $\Phi^{[k]}$ is analytic on $(-1,1)$, has all derivatives nonnegative, and*

$$0 \leq \Delta^k \Phi^{(k)}(x - k\Delta) \leq \Phi^{[k]}(x) \leq \Delta^k \Phi^{(k)}(x), \tag{29}$$

*where $\Phi^{(k)}$ is the kth derivative of $\Phi$.*

*Proof.* Everything follows immediately from

$$\Phi^{[k]}(x) = \int_0^\Delta \cdots \int_0^\Delta \Phi^{(k)}(x - h_1 - \cdots - h_k)\,\mathrm{d}h_1 \cdots \mathrm{d}h_k$$

and Lem I.7. $\square$

**Fact I.9** (Entropy bound on sum of binomial coefficients). *For any $k \leq d$ and $R \leq d/2$,*

$$\sum_{r=0}^{R} \binom{d-k}{r} \leq 2^{H(p)(d-k)}$$

*where $p = R/(d-k)$, and $H$ is the binary entropy*

$$H(p) = -p \log_2 p - (1-p) \log_2(1-p).$$

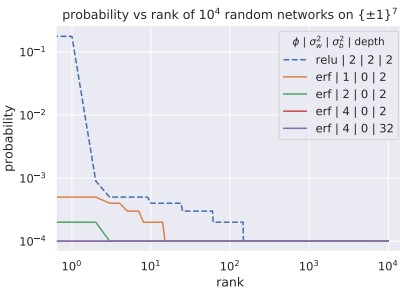

Figure 19: **The same experiments as Fig. 1 but over** $\{0,1\}^7$**.**

## J  THE $\{0,1\}^d$ VS THE $\{\pm 1\}^d$ BOOLEAN CUBE

Valle-Pérez et al. (2018) actually did their experiments on the $\{0,1\}^d$ boolean cube, whereas here, we have focused on the $\{\pm 1\}^d$ boolean cube. As datasets are typically centered before feeding into a neural network (for example, using Pytorch's `torchvision.transform.Normalize`), $\{\pm 1\}^d$ is much more natural. In comparison, using the $\{0,1\}^d$ cube is equivalent to adding a bias in the input of a network and reducing the weight variance in the input layer, since any $x \in \{\pm 1\}^d$ corresponds to $\frac{1}{2}(x+1) \in \{0,1\}^d$. As such, one would expect there is *more bias toward low frequency components with inputs from* $\{0,1\}^d$.

Nevertheless, here we verify that our observations of Section 4 above still holds over the $\{0,1\}^d$ cube by repeating the same experiments as Fig. 1 in this setting (Fig. 19). Just like over the $\{\pm 1\}^d$ cube, the relu network biases significantly toward certain functions, but with erf, and with increasing $\sigma_w^2$, this lessens. With depth 32 and $\sigma_w^2$, the boolean functions obtained from erf network see no bias at all.

