# OpenReview forum: "A Fine-Grained Spectral Perspective on Neural Networks"
_ICLR.cc/2020/Conference — Reject_

### Official Review · AnonReviewer2 · 2019-10-22
**Official Blind Review #2**

**Rating:** 6

**Review:**

Updates:

Thanks for the updates.

I find the new theoretical results interesting and potentially useful,  which shows, in the large $d$ setting, spectrums of CKs/NTKs  for boolean cube, sphere and isotropic Gaussian are closed to each other in some sense. Thus, I raise my score to weakly accepted but lower down my confidence level since I am not that familiar with Boolean cube literature.


------------------------------------------------------
The study of extremely over-parameterized networks (i.e. infinitely width networks) has become one of the most active research directions in theory deep learning. The key objects in understanding such networks are the conjugate kernel [1, 2] (CK defined in the paper) and the Neural tangent kernels [3] (NTK). The CK characterizes how the network looks like at initialization (connection to Gaussian processes as well) and the NTK is very useful to characterize the gradient descent training dynamics of large width networks in the kernel regime. Understanding properties of such kernels, in particular, their spectra distribution and eigenspace, could be potentially an important step towards a finer-gained understanding of generalization in neural networks.

The main contribution of this paper is the development of the spectral theory of CK and NTK on boolean cube (similar or weaker results on uniform distribution in spheres and Gaussian distribution in R^n). More precisely, the authors show that, over the space of boolean cube, the CK/NTK could be diagonalized using the Fourier basis and the eigenvalues depend only on the frequency (i.e. the degree of the monomials); Thm 3.1. The authors also develop some computation tools to compute the spectra; Lemma 3.2.   Using the tools developed in this paper, the authors are able to clarify some of the interesting observations found by other researchers. Most noticeably, the authors show that the observation in [4] 'neural network is biased towards simple functions' is NOT universal. Whether this statement is correct or not depends heavily on the choice of activation function (e.g. Relu v.s. Erf) and hyper-parameters (e.g. weight variance, depths).  There are also some other interesting empirical findings: the optimal depth of a neural network depends on the complexity (i.e. degree in the boolean cube setting) of the function to learn, CK (i.e. training only the last layer) tends to be more useful for learning less complex functions, etc.

Overall, this is a nice paper. I am leaning for a weakly accept.


[1] Amit Daniely, Roy Frostig, and Yoram Singer. Toward Deeper Understanding of Neural Networks:
The Power of Initialization and a Dual View on Expressivity. arXiv:1602.05897 [cs, stat], February
2016.
[2] Jaehoon Lee, Yasaman Bahri, Roman Novak, Sam Schoenholz, Jeffrey Pennington, and Jascha
Sohl-dickstein. Deep Neural Networks as Gaussian Processes. In International Conference on
Learning Representations, 2018.
[3] Arthur Jacot, Franck Gabriel, and Clément Hongler. Neural Tangent Kernel: Convergence and
Generalization in Neural Networks. arXiv:1806.07572 [cs, math, stat], June 2018. 00000
[4] Guillermo Valle-Pérez, Chico Q. Camargo, and Ard A. Louis. Deep learning generalizes because
the parameter-function map is biased towards simple functions. arXiv:1805.08522 [cs, stat], May
2018.


**Experience Assessment:**

I have read many papers in this area.

**Review Assessment: Checking Correctness Of Derivations And Theory:**

I did not assess the derivations or theory.

**Review Assessment: Checking Correctness Of Experiments:**

I did not assess the experiments.

**Review Assessment: Thoroughness In Paper Reading:**

I read the paper at least twice and used my best judgement in assessing the paper.

---

> ### Author Response · Authors · 2019-11-07
> **Rebuttal for review #2 (1)**
>
> Thank you Reviewer #2 for taking the time out to review this paper.
>
> 1) Rejecting a paper because it uses a novel technique seems wrong. Research is all about finding new bridges between different areas. We hope that the reviewer can see our paper in this light, as it connects the extremely well-studied area of boolean analysis, with the modern and exciting world of neural networks. It is true that few papers have done that before, which in our opinion should be a strong point of the paper rather than an argument for rejection.
>
> 2) Could you clarify what "important open problems" do you wish to see solved?

---

> ### Author Response · Authors · 2019-11-15
> **Rebuttal for review #2 (2)**
>
> To add on to our response to 2), we verify our main claims hold over the sphere and the standard Gaussian along with realistic datasets like MNIST and CIFAR10 via new experiments and theorems
>
> * New experimental results (discussed in detail in the new Section B)
> *** Our theory tells us that for each target function, there should be an optimal depth: networks that are deeper or shallower do monotonically worse (this is discussed in Section 5). In the original submission, this is verified over the boolean cube in Figure 3(b). We verify this phenomenon is very real over the sphere and Gaussian as well, in addition to real datasets like MNIST and CIFAR10. See Figures 6 and 7. As remarked in the paper, the existence of the optimal depth is not obvious at all, as conventional deep learning wisdom would have one believe that adding depth should always help.
> *** Our theory suggests that training the last layer should be better for learning “simpler” features, while training all layers is better for learning “complex” features. In the original submission, we verified this on the boolean cube in Figure 3(c). We show the same phenomenon holds over the sphere and standard Gaussian in new experiments. See Figure 8.
> *** We have added a new section on predicting the maximal learning rate of an MLP using the spectral theory we developed. See the new Section 7. Our prediction (1/max_eigenvalue) is derived for the boolean cube, but in fact is highly predictive for data distributed on the sphere or from standard Gaussian, as well as real datasets like MNIST and CIFAR10; see Figure 5. We hope this new finding will significantly simplify learning rate search in practical hyperparameter tuning.
> *** Note that the experiments on simplicity bias as formulated in Valle-Perez et al. (2018) require finiteness of domain, so we cannot effectively replicate them on the sphere and the Gaussian, which all have uncountable support.
>
> * New theoretical results
> *** We show that, as d -> infty, the top distinct eigenvalues of the boolean cube, the sphere, and the standard Gaussian all coincide. See the new Corollary H.21, Theorem H.19, Theorem H.13, and the old Theorem I.6.

---

### Official Review · AnonReviewer3 · 2019-10-23
**Official Blind Review #3**

**Rating:** 3

**Review:**

Aiming to resolve the question whether and why deep networks are biased towards simple functions, this paper gives a spectral analysis on neural networks' conjugate kernel(CK) and neural tangent kernel(NTK) on boolean cube. The eigenfunctions are identified and the eigenvalues are shown computable in polynomial time. Another main contribution of this paper is showing that the simplicity bias exists at least in a weak sense.

I believe that this paper should be weakly rejected because it made more claims than what it can show in that the analysis doesn't work in real space, and the authors did not really show the simplicity bias. The following are my detailed comments.

First, the whole analysis is based on boolean cube. Although the paper has shown empirically that in high dimension the uniform binary distribution is close enough to the uniform sphere distribution, it doesn't suffice to substitute boolean cube for sphere in real space. The spectral analysis in this paper is heavily due to working on boolean cube. The boolean cube is finite, which guarantees any inner-product kernel function $K(x,y) = \Phi (<x,y>)$ can be diagonalized by finite many monomial functions, And there are only O(d) different eigenvalues, which enables efficient computation. These techniques are not easy to be transferred to real space. The experiment shows that the first five eigenvalues in boolean cube, sphere, gaussian is close, but key problems here are first, in practive the dimension $d$ could be smaller and second, sphere and gaussian have infinitely many eigenvalues while boolean cube has $2^d$ eigenvalues. The experiment cannot really justify that all eigenvalues are close (only first several are shown), not to mention the tail eigenvalues over the first $2^d$-th.

Even if we assume that boolean cube is a reasonable choice, we should notice the goal of computing eigenvalues is to eventually show the inductive bias toward 'simple functions'. However, the authors failed to show it at least from the following perspectives:
1) This paper did not show the trend of eigenvalues, but only the weak version of, for example, $\mu_{2k-2} > \mu_{2k}$. In the limiting case, it is more reasonable to fix dimension $d$ rather than the degree $k$.
2) Working on boolean cube leads to limited complexity. The most complicated base function is restricted to $\mathcal{X}_S$ where $S = \{1, 2, \dots, d\}$. So the weak simplicity bias theorem actually only describes the relation among finite $d$ eigenvalues.
3) No optimization arguments appear in this paper. Based on the spectral analysis, it is not rigorous enough to claim the networks are biased to simple functions, given that the target function consists of simple multilinear monomial functions.

Since the boolean spectra is not a reliable measure, the further experiments under such a measure is therefore put under doubt.

To summarize, this paper definitely contains some rigorous analysis which I appreciate, but it made some claims that are not verified. More importantly, the boolean cube is not the appropriate domain which is hard to generalize to real space and the simplicity bias theorem in this paper is to some extent weak. Therefore, I suggest rejecting this paper in its current form.

**Experience Assessment:**

I have read many papers in this area.

**Review Assessment: Checking Correctness Of Derivations And Theory:**

I carefully checked the derivations and theory.

**Review Assessment: Checking Correctness Of Experiments:**

I assessed the sensibility of the experiments.

**Review Assessment: Thoroughness In Paper Reading:**

I read the paper thoroughly.

---

> ### Author Response · Authors · 2019-11-15
> **Rebuttal for review #3 (1)**
>
>
> Thank you Reviewer 2 for taking out the time to read our paper and provide feedback. We first summarize your main concerns below and also summarize our response. Then we address your detailed concerns one by one.
>
> Main Concern 1: The simplicity bias result is too weak.
>
> Rebuttal Summary:
> * We would first like to address possible misconceptions about the contributions of the paper, as the summary of our paper given in the review does not quite reflect its main claims. Our main contribution, as succinctly summarized at the end of Section 1, and as accurately summarized by Reviewers 1 and 3, is developing the spectral theory of the CK and NTK, and as consequences, we clarify a) the simplicity bias, b) the effect of depth on learnability, and c) the difference between CK and NTK; the latter two points are missing in the reviewer’s summary here. The simplicity bias is not the sole “punchline” of this paper, as the summary seems to suggest.
> * In addition, regarding the simplicity bias, our main point is that it is *not* universal (and thus cannot exist in a strong sense). Therefore, we cannot prove a result on the trend of the eigenvalues as the reviewer suggests, as the eigenvalues can possibly all be equal; see Figure 1. However, to say that the eigenvalues have no structure is incorrect, and the weak simplicity bias theorem (Thm 4.1) tries to paint a more accurate picture of the eigenvalues. It seems strange for the reviewer to penalize this result for stating the fact.
>
> Main Concern 2: The boolean cube theory is not reflective of the kernel spectra over the sphere and standard Gaussian, so that the conclusions over the boolean cube may not hold over these distributions.
>
> Rebuttal Summary: We verify our main claims hold over the sphere and the standard Gaussian via new experiments and theorems
>
> * New experimental results (discussed in detail in the new Section B)
> *** Our theory tells us that for each target function, there should be an optimal depth: networks that are deeper or shallower do monotonically worse (this is discussed in Section 5). In the original submission, this is verified over the boolean cube in Figure 3(b). We verify this phenomenon is very real over the sphere and Gaussian as well, in addition to real datasets like MNIST and CIFAR10. See Figures 6 and 7. As remarked in the paper, the existence of the optimal depth is not obvious at all, as conventional deep learning wisdom would have one believe that adding depth should always help.
> *** Our theory suggests that training the last layer should be better for learning “simpler” features, while training all layers is better for learning “complex” features. In the original submission, we verified this on the boolean cube in Figure 3(c). We show the same phenomenon holds over the sphere and standard Gaussian in new experiments. See Figure 8.
> *** We have added a new section on predicting the maximal learning rate of an MLP using the spectral theory we developed. See the new Section 7. Our prediction (1/max_eigenvalue) is derived for the boolean cube, but in fact is highly predictive for data distributed on the sphere or from standard Gaussian, as well as real datasets like MNIST and CIFAR10; see Figure 5. We hope this new finding will significantly simplify learning rate search in practical hyperparameter tuning.
> *** Note that the experiments on simplicity bias as formulated in Valle-Perez et al. (2018) require finiteness of domain, so we cannot effectively replicate them on the sphere and the Gaussian, which all have uncountable support.
>
> * New theoretical results
> *** We show that, as d -> infty, the top distinct eigenvalues of the boolean cube, the sphere, and the standard Gaussian all coincide. See the new Corollary H.21, Theorem H.19, Theorem H.13, and the old Theorem I.6.
>
> * There is possibly some confusion regarding $\mu_k$: this is not the $k$the largest eigenvalue, but rather the common eigenvalue associated to the $\binom d k$-dimensional eigenspace of degree $k$ polynomials over the boolean cube.

---

> ### Author Response · Authors · 2019-11-15
> **Rebuttal for review #3 (2)**
>
>
> We respond to individual concerns in detail below.
>
> > Although the paper has shown empirically that in high dimension the uniform binary distribution is close enough to the uniform sphere distribution, it doesn't suffice to substitute boolean cube for sphere in real space.
>
> As mentioned in the rebuttal summary, 1) we show the spectral closeness of the kernel over boolean cube, sphere, and Gaussian in new theoretical results; see the new Corollary H.21, Theorem H.19, Theorem H.13, and the old Theorem I.6. 2) We also verify our main claims over the sphere and Gaussian via new experiments; see the new Section 7 and Section B, as well as the new Figures 5, 6, 7, and 8.
>
> > The spectral analysis in this paper is heavily due to working on boolean cube. The boolean cube is finite, which guarantees any inner-product kernel function can be diagonalized by finite many monomial functions, And there are only O(d) different eigenvalues, which enables efficient computation. These techniques are not easy to be transferred to real space.
>
> In light of the fact that boolean cube eigenvalues are close to eigenvalues over the sphere and the Gaussian, as we show empirically in the original submission and theoretically in the updated version (see the new Corollary H.21, Theorem H.19, Theorem H.13, and the old Theorem I.6), the efficiency of computing the boolean cube eigenvalues is in fact a key contribution of our paper: We can relatively quickly compute them and gain a fairly accurate approximation of the eigenvalue picture over the sphere and the Gaussian.
>
> > The experiment shows that the first five eigenvalues in boolean cube, sphere, gaussian is close, but key problems here are first, in practive the dimension could be smaller
>
> We believe in practice the dimension is typically larger: $d = 128$ (as in our experiments) dramatically pales compared to Imagenet's input dimension $3\times256^2 = 196608$, and even to the those of the smaller common datasets like CIFAR10 ($d = 3\times 32^2 = 3072$) and MNIST ($d = 24^2 = 576$).
>
> > and second, sphere and gaussian have infinitely many eigenvalues while boolean cube has eigenvalues. The experiment cannot really justify that all eigenvalues are close (only first several are shown), not to mention the tail eigenvalues over the first $2^d$-th.
>
> We summarize our response below, but see Section H.4 in the new version of our draft for a detailed argument:
>
> We want to point out that, 1) over the boolean cube, the top 6 distinct eigenvalues $\mu_0, \ldots, \mu_5$ (shown in the Figure 18 discussed in the review) cover eigenspaces of total dimension $\binom d 0 + \cdots + \binom d 5$, which is 275,584,033 when $d = 128$, and 2) because we only have finite training data, we can ever only detect the top eigenvalues (purely out of rank considerations), and in the above example, we would need over a quarter billion samples (10 times the size of Imagenet!) to see possible differences in the 6th distinct eigenvalue. Therefore, realistically speaking, generally only the top few eigenvalues matter, and certainly that’s true for all of our main claims in the paper.

---

> ### Author Response · Authors · 2019-11-15
> **Rebuttal for review #3 (3)**
>
>
> > Even if we assume that boolean cube is a reasonable choice, we should notice the goal of computing eigenvalues is to eventually show the inductive bias toward 'simple functions'.
>
> Again, I’m afraid we have to disagree with this statement, as 1) we want to show that the strong form of the bias as observed in Valle-Perez et al. CANNOT exist universally (this is the subject of Section 4, and is clearly explained in the second sentence of that section), and the best we can hope for is the weak version we proved, and 2) our spectral theory has many applications, not just to clarify the simplicity bias, as summarized in the *Our Contributions* part of the Introduction.
>
> > 1) This paper did not show the trend of eigenvalues, but only the weak version of, for example, $\mu_{2k-2} > \mu_{2k}$.
>
> As Figure 1 and Section 4 shows, we cannot prove a result like “the eigenvalues, ranked in decreasing order, decay at a polynomial rate” because the eigenvalues can possibly all be equal.
>
> > In the limiting case, it is more reasonable to fix dimension rather than the degree.
>
> We respectfully disagree. In our experiments, we used $d=128$, where the top 6 distinct eigenvalues $\mu_0, \ldots, \mu_5$ already account for the top 275,584,033 dimensional eigenspaces, and the effect of the $\mu_6$ would require at least that many training samples to see. As discussed above, $d=128$ is quite small compared to standard datasets like MNIST, CIFAR10, and Imagenet, which all have much less data than 275,584,033; simultaneously, $d=128$ is much smaller than the input dimensions of these datasets. Therefore, it makes more sense to consider the effect of large $d$ rather than large $k$. Note that, even with $k$ fixed, when $d$ increases, the top $k$ distinct eigenvalues $\mu_0, \ldots, \mu_k$ cover an increasing number $\Theta(d^k)$ dimensions of eigenspaces. Furthermore, when $d$ is large, $\mu_0, \ldots, \mu_k$ for some fixed $k$ will account for more than 99% of the total variance of the kernel; see Corollary H.21 in the new draft.
>
> > 2) Working on boolean cube leads to limited complexity. The most complicated base function is restricted to $\mathcal \chi_S$ where $S = \{1, \ldots, d\}$. So the weak simplicity bias theorem actually only describes the relation among finite $d$ eigenvalues.
>
> We do not understand the reviewer’s claim that boolean cube limits complexity. For example, $\mathcal \chi_S$ in the review, also known as the parity function, is known to be difficult for neural networks to learn (Nye and Saxe, 2018).
>
> (Minor point: it’s actually $d+1$ eigenvalues, for degree $k = 0, 1, \ldots, d$)
>
> > 3) No optimization arguments appear in this paper. Based on the spectral analysis, it is not rigorous enough to claim the networks are biased to simple functions, given that the target function consists of simple multilinear monomial functions.
>
> Again, our main point is that the simplicity bias observed in Valle-Perez et al. (2018) is NOT universal, and our notion of “simplicity bias” follows from there, which only concern the network at initialization.
>
> > Since the boolean spectra is not a reliable measure, the further experiments under such a measure is therefore put under doubt.
>
> As summarized above, we have verified our main claims over the sphere and standard Gaussian distributions, and some of them also over MNIST and CIFAR10. As argued in our new theoretical results (see the new Corollary H.21, Theorem H.19, Theorem H.13, and the old Theorem I.6), we confirm that in high dimensions the spectrum of the kernel over the boolean cube is very close to its spectrum over the sphere and the standard Gaussian.
>
>
> Rebuttal conclusion:
>
> If we have addressed the concerns of the reviewer, please consider raising your score. Thank you!
>
>
> Maxwell Nye and Andrew Saxe. Are Efficient Deep Representations Learnable? 2018. https://arxiv.org/abs/1807.06399
> Guillermo Valle-Perez, Chico Q. Camargo, Ard A. Louis. Deep learning generalizes because the parameter-function map is biased towards simple functions. ICLR 2019. http://arxiv.org/abs/1805.08522

---

### Official Review · AnonReviewer1 · 2019-10-25
**Official Blind Review #1**

**Rating:** 6

**Review:**

This paper examined the spectrum of NNGP and NTK kernels and answer several questions about deep networks using both analytical results and experimental evidence:
* Are randomly initialized and trained deep networks biased to simple functions?
* How does this change with depth, activation function, and initialization?

All studies are conducted on a space of inputs that is a boolean cube. The input distribution is assumed to be uniform. Though it is argued in Section 3 that the results also generalize to uniform distributions on spheres and isotropic Gaussian distributions. Although this boolean cube setting is followed from previous works on the same topic, it does limit the scope of the paper. Discussions on how this assumption relates to practical problems are missing from the paper.

Putting aside the limitations of restricting the input distributions on boolean cubes (and other similar choices), I really like the paper, which demonstrates the powerfulness of spectral analysis. I also found that many analytical results (e.g., computing eigenvalues of a kernel operator with respect to uniform distributions on a boolean cube) in the paper are highly nontrivial to derive, which adds to the value of the paper. These results might seem restricted in terms of deep network theory because of the assumptions on input distributions, but I do believe the methods used can be of interest to a wider audience.

Some questions:
* In Figure 1, the 10^4 boolean function samples are sorted according to frequency (rank). What precisely is the frequency (rank) here? It shouldn't be the frequency that corresponds to the eigendecomposition because each function sample could always have multiple components with different frequencies.
* In Figure 1b, the y-axis is described as normalized eigenvalues, which seems different from degree k fractional variance defined in the next section. The degree k fractional variance is the sum of all normalized eigenvalues for degree k eigenfunctions. Is this difference intended or it is a mistake?
* Is the ground truth degree k polynomial used in experiments defined somewhere in the paper?

On writing and clarity. Overall I find this paper well-written and a pleasure to read. Some minor issues are
* The definition of "neural kernels" seems unnecessary and a bit sudden. It would be helpful to include the definition of Phi just after Eq. (2) for CK and NTK.
* For introducing boolean analysis and Fourier series, it might be better to include the formula that explicit shows the expansion f(x) = \sum_{S} f^p(S) X_S(x) before introducing Theorem 3.1.


**Experience Assessment:**

I have published one or two papers in this area.

**Review Assessment: Checking Correctness Of Derivations And Theory:**

I assessed the sensibility of the derivations and theory.

**Review Assessment: Checking Correctness Of Experiments:**

I assessed the sensibility of the experiments.

**Review Assessment: Thoroughness In Paper Reading:**

I read the paper thoroughly.

---

> ### Author Response · Authors · 2019-11-15
> **Rebuttal for review #1**
>
> Thank you Reviewer 1 for taking the time to read out work and to provide valuable feedback! We summarize your main concern below and respond with a summary of our rebuttal. Then we address your detailed concerns.
>
> Main Concern: There is limited relevance to machine learning on real datasets.
>
> Rebuttal summary: We ran new experiments on MNIST and CIFAR10 and verify our main claims hold over these datasets as well.
> * Most practically useful: We have added a new section on predicting the maximal learning rate of an MLP using the spectral theory we developed. See the new Section 7. Our prediction (1/max_eigenvalue) is derived for the boolean cube, but in fact is highly predictive for MNIST and CIFAR10 as well (in addition to data uniformly distributed on the sphere or sampled from the standard Gaussian); see Figure 5. We hope this new finding will significantly simplify learning rate search in hyperparameter tuning. We did not have enough time during the rebuttal to test whether max_lr / 2 is the best learning rate, as one would predict from a quadratic problem, but we hope to do so after the rebuttal.
> * Our theory tells us that for each target function, there should be an optimal depth: networks that are deeper or shallower do monotonically worse (this is discussed in Section 5). We verify this phenomenon is very real over MNIST and CIFAR10 as well (in addition to data uniformly distributed on the sphere or sampled from the standard Gaussian). See Figure 7.
> * Note that our other main claims concern the effect of varying the “complexity” of the target function, but the notion of “complexity” is quite subtle on MNIST and CIFAR10, unlike in our theoretical distributions (boolean cube, sphere, Gaussian), where polynomial degree is the natural complexity measure. Therefore, we did not investigate those claims on MNIST and CIFAR10.
>
> Answers to individual questions:
>
> > In Figure 1, the 10^4 boolean function samples are sorted according to frequency (rank). What precisely is the frequency (rank) here? It shouldn't be the frequency that corresponds to the Eigendecomposition because each function sample could always have multiple components with different frequencies.
>
> We are following Valle-Perez et al. in our notation here. We simply rank the 10^4 boolean function samples we drew in the order of how often they appear. For example, the most frequent functions that appear for the relu network are the “all +1” function and the “all -1” function (the left side of the blue dashed curve) that each appear with probability > 10%. You are right the word “frequency” here is ambiguous and possibly misleading, since we mean “number of times this function appears in our sample” rather than the frequency in the eigendecomposition. We have fixed this in the new version.
>
> > In Figure 1b, the y-axis is described as normalized eigenvalues, which seems different from degree k fractional variance defined in the next section. The degree k fractional variance is the sum of all normalized eigenvalues for degree k eigenfunctions. Is this difference intended or it is a mistake?
>
> This is intentional, as we wanted to show that, with erf nonlinearity and sufficiently high depth, the eigenvalues *themselves* —- not the fractional variance —- become all equal. This means that the Gaussian process corresponding to the kernel becomes white noise (but this is not the case if the fractional variances become equal).
>
> > Is the ground truth degree k polynomial used in experiments defined somewhere in the paper?
>
> In Section C in the Appendix, we have described the entire experimental procedures used for our experiments. As described there, we sample 10 monomials randomly, and then sample their coefficients randomly.
>
>
> > Minor issues
> Thanks for these helpful suggestions. We’ll update the draft accordingly.
>
> If we have addressed all of your concerns, please consider raising your score. Thanks again for the helpful review!

---

### Author Response · Authors · 2019-11-15
**Summary of Changes after Rebuttal**


1) New experimental results (discussed in detail in the new Section B)
*** We have added a new section on predicting the maximal learning rate of an MLP using the spectral theory we developed. See the new Section 7. Our prediction (1/max_eigenvalue) is derived for the boolean cube, but in fact is highly predictive for data distributed on the sphere or from standard Gaussian, as well as real datasets like MNIST and CIFAR10; see Figure 5. We hope this new finding will significantly simplify learning rate search in practical hyperparameter tuning.
*** Our theory tells us that for each target function, there should be an optimal depth: networks that are deeper or shallower do monotonically worse (this is discussed in Section 5). In the original submission, this is verified over the boolean cube in Figure 3(b). We verify this phenomenon is very real over the sphere and Gaussian as well, in addition to real datasets like MNIST and CIFAR10. See Figures 6 and 7. As remarked in the paper, the existence of the optimal depth is not obvious at all, as conventional deep learning wisdom would have one believe that adding depth should always help.
*** Our theory suggests that training the last layer should be better for learning “simpler” features, while training all layers is better for learning “complex” features. In the original submission, we verified this on the boolean cube in Figure 3(c). We show the same phenomenon holds over the sphere and standard Gaussian in new experiments. See Figure 8.

2) New theoretical results
*** We show that, as d -> infty, the top few distinct eigenvalues of the boolean cube, the sphere, and the standard Gaussian all coincide. See the new Corollary H.21, Theorem H.19, Theorem H.13, and the old Theorem I.6.

---

### Decision · Program_Chairs · 2019-12-19

**Decision:**

Reject

**Comment:**

The authors develop a spectral analysis on the boolean cube for the neural "conjugate kernel" (CK) and "tangent kernel" (NTK). The analysis sheds light into inductive biases of neural networks, such as whether they are biased to simple functions.

This work contains rigorous analysis and theory which is useful for further discussions. However, the theory and insights do not feel complete. One important drawback is that the analysis is limited by the boolean cube setting; this also means that it is more difficult to link theory to practical scenarios. This has been discussed a lot during the rebuttal and among reviewers. Empirical validation has attempted to deal with these concerns, but it would be useful to have this validation coming from theory, or at least have further relevant theoretical insights. This could happen by further building on the theorem provided in the rebuttal for eigenvalue behavior when d is large.